# Latent Geometry-Driven Network Automata for Complex Network Dismantling

**Thomas Adler**[*1,2], **Marco Grassia**[*3], **Ziheng Liao**[1,2], **Giuseppe Mangioni**[†3], **Carlo V. Cannistraci**[†1,2,4]

[1]Center for Complex Network Intelligence (CCNI), Tsinghua Laboratory of Brain and Intelligence (THBI), Department of Psychological and Cognitive Sciences
[2]Department of Computer Science and Technology, Tsinghua University
[3]Department of Electric, Electronic and Computer Engineering, University of Catania
[4]Department of Biomedical Engineering, Tsinghua University

`thomas0299@gmail.com,  marco.grassia@unict.it,  lzhalpha@gmail.com,`
`giuseppe.mangioni@unict.it,  kalokagathos.agon@gmail.com`

[*]These authors contributed equally to this work.
[†]Corresponding authors.

## Abstract

Complex networks model the structure and function of critical technological, biological, and communication systems. Network dismantling, the targeted removal of nodes to fragment a network, is essential for analyzing and improving system robustness. Existing dismantling methods suffer from key limitations: they depend on global structural knowledge, exhibit slow running times on large networks, and overlook the network's latent geometry, a key feature known to govern the dynamics of complex systems. Motivated by these findings, we introduce Latent Geometry-Driven Network Automata (LGD-NA), a novel framework that leverages local network automata rules to approximate effective link distances between interacting nodes. LGD-NA is able to identify critical nodes and capture latent manifold information of a network for effective and efficient dismantling. We show that this latent geometry-driven approach outperforms all existing dismantling algorithms, including spectral Laplacian-based methods and machine learning ones such as graph neural networks. We also find that a simple common-neighbor-based network automata rule achieves near state-of-the-art performance, highlighting the effectiveness of minimal local information for dismantling. LGD-NA is extensively validated on the largest and most diverse collection of real-world networks to date (1,475 real-world networks across 32 complex systems domains) and scales efficiently to large networks via GPU acceleration. Finally, we leverage the explainability of our common-neighbor approach to engineer network robustness, substantially increasing the resilience of real-world networks. We validate LGD-NA's practical utility on domain-specific functional metrics, spanning neuronal firing rates in the Drosophila Connectome, transport efficiency in flight maps, outbreak sizes in contact networks, and communication pathways in terrorist cells. Our results confirm latent geometry as a fundamental principle for understanding the robustness of real-world systems, adding dismantling to the growing set of processes that network geometry can explain.

## 1 Introduction

Complex networks are the backbone of our modern world, from the biological pathways within a cell to global financial and transportation systems (Newman, 2003). While the interconnected nature of these systems is often a source of efficiency and strength, it also introduces profound vulnerabilities. A localized failure can be absorbed, or it can trigger a cascade of disruptions leading to a systemic collapse. Understanding this fragility is crucial, as the consequences are far-reaching: targeted disruptions can compromise cellular function in metabolic networks, dictate the spread of a

virus through a social fabric, or cause catastrophic blackouts in power grids and failures in financial markets (Artime et al., 2024; Albert et al., 2000). The formal study of these vulnerabilities is known as network dismantling. It addresses a fundamental question: What is the most efficient way to fragment a network by removing a minimal set of nodes or links, to disrupt its structural integrity and functional capacity? Answering this question is essential not only for predicting the impact of malicious attacks but, more importantly, for designing robust and resilient systems that can withstand them. The task of dismantling serves a dual purpose. It determines whether a system is robust and how to reinforce desirable networks, for example preventing system failure in a flight network or security compromises in internet infrastructure. Conversely, it reveals how to disrupt undesirable systems, severing communications in terrorist cells or halting the spread of an epidemic. Efficient network dismantling is challenging because identifying the minimal set of nodes for optimal disruption is an NP-hard problem: no known algorithm can solve it efficiently for large networks (Artime et al., 2024), forcing the field to rely on heuristic approximations. This difficulty arises not only from the prohibitively large solution space but also from the structural complexity of real-world networks, which exhibit heterogeneous, fat-tailed connectivity (Barabási & Albert, 1999; Broido & Clauset, 2019; Voitalov et al., 2019; Serafino et al., 2021), modular and community structures (Newman, 2012), hierarchies (Ravasz & Barabási, 2003; Clauset et al., 2008), higher-order structures (Battiston et al., 2021; Lambiotte et al., 2019), and a latent geometry (Muscoloni et al., 2017; Wu et al., 2015; Boguñá et al., 2021; Krioukov et al., 2010; Serrano et al., 2008).

Node Betweenness Centrality (NBC) is a network centrality measure (Freeman, 1977) that quantifies the importance of a node in terms of the fraction of the shortest paths that pass through it. NBC-based attacks, where nodes are removed in order of their betweenness centrality, is considered one of, if not the best, method for network dismantling (Engsig et al., 2024; Servedio et al., 2025; Motter & Lai, 2002; Holme et al., 2002). However, like many other dismantling techniques, it requires global knowledge of the entire network topology, and its high computational cost limits its scalability to large networks. These limitations are shared by many other state-of-the-art dismantling methods, which additionally rely on black-box machine learning models, and are rarely validated across large, diverse sets of real-world networks (see Tables 1, 6, and 5).

Latent geometry has been recognized as a key principle for understanding the structure and complexity of real-world networks. Recent works in network science suggest that the latent geometry of complex networks could explain critical network characteristics such as small-worldness, degree heterogeneity, clustering, and navigability, and drives critical processes like efficient information flow (Boguñá et al., 2021; 2009; Kleinberg, 2000; Wu et al., 2015; Serrano et al., 2008; Krioukov et al., 2010; Muscoloni & Cannistraci, 2019; 2018a). Work by Muscoloni et al. (2017) revealed that betweenness centrality is a global latent geometry estimator: it approximates node distances in an underlying geometric space. They also introduced Repulsion-Attraction network automata rule 2 (RA2), a local latent geometry estimator that uses only first-neighbor connectivity. RA2 performed comparably to NBC in tasks such as network embedding and community detection, despite relying solely on local information. This raises the first question: can latent geometry, whether estimated globally or locally, guide effective network dismantling? If complex systems run on a latent manifold, estimating it may offer a more efficient way to disrupt connectivity. The second question concerns efficiency. While both NBC and RA2 have $O(Nm)$ complexity ($N$ as the number of nodes and $m$ the number of links), RA2 is significantly faster in practice because its local computations avoid NBC's large computational overhead. This motivates exploring whether local latent geometry estimators can match the dismantling performance of global methods like NBC while offering lower running time.

Motivated by these questions, we introduce the Latent Geometry-Driven Network Automata (LGD-NA) framework. Our first and primary contribution is the principle of (1) Latent Geometry-Driven (LGD) dismantling, where methods estimate effective node distances on a network's latent manifold to expose critical structural information. Specifically, our (2) LGD-NA framework uses local network automata rules to approximate these geometric distances; a node's summed distance to its neighbors estimates how critical it is for dismantling. Within this framework, we discovered that a (3) simple common-neighbor-based rule, which we term Common Neighbor Dissimilarity (CND), is highly effective, achieving performance close to the state-of-the-art method, NBC. We prove the effectiveness of our approach through (4) comprehensive experimental validation on an ATLAS of 1,475 real-world networks across 32 complex systems domains, the largest and most diverse collection to date, showing that LGD-NA consistently outperforms all other existing dismantling algorithms, including machine learning and spectral Laplacian-based methods. To enable dismantling at large scales,

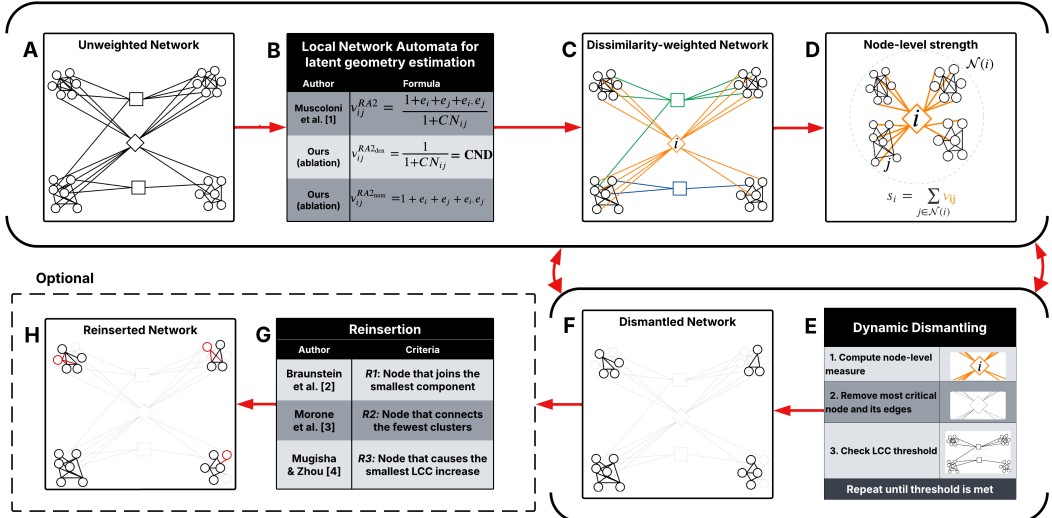

Figure 1: Overview of the LGD Network Automata framework. **A**: Begin with an unweighted and undirected network. **B**: Estimate latent geometry by assigning a weight $\nu_{ij}$ to each edge between nodes $i$ and $j$ using local latent geometry estimators. **C**: Construct a dissimilarity-weighted network based on these weights. **D**: Compute node strength as the sum of geometric weights to all neighbors in $\mathcal{N}(i)$: $s_i = \sum_{j \in \mathcal{N}(i)} \nu_{ij}$ **E–F**: Perform dynamic dismantling by iteratively computing node strengths, removing the node with the highest $s_i$ and its edges, and checking whether the normalized size of the largest connected component (LCC) has dropped below a threshold. **G–H** (optional): Reinsert dismantled nodes using a selected reinsertion method.

we implement (5) GPU-acceleration for LGD-NA, yielding remarkable running time advantages over methods like NBC. Finally, using the explainability of our CND measure, we introduce a new method for (6) engineering network robustness, substantially reducing the effectiveness of the best dismantling methods. We further validate the practical utility of our dismantling framework and robustness engineering method by demonstrating their impact on domain-specific functional metrics, including neuronal firing rates in the Drosophila Connectome, flight map efficiency, epidemic sizes, and communication reachability in terrorist cells.

## 2 RELATED WORK

**Latent Geometry of Complex Networks.** Many real-world networks are shaped by latent geometric manifolds of the complex systems that govern their topology and dynamics. These hidden geometries explain essential structural features such as small-worldness, degree heterogeneity, clustering, and community structure (Boguñá et al., 2021; Wu et al., 2015; Serrano et al., 2008; Muscoloni & Cannistraci, 2018a; Zuev et al., 2015; Muscoloni & Cannistraci, 2018b; Muscoloni et al., 2017). The underlying metric space is not only descriptive but functional: it facilitates efficient routing and navigation with limited global knowledge (Kleinberg, 2000; Boguñá et al., 2009; Krioukov et al., 2010; Muscoloni & Cannistraci, 2019). Such properties emerge consistently across diverse systems, including biological, social, technological, and socio-ecological networks (Boguñá et al., 2021; Wu et al., 2015). Latent geometries also enable predictive modeling of dynamical processes such as network growth (Papadopoulos et al., 2012; Muscoloni & Cannistraci, 2018b;a), and epidemic spreading (Brockmann & Helbing, 2013).

**Latent Geometry Estimators.** Latent geometry estimators assign edge weights to approximate pairwise distances of linked nodes in the hidden geometric manifold. Among them, network automata rules based on the Repulsion-Attraction (RA) criterion use only local topological information to infer proximity in the latent space (Muscoloni et al., 2017). RA is grounded in the theory of network navigability (Boguñá et al., 2009), which posits that nodes with many non-overlapping neighbors tend to occupy distant regions in the latent space. Edges between such nodes receive higher dissimilarity scores due to strong repulsion, while those with many common neighbors are scored lower due

Table 1: Number of real-world networks tested by dismantling algorithms (see Table 10).

| Algorithm | Year | Networks | Ref. |
|---|---|---|---|
| Collective Influence (CI) | 2016 | 2 | Morone et al. (2016) |
| CoreHD | 2016 | 12 | Zdeborová et al. (2016) |
| Explosive Immunization (EI) | 2016 | 5 | Clusella et al. (2016) |
| Min-Sum (MS) | 2016 | 2 | Braunstein et al. (2016) |
| GND | 2019 | 10 | Ren et al. (2019) |
| Resilience Centrality | 2020 | 4 | Zhang et al. (2020) |
| GDM | 2021 | 57 | Grassia et al. (2021) |
| CoreGDM | 2023 | 15 | Grassia & Mangioni (2023) |
| Domirank Centrality | 2024 | 6 | Engsig et al. (2024) |
| Fitness Centrality | 2025 | 5 | Servedio et al. (2025) |
| **LGD-NA** | **2025** | **1,475** | **Ours** |

Table 2: Summary of real-world networks tested in this paper (see Table 9).

| Field | Subfields | Types | Networks |
|---|---|---|---|
| Biomolecular | 5 | PPI, Genetic, Metabolic, Molecular, Transcription | 27 |
| Brain | 1 | Connectome | 529 |
| Covert | 2 | Covert, Terrorist | 89 |
| Foodweb | 1 | Foodweb | 71 |
| Infrastructure | 7 | Flight, Nautical, Power grid, Rail, Road, Subway, Trade | 314 |
| Internet | 1 | Internet | 206 |
| Misc | 8 | Citation, Copurchasing, Game, Hiring, Lexical, Phone call, Software, Vote | 38 |
| Social | 7 | Coauthorship, Collaboration, Contact, Email, Friendship, Social network, Trust | 201 |
| **Total** | **32** | | **1,475** |

to attraction. RA1 and RA2 are network automata rules for approximating linked nodes' pairwise distances on the latent manifold of a complex network. These rules are categorized as network automata because they adopt only local information to infer the score of a link in the network without the need for pre-training of the rule. Note that RA1 and RA2 are **predictive network automata** that differ from generative network automata, which are rules created to generate artificial networks (Barabási & Albert, 1999; Papadopoulos et al., 2012; Muscoloni & Cannistraci, 2018b). They were introduced to serve as pre-weighting strategies for approximating angular distances associated with node similarities in hyperbolic network embeddings. RA2 performed slightly better than RA1, so for this reason we will only consider RA2 in this study. RA2 defines dissimilarity between nodes $i$ and $j$ as:

$$\text{RA2}(i,j) = \frac{1 + e_i + e_j + e_i \cdot e_j}{1 + \text{CN}_{ij}}.$$

where $\text{CN}_{ij}$ is the number of common neighbors of nodes $i$ and $j$, and $e_i$ and $e_j$ are the external degrees of $i$ and $j$, representing the count of neighbors of $i$ and $j$ that are not involved in the common neighbors interactions. In the same work, Muscoloni et al. also showed that betweenness centrality is a global latent geometry estimator. By comparing it with RA2, they demonstrated that both global (betweenness centrality) and local (RA) estimators can effectively capture latent geometry, achieving strong results in network embedding and community detection. See Table 3 for a comparison of estimators and Figure 5 for illustrative examples. See also Appendix C, where we validate the ability of these latent-geometry estimators in identifying node importance and estimate link distances in networks with a known geometry.

**Topological centrality measures.** Degree, betweenness centrality, and their variants have all been used in the majority of dismantling studies (Artime et al., 2024), with betweenness centrality having been found to be the most effective strategy when applying dynamic dismantling, meaning the scores are recomputed after every step. Degree centrality ranks nodes by their number of neighbors, and betweenness centrality (Freeman, 1977) counts how frequently a node lies on shortest paths. Other centrality variants include eigenvector centrality (Bonacich, 1972), which gives higher scores to nodes connected to other influential nodes. PageRank (Page et al., 1999), based on a random walk model, favors nodes that receive many and high-quality links. Beyond these classical measures, several centrality indices have been developed specifically to capture aspects of network resilience. Fitness centrality (Servedio et al., 2025), adapted from economic complexity theory, evaluates

node importance through the capabilities of neighbors while penalizing connections to weak nodes. DomiRank (Engsig et al., 2024) centrality models a competitive dynamic in which nodes gain or lose dominance, or importance, based on the relative strength of their neighbors. Resilience centrality (Zhang et al., 2020), derived from a dynamical systems reduction, quantifies how a node's removal alters the system's resilience (see Table 5).

**Statistical and Machine Learning Network Dismantling.** We focus on network dismantling for targeted attacks, where the goal is to fragment a network as efficiently as possible by removing selected nodes. Message passing-based methods such as Belief Propagation-guided Decimation (BPD) (Mugisha & Zhou, 2016) and Min-Sum (MS) (Braunstein et al., 2016) use message-passing algorithms to decycle the network and then fragment the resulting forest with a tree-breaker algorithm, while CoreHD (Zdeborová et al., 2016) achieves decycling by iteratively removing the highest-degree nodes from the 2-core of the network and also includes a tree-breaker algorithm. Decycling and dismantling are, in fact, closely related tasks, as a tree (or a forest) can be dismantled almost optimally (Braunstein et al., 2016). Generalized Network Dismantling (GND) (Ren et al., 2019) targets nodes that maximize an approximated spectral partitioning. Collective Influence (CI) (Morone et al., 2016) targets nodes with maximal influence on their neighborhoods, and Explosive Immunization (EI) (Clusella et al., 2016), uses explosive percolation dynamics. Machine learning-based methods include Graph Dismantling with Machine Learning (GDM) (Grassia et al., 2021), which trains graph neural networks to predict optimal attack strategies in a supervised manner. FINDER (Fan et al., 2020b) uses reinforcement learning instead to autonomously learn dismantling strategies without needing labeled data. CoreGDM (Grassia & Mangioni, 2023) combines ideas from CoreHD and GDM as it attacks the 2-core of the network but uses machine learning models trained on optimal dismantling solutions to guide node removal (see Table 6).

## 3 Latent Geometry-Driven Network Automata

We introduce the Latent Geometry-Driven Network Automata (LGD-NA) framework. LGD-NA adopts a parameter-free network automaton rule, such as RA2, to estimate latent geometric linked node pairwise distances and to assign edge weights based on these geometric distances. Then, it computes for each node its network centrality as a sum of the weights of adjacent edges. The higher this sum, the more a node dominates numerous and far-apart regions of the network, becoming a prioritized candidate for a targeted attack in the network dismantling process. This prioritized node is then removed from the network, and the procedure is iteratively repeated until the network is dismantled (see Figure 1 for a full breakdown).

### 3.1 Latent Geometry-Driven dismantling

Our first contribution is Latent Geometry-Driven (LGD) dismantling, where any function can be used to estimate edge weights that represent effective distances between nodes, capturing the network's underlying latent geometry. These inferred weights are used to construct a dissimilarity-weighted network, encoding a hidden geometric structure beneath the observable topology and allowing the dismantling process to prioritize nodes according to their geometric centrality in the latent manifold. Latent geometric structures have been shown not only to explain key properties of complex networks, but also to support the understanding of dynamical processes such as navigation, routing, and epidemic spreading. Building on the idea that network geometry captures essential structural and dynamical properties of complex systems, LGD dismantling is guided by a geometric intuition about how nodes connect distant regions in the latent space. If two nodes are connected to many different nodes but have little overlap in their neighborhoods, they are likely to be far apart in the network's latent space. An edge between them, therefore, connects distant regions of the network. A node that has many such edges is central to holding the network together, as it links otherwise separate areas. We propose that removing those geometrically central nodes is an effective way to fragment the network. Muscoloni et al. (2017) also offered evidence that betweenness centrality can be used as a latent geometry estimator, hence, NBC is a global topology centrality measure which can be used for latent geometry-driven dismantling.

### 3.2 LGD NETWORK AUTOMATA

Our second contribution is the introduction of a parameter-free network automaton framework for LGD dismantling. In this framework, node importance is estimated by aggregating edge geometric weights into node strengths, and the network is dismantled iteratively by removing the nodes with the highest strength and all their edges. The underlying intuition is that nodes that connect to many external, non-overlapping regions are geometrically central and thus more structurally important, leading to higher strength values. Formally, we begin with an undirected, unweighted network without isolated components. A network automaton rule, such as RA2, that is able to adopt local topology to estimate latent geometry, is applied to assign a weight $\nu_{ij}$ to the edge between node $i$ and node $j$, representing the estimated geometric distance between the two nodes. We get a dissimilarity-weighted network from these edge weights. The strength $s_i$ of node $i$ is then calculated by summing the geometric weights of all its edges, that is, the weights to all its neighbors in the set $\mathcal{N}(i)$:

$$s_i = \sum_{j \in \mathcal{N}(i)} \nu_{ij}.$$

In this paper, we adopt three types of LGD network automata rules. The first rule is RA2, which was proposed by Muscoloni et al. (2017) for hyperbolic network embedding purposes. The second rule is proposed in this study as an ablation test of the RA2 rule. It is the denominator of the RA2, which we call common neighbors dissimilarity (CND), defined as:

$$\nu_{ij} \rightarrow \text{CND}(i, j) = \frac{1}{1 + \text{CN}_{ij}}.$$

where $\text{CN}_{ij}$ is the number of common neighbors between nodes $i$ and $j$. Here, the lower the number of common neighbors two interacting nodes have, the more geometrically distant they are, and thus a higher edge weight is assigned between these two nodes. The rationale for proposing a network automaton rule based only on the common neighbors denominator term of RA2 is to account for the mere attraction between a node and its neighbors. Neglecting the repulsion part associated with the external links (the numerator of RA2) makes sense in a dismantling task because any time we compute the common neighbors of a seed node with one of its neighbors, we indirectly account for the exclusion of nodes that are not in the topological neighborhood of the seed node. For completeness, we also investigate a third rule as an ablation test of RA2 in which we consider only the external links term in the RA2 numerator, expecting that the mere RA2 numerator should also work, but not as well as the common neighbor-based denominator. Indeed, a previous study offers evidence that common neighbors are among the topological features most associated with community organization and mesoscale network geometry (Bianconi et al., 2014).

## 4 EXPERIMENTS

### 4.1 EVALUATION PROCEDURE

We evaluate all dismantling methods using a widely accepted procedure in the field of network dismantling (Artime et al., 2024). For each method, nodes are removed sequentially according to the order it defines. After each removal, we track the normalized size of the Largest Connected Component (LCC), defined as the ratio $\text{LCC}(x)/|N|$, where $|N|$ is the total number of nodes in the original network and $\text{LCC}(x)$ is the number of nodes in the largest component after $x$ removals. This process continues until the LCC falls below a predefined threshold. A commonly accepted threshold in dismantling studies is 10% of the original network size. To quantify dismantling effectiveness, we compute the Area Under the Curve (AUC) of the LCC trajectory throughout the removal process, which records the normalized LCC size at each step. A lower AUC indicates a more efficient dismantling, as it reflects an earlier and sharper disruption of network connectivity. The AUC is computed using Simpson's rule. See Figures 6, 10, 14, and 19 for visual illustrations of the LCC curve.

### 4.2 OPTIONAL REINSERTION STEP

After reaching the dismantling threshold, we optionally perform a reinsertion step to reduce the dismantling cost, defined as the number of removals. Nodes are sequentially reinserted back into

the network, one at a time, until the LCC of the remaining network just meets or exceeds the predetermined dismantling threshold. Reinsertion can significantly improve dismantling performance; recent work shows that simple heuristics with reinsertion can match or outperform complex algorithms that include reinsertion by default (Fan et al., 2020a). As a result, we enforce two constraints to ensure the reinsertion step does not override the original dismantling method: (1) reinsertion cannot reinsert all nodes to recompute a new dismantling order, and (2) reinsertion must use the reverse dismantling order as a tiebreak. If a method includes reinsertion by default, we also evaluate its performance without reinsertion for a fair comparison (see Table 7).

## 4.3 ATLAS DATASET

Our fourth contribution is the breadth and diversity of real-world networks tested in our experiments, demonstrating the generality and robustness of LGD-NA across domains and scales. We build an ATLAS of 1,475 real-world networks across 32 complex systems domains, which is the largest and most diverse collection of real-world networks to date used for testing in network dismantling studies. We first test all methods across networks of up to 5,000 nodes and 205,000 edges without reinsertion ($n = 1,296$), and 38,000 edges with reinsertion ($n = 1,237$). To assess the practical running time of the best performing methods, we evaluate NBC and RA2 on even larger networks of up to 23,000 nodes and 507,000 edges ($n = 1,475$). Current state-of-the-art dismantling algorithms have been evaluated on no more than 57 real-world networks (see Table 1), with most algorithms tested on fewer than a dozen. Our experiments cover 1,475 networks, representing a substantial expansion. A key aspect of our ATLAS dataset is the diversity of network types (see Table 2). We test across 32 different complex systems domains, ranging from protein-protein interaction (PPI) to power grids, international trade, terrorist activity, ecological food webs, internet systems, brain connectomes, and road maps. Since fields vary in both the number of networks and their characteristics, we evaluate dismantling methods using a mean field approach, ensuring that fields with more networks do not dominate the overall evaluation. Also, because dismantling performance varies in scale across fields, we compute a mean field ranking to make results comparable across domains.

## 4.4 LGD-NA PERFORMANCE AND COMPARISON TO OTHER METHODS

We compare our LGD-NA framework against the best-performing dismantling algorithms in the literature. Main results are visualized in Figure 2, and full quantitative results, including side-by-side comparisons of absolute AUC and mean-field ranks for all methods and fields, are reported in Tables 23 through 35 in the Appendix.

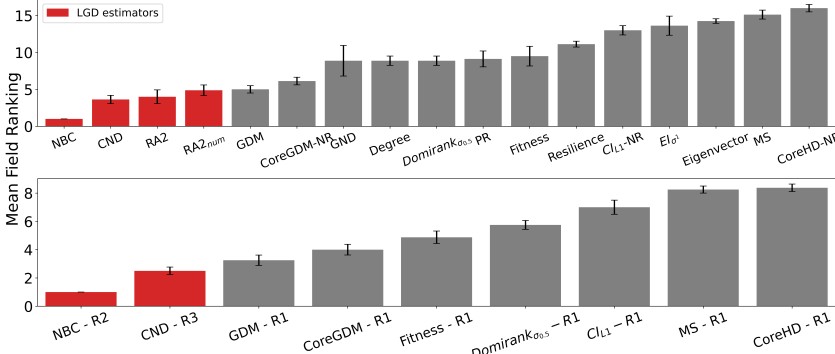

Figure 2: Mean field ranking for each dismantling method without reinsertion ($n = 1,296$; upper panel) and with reinsertion ($n = 1,237$; lower panel), for dynamic dismantling. In the lower panel, a subset of the best-performing methods from each category is paired with their respective best-performing reinsertion strategy. Methods based on latent geometry are shown in red. NR denotes variants where the original reinsertion step was disabled. Error bars indicate the standard error of the mean (SEM).

First, we find that all latent geometry network automata, NBC, RA2, and its variants, achieve top dismantling performance, both with and without reinsertion. These findings show that estimating the latent geometry of a network effectively reveals critical nodes for dismantling, confirming our first contribution. For each method, we evaluate three reinsertion strategies and report the best

result. We show in Figure 9 that using different reinsertion methods does not change the mean field ranking of the dismantling methods, and in Figures 11 and 12 that the improvement in performance varies across fields and reinsertion methods (see Figure 10). We also adopt a dynamic dismantling process for the network automata rules and all centrality measures, where we recompute the scores after each dismantling step, as it consistently outperforms the static variant (see Figure 13 for an example of the improvements for CND and Figure 14). Second, we find that local network automata rules RA2, CND, and RA2$_{num}$, which adopt only the local network topology around a node, are highly effective. In particular, RA2 and its variants consistently outperform all other non-latent geometry-driven dismantling algorithms, including those relying on global topological measures or machine learning. This confirms our second contribution. See Figure 6 for illustrative examples where the local network automata rules outperform NBC. In addition, Appendix C validates the ability of our latent-geometry-based network automata rules in identifying node importance and estimating latent geometric distances. Third, we find that the simplest RA2 variant, based solely on inverse common neighbors, which we refer to as common neighbor dissimilarity (CND), achieves the best performance among all local network automata rules. This is our third contribution and demonstrates that even minimal local topology-based information can effectively approximate latent geometry useful for effective dismantling. NBC strictly dominates as the top-ranking method across all fields. However, among the second-best performers, the LGD-NA methods lead in the majority of domains: CND ranks second in Internet networks, RA2 in Biomolecular and Brain networks, and RA2$_{num}$ in Covert networks. The only fields where non-LGD-NA methods rank second are Foodweb (Fitness Centrality), Infrastructure (GDM), and Social networks (GND). LGD-NA consistently outperforms all other non-latent geometry-driven dismantling algorithms, including those relying on spectral Laplacian-based methods and machine learning. The only measure that still outperforms LGD-NA is the NBC metric (also latent-geometry-driven), applied to dynamic dismantling. These results strongly demonstrate the practical reliability of our latent geometry-driven dismantling framework, LGD-NA.

## 4.5 GPU ACCELERATION OF LGD-NA FOR LARGE-SCALE DISMANTLING

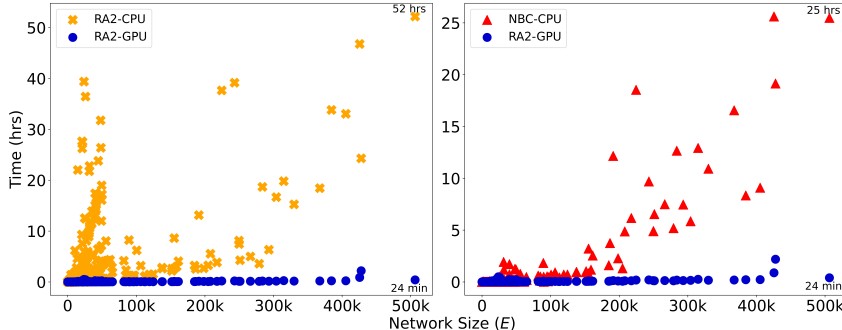

Figure 3: Runtime (in hours) is plotted against network size, measured by the number of edges, $E$, for dynamic dismantling. The annotated time indicates the runtime for the largest network. Evaluated on networks of up to 23,000 nodes and 507,000 edges ($n = 1{,}475$).

We implement GPU acceleration for all three LGD-NA variants by reformulating the required computations as matrix operations. On large networks, this enables a significant speedup in running time. When comparing RA2 and NBC, on the largest network, GPU-accelerated RA2 is 130 times faster than its CPU counterpart, highlighting the inefficiency of matrix multiplication on CPU. It is also over 63 times faster than NBC running on CPU, thanks to our GPU-optimized implementation. Note that NBC on CPU remains faster than RA2 on CPU, again due to the limitations of CPU-based matrix operations. We report only the CPU running time for NBC, as its GPU implementation did not yield any speedup (see Table 15). While some studies report GPU implementations of NBC with improved performance (Fan et al., 2017; Shi & Zhang, 2011; Pande & Bader, 2011; McLaughlin & Bader, 2018; Sariyüce et al., 2013; Bernaschi et al., 2016), these are often limited by hardware-specific optimizations, data-specific assumptions (e.g., small-world, social, or biological networks), and the use of heuristics that are tailored to specific settings rather than offering general solutions. Moreover, publicly available code is rare, making these approaches difficult to reproduce or integrate. Overall, NBC is not naturally suited for GPU implementation, as it does not rely on matrix multiplication, but is based on computing shortest path counts between all node pairs. Overall, while NBC achieves better dismantling performance, its high computational cost makes it impractical for large-scale use.

In contrast, thanks to our GPU-optimized implementation, our local latent geometry estimators based on network automata rules are the only viable option for efficient dismantling at scale. Here, we look at the details of our matrix operations for the LGD-NA measures. First, the common neighbors matrix is computed as

$$\mathbf{CN}_{\text{L2}} = \mathbf{A} \circ (\mathbf{A}^2)$$

where $\mathbf{A}$ is the adjacency matrix and $\circ$ denotes element-wise multiplication. Here, $\mathbf{A}^2$ counts the number of paths of length two (i.e., common neighbors) between all node pairs. The Hadamard product with $\mathbf{A}$ ensures that values are only retained for existing edges. Next, we compute the number of external links a node has relative to each of its neighbors. Given the degree matrix $\mathbf{D}$, the external degree matrix is:

$$\mathbf{E}_{\text{L2}} = \mathbf{A} \circ (\mathbf{D} - \mathbf{CN}_{\text{L2}} - \mathbf{A})$$

Each entry $(i, j)$ of $\mathbf{E}_{\text{L2}}$ represents the external degree of node $i$ with respect to node $j$: the number of neighbors of $i$ that are neither connected to $j$ nor directly connected to $j$ itself. Non-edges are zeroed out. These matrices allow efficient construction of RA2 and its variants using only matrix operations. The time complexity is $\mathcal{O}(N^3)$, with the common neighbor matrix being the dominant operation, for dense graphs, and $\mathcal{O}(Nm)$ for sparse graphs, $N$ being the number of nodes and $m$ the number of links. On CPU, matrix multiplication is typically memory-bound and limited by sequential operations. GPUs, however, are optimized for matrix operations, leveraging thousands of parallel threads. This results in a substantial speedup when implementing the GPU version. Finally, we show in Appendix J that in controlled settings with nPSO networks the GPU advantage becomes apparent when networks exceed 1,000 nodes or 100,000 edges.

### 4.6 LEVERAGING CND EXPLAINABILITY TO ENGINEER NETWORK ROBUSTNESS

A key advantage of our LGD-NA framework is its explainability. Indeed, we can directly explain why any of our network-automata-based and latent-geometry-driven measures prioritize specific nodes for dismantling. CND, our most performant network automata rule for dismantling, makes this explainability even more straightforward and shows that the vulnerability of a node is strongly related to the number of links its neighbors share with one another. The higher this number, the more common neighbors exist between the adjacent nodes of a vulnerable target node. This means that, to enhance the robustness of the network to the failure of a critical node, we should simply increase the number of links between its adjacent nodes. The strategy is as follows. First, identify the nodes with the highest dismantling scores according to a given measure. Here, we consider NBC, a global shortest-path count-based measure, and CND, a local topology common-neighbor-based network automata measure, because they use different rationales to estimate critical nodes and are the two best-performing measures in this study. Second, for these critical nodes, add new links between their adjacent nodes that are not already connected to each other. Robustness is defined as the ability of a system to continue functioning when subjected to perturbations (Artime et al., 2024). In this initial context, we define attack tolerance, quantified by the LCC AUC, as a robustness measure itself, representing the system's structural integrity under dismantling attacks. We validate our reinforcement strategy in Table 12 and Figure 15. We clearly show that adding links between the adjacent nodes of the most critical nodes significantly increases the AUC—and therefore the robustness—by 36% to 95% for 1% of added links, and by 59% to 259% for 10% of added links. Remarkably, by reinforcing only the top 1% of nodes, we increase network robustness regardless of the dismantling method used—whether it is our CND or NBC.

### 4.7 REAL-WORLD APPLICATIONS: FAULT TOLERANCE, SECURITY, AND COMMUNICATIONS

To demonstrate the practical utility of LGD-NA, we evaluate its performance on four distinct real-world systems using domain-specific functional metrics. First, we use the Drosophila Connectome (Shiu et al., 2024), where we utilize a Spiking Neural Network (SNN) model of the sugar-sensing circuit. The metric is the sensory neuron firing rate required to trigger the proboscis extension response. Second, the Terrorist Cell (Gutfraind & Genkin, 2017), where we analyze the network responsible for the 2015 Paris and 2016 Brussels attacks. The metric is Commander Reach, defined as the percentage of operatives able to communicate with at least one of the three key commanders. Third, the Flight Map (Cardillo et al., 2013), where we measure Global Efficiency ($E_{glob}$). Fourth, a School Contact Network (Mastrandrea et al., 2015), where we simulate an epidemic using an SEIR model (Anderson & May, 1991). The metric is the Final Outbreak Size. Our results in Figure 4 show that dismantling strategies effectively degrade the functional performance across all four systems. In

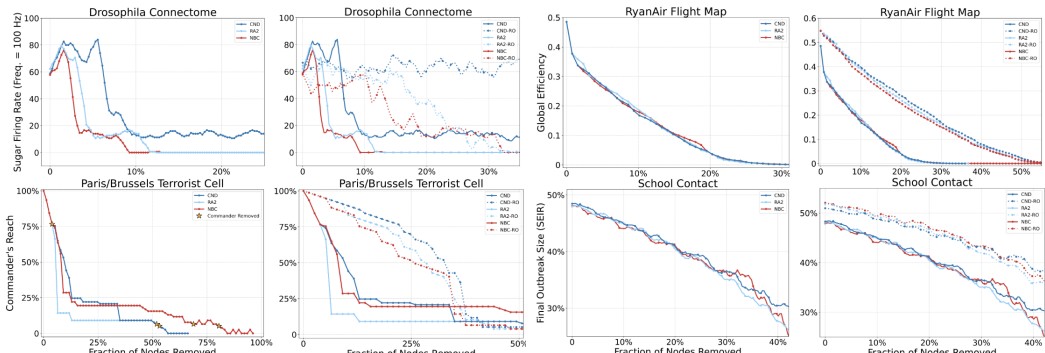

Figure 4: Dynamic dismantling process for four real-world networks with field-specific functional metrics, for NBC, CND, and RA2. The final evaluation metric is the Area Under the Curve (AUC). Dashed line represents the dynamic dismantling process for reinforced networks. See Figure 13 for full results.

the Drosophila Connectome and Terrorist Cell, we observe particularly sharp drops in performance metrics after removing only a small fraction of nodes ( 5%). We observe a more gradual deterioration in the global efficiency of the Flight Map and the viral spread within the School Contact network. This functional collapse is particularly significant for the two adversarial scenarios (Terrorist Cell and School Contact Network): it confirms that LGD-NA is effective for security and communication disruption, efficiently suppressing epidemic outbreaks and isolating hostile leadership with minimal intervention. We subsequently applied our strategy for engineering network robustness to these four scenarios, demonstrating its effectiveness. As shown in Table 13, the reinforced networks are significantly harder to dismantle, achieving robustness gains of up to 363%. This increased resilience is evident across both our original topological metric (LCC AUC) and the domain-specific functional metrics defined for each case. For the Drosophila Connectome, this analysis informs the resilient and redundant design of fault-tolerant neuromorphic circuits by mimicking its biological wiring (Suárez et al., 2021; Ham et al., 2021). In the Flight Map, it identifies specific hubs where reinforcement prevents systemic failure. Finally, for adversarial networks, our robustness analysis serves a diagnostic purpose when faced with incomplete data. Since social networks, and especially covert ones, often contain unobserved links (e.g., dormant ties or unreported contacts), calculating an empirical robustness ceiling allows us to estimate the margin of error required for successful security operations with partial observability.

## 5 CONCLUSION

We acknowledge the dual-use potential of this research, as understanding network vulnerabilities is critical for both designing targeted attacks and engineering robust defensive strategies. To mitigate this, we proactively demonstrate a constructive application for enhancing network robustness and believe the societal benefit of openly publishing these defensive tools outweighs the risk of misuse. In summary, we introduced Latent Geometry-Driven Network Automata (LGD-NA), a framework that achieves state-of-the-art network dismantling using only local topological information. By applying simple network automata rules to estimate a network's latent geometry, LGD-NA identifies critical nodes with significant speed advantages over global methods like Node Betweenness Centrality (NBC). Across 1,475 real-world networks and 32 complex systems domains, it consistently outperforms all other dismantling algorithms, including those based on machine learning (e.g., Graph Neural Networks) and spectral Laplacian-based ones. Notably, our minimalistic Common Neighbor Dissimilarity (CND) measure matches NBC's efficacy while being orders of magnitude faster. Leveraging the explainability of CND, we introduce a novel strategy to engineer network robustness. Crucially, we demonstrate the practical utility of our framework across diverse domains, from informing the design of neuromorphic circuits and reinforcing transport hubs, to disrupting terrorist cells. This work establishes latent geometry as a powerful and efficient principle for both explaining vulnerabilities and engineering stronger networks.

ACKNOWLEDGEMENTS

This work was supported by The National Natural Science Foundation of China grant number W2531064 to CVC. The Zhou Yahui Chair Professorship award of Tsinghua University to CVC. The National High-Level Talent Program of the Ministry of Science and Technology of China grant number 20241710001 to CVC. MG and GM acknowledge partial financial support from the University of Catania in the form of a PIAno di inCEntivi per la Ricerca di Ateneo (PIACERI). The Authors thank Mo Yang for her administrative support.

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

# A    LATENT GEOMETRY ESTIMATORS

Table 3: Comparison of latent geometry estimators and their variants. $\nu_{ij}$ is the weight of the link between nodes $i$ and $j$; $e_i$ and $e_j$ denote the number of external links of nodes $i$ and $j$, respectively; $\text{CN}_{ij}$ is the number of common neighbors shared by $i$ and $j$. *Information Locality* denotes the type of structural information required to assign a score to each node for dismantling. *Time Complexity* denotes the time complexity for dynamic dismantling using each estimator on sparse graphs, without reinsertion. $N$: number of nodes. $m$: number of links.

| Estimator | Author | Year | Formula | Information Locality | Time Complexity |
|---|---|---|---|---|---|
| Repulsion Attraction 2 | Muscoloni et al. (2017) | 2017 | $\nu_{ij}^{RA2} = \frac{1+e_i+e_j+e_i.e_j}{1+CN_{ij}}$ | Local | $\mathcal{O}(N(Nm))$ |
| RA2 denominator-ablation (CND) | Ours | 2025 | $\nu_{ij}^{RA2\text{den}} = \frac{1}{1+CN_{ij}} = CND$ | Local | $\mathcal{O}(N(Nm))$ |
| RA2 numerator-ablation | Ours | 2025 | $\nu_{ij}^{RA2\text{num}} = 1 + e_i + e_j + e_i.e_j$ | Local | $\mathcal{O}(N(Nm))$ |

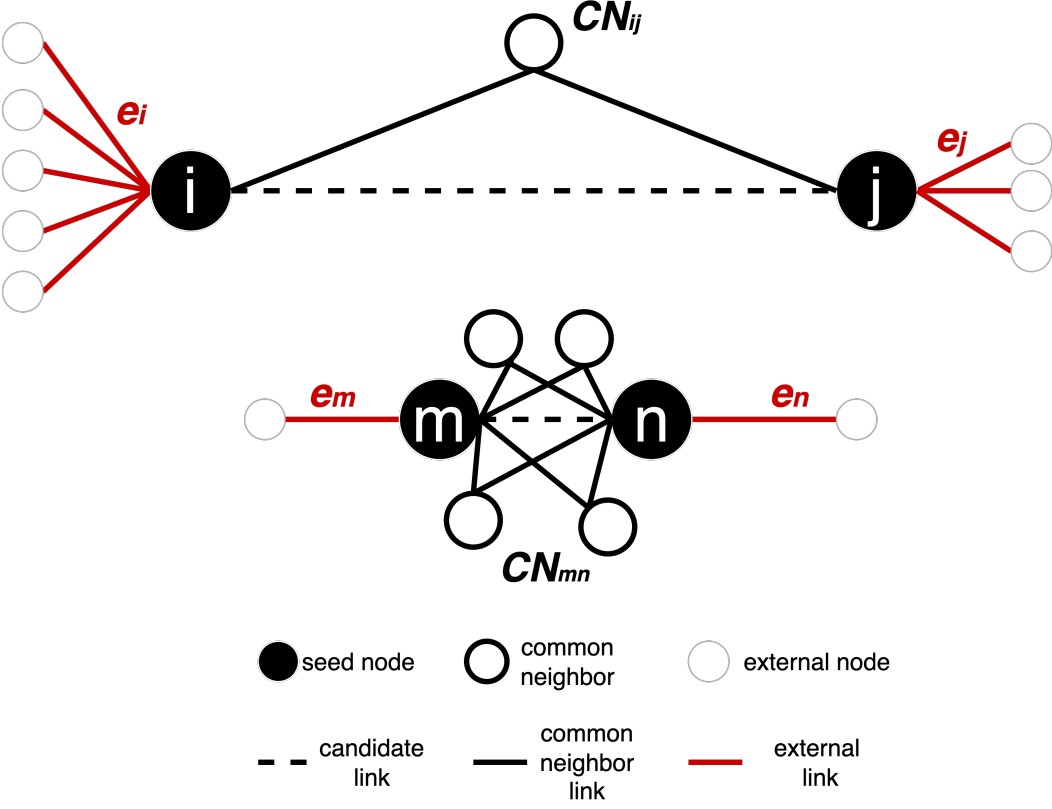

Figure 5: Illustration of how RA2 measures are computed on two toy networks. Seed nodes are shown in black; common neighbors (CN) are shown in white with a black border, and external nodes are white with a grey border. The dashed line is the edge that is being assigned a weight. External links $e$ denote the number of edges connecting a node to nodes outside its CN set, here in red. In black, the links to common neighbors. For the link $\nu_{ij}$ in the top network, $e_i = 5$, $e_j = 3$, and $\text{CN}_{ij} = 1$. For the link $\nu_{mn}$ in the bottom network, $e_m = 1$, $e_n = 1$, and $\text{CN}_{mn} = 4$.

**Latent Geometry-Driven Network Automata rule.**    Figure 5 illustrates how RA2-based network automata rules assign edge weights by estimating geometric distances using only local topological features. The two toy subnetworks demonstrate how the RA2 rule and its variants distinguish between geometrically distant and close node pairs. In the top subnetwork, nodes $i$ and $j$ have only one common neighbor and are each connected to many external nodes ($e_i = 5$, $e_j = 3$), indicating a weak integration in a local community and stronger connectivity to distinct parts of the network. According

to the Repulsion-Attraction rule, this suggests a larger latent distance due to high repulsion and low attraction. In contrast, in the bottom subnetwork, nodes $m$ and $n$ share four common neighbors and have only one external link each ($e_m = 1$, $e_n = 1$). This pattern indicates a stronger local community and a higher likelihood that the nodes are geometrically close in the latent space, with a lower dissimilarity score. These examples highlight how latent geometry-driven RA2-based network automata rules estimate hidden distances: fewer common neighbors and more external links suggest geometrical separation, while many common neighbors and few external links imply proximity in the latent manifold.

**Why is RA2 a latent geometry estimator?** In geometric networks, nearby nodes form dense, closed neighborhoods: they share many common neighbors (high $CN_{ij}$) and have few "external" links (small $e$). Distant node pairs show the opposite pattern. The Repulsion-Attraction rule 2 (RA2) captures these patterns in their formulation: any RA variant that decreases with $CN_{ij}$ and increases with external connectivity, $e$ is therefore monotonic with latent distance: small values indicate proximity; large values indicate separation. Crucially, this relies on topological proximity, not a specific geometric space, so it applies across hyperbolic, Euclidean, or elliptic latent geometries. The only assumption to apply this estimator of underlying geometry is that the topology displays:

- node heterogeneity (meaning that the node degree distribution displays a standard deviation different from zero).
- homophily (similar nodes link together), for instance geometric proximity in latent space causes nodes that are geometrically close have overlapping neighborhoods.

We can aggregate RA2 to score node criticality by summing its pairwise RA2 to neighbors. This turns local edge-level "distance" into a node-level bridging load:

- Few adjacent nodes (neighbors) with mostly short links yields a small RA2. This node is peripheral and non-critical; removal has little global effect.
- Many neighbors with mostly short links still yields a modest RA2 (short links contribute little). The node is locally redundant; removal is buffered by community structure.
- Many neighbors with a mix of short and long links yields a large RA2 because long links carry high RA2. The node simultaneously anchors a local community and bridges distant regions; removing it is likely to disconnect communities and degrade global connectivity.
- Few neighbors with many long links (rare under geometric attachment) still yields a large RA2; such nodes are likewise critical inter-community hubs.

As a result, RA2 encodes latent separation from purely local topology. Summing RA2 over a node's incident edges ranks nodes by how much long-range connectivity they support. Dismantling the highest-scoring nodes precisely targets those bridges whose removal most effectively fragments the network.

**Time Complexity.** We analyze the time complexity for the full dynamic dismantling process (excluding reinsertion) for the latent geometry-driven network automata rules in Table 3, where dynamic means recomputing the dismantling measure after each node removal. For RA2 and its variants, the dominant operation is the computation of the common neighbor (CN) matrix. This operation has a time complexity of $\mathcal{O}(N^3)$ for dense graphs and $\mathcal{O}(Nm)$ for sparse graphs, where $N$ is the number of nodes and $m$ is the number of links. Assuming $N$ dismantling steps in the worst-case scenario, the overall time complexity becomes $\mathcal{O}(N(Nm))$ for sparse graphs. The assumption of $N$ dismantling steps applies to all the time complexity analyses of dynamic dismantling methods.

## B   THEORETICAL DISTINCTIONS BETWEEN GRAPH METRICS AND LATENT MANIFOLDS

To avoid ambiguity regarding the use of manifold theory in complex systems, we clarify the distinction between topological descriptors and latent geometric spaces. In this work, we define the latent manifold as the hidden, lower-dimensional structure that captures the essential configuration of the system.

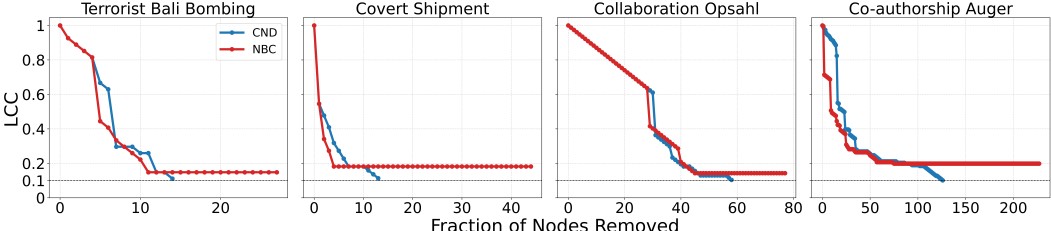

Figure 6: Dynamic dismantling process on example networks comparing local network automata rule RA2 and its variants versus NBC, when the former outperforms NBC in terms of AUC. The plot shows the normalized size of the largest connected component (LCC) as a function of the fraction of nodes removed, with a target LCC threshold of 10%. The final evaluation metric is the Area Under the Curve (AUC) of the LCC trajectory.

To infer the latent manifold from high-dimensional data, a range of general dimensionality reduction and manifold learning techniques can be applied. These approaches seek to map the data points into a continuous, lower-dimensional space where geometric proximity reflects similarity in the original space. They can be broadly categorized as methods preserving local structure (e.g., t-SNE, UMAP, and Minimum Curvilinear Embedding (MCE)), methods based on calculating intrinsic distances (e.g., Isomap (ISO) and its variants), spectral methods (e.g., Laplacian eigenmaps and Diffusion Maps), and deep learning techniques (e.g., Autoencoders and VAEs) that learn the latent code necessary for data reconstruction. Finally, specialized manifold learning approaches in dynamical systems (e.g., Koopman operator theory) can transform complex, nonlinear dynamics into simpler, linear representations within a manifold.

When data is organized as a complex network, the latent manifold is typically inferred using network embedding techniques specifically designed to preserve the network's topology. These methods fall into three broad categories: spectral methods (e.g., spectral clustering) which use the algebraic properties of the graph matrices; deep learning approaches (e.g., DeepWalk, Node2Vec, and Graph Autoencoders (GAE)) which learn representations using neural networks trained on structural information like random walks, while Graph Neural Networks (GNNs) have emerged as the state-of-the-art for learning task-specific embeddings using topology and node/edge features; and geometric approaches such as Hyperbolic network embeddings. These geometric methods (e.g., Poincaré embeddings, Hypermap) utilize non-Euclidean geometries, such as negative curvature, to efficiently capture the hierarchical and scale-free properties of complex networks. Specific algorithms like LPCS generate node coordinates by analyzing and ordering the network's community structure.

We distinguish this latent manifold from graph metrics. For example, standard topological graph descriptors such as small-worldness, community structure, and degree heterogeneity are not direct descriptors of the manifold themselves. However, since the network is sampled from a specific latent space, these observed properties are influenced by the manifold's geometry. Consequently, these topological metrics do characterize the topology of a network that is embedded in a specific latent space: for instance, small-worldness suggests short geodesic distances, community structure can imply stratification or clustering, and degree heterogeneity may reflect features such as local curvature or singularities. Our approach uses the topology of the observable network to infer the geometric distance between nodes within the network's latent manifold, thereby allowing us to exploit the manifold's geometric properties for dismantling.

Note that we include a GNN-based dismantling algorithm, Graph Dismantling with Machine learning (GDM) (Grassia et al., 2021), in our experiments. GDM is a GNN that is trained on optimally dismantled networks, and is considered a state-of-the-art dismantling algorithm (Artime et al., 2024; Grassia et al., 2021) that can implicitly capture features of the underlying latent geometry of the target network. The fact that our LGD-NA methods consistently outperform GDM in all situations suggests that our estimators might be yielding a more accurate estimation of the target network's latent geometry."

## C  GEOMETRIC VALIDATION OF LATENT GEOMETRY ESTIMATORS

To provide visual and empirical validation for our latent-geometry estimators, we analyze the ability of our latent-geometry estimators to identify node importance and estimate link distances using synthetically generated networks with a known geometry. As previously mentioned, the RA measures were introduced to serve as pre-weighting strategies for approximating angular distances associated with node similarities in hyperbolic network embeddings (Muscoloni et al., 2017).

To investigate this, we synthetically generate networks using the non-uniform Popularity-Similarity Optimization (nPSO) model (Muscoloni & Cannistraci, 2018b). The nPSO model is built on the principle that radial coordinates represent hierarchy (popularity) while angular coordinates represent similarity. It produces networks that are both scale-free (characterized by a power-law degree distribution, meaning a network has a few highly connected hubs while the majority of nodes have few links) and clustered with distinct communities, closely mimicking the structure of many real-world complex systems. We utilize the nPSO network model specifically for this task because these networks are generated with known node coordinates and a known underlying hyperbolic geometry, making them highly suitable for validating geometry-related measures in network science.

We generate various nPSO networks keeping the number of nodes ($N = 500$) and communities ($C = 5$) fixed. We test different network topologies by varying:

- The power-law exponent $\gamma \in \{2, 3\}$ represents common bounds for real-world scale-free networks. With $\gamma = 3$, fewer high-degree hubs exist, creating less hierarchy (seen through the radial coordinates) and reduced network hyperbolicity (meaning that they become more similar to a Erdos-Renyi random graph compared to when $\gamma = 2$).

- The number of nodes a new node will connect to when being added to the network, $m \in \{10, 20, 50\}$. This value represents approximately half of the average node degree, making the network more or less connected. This results in networks with three different density levels $\rho \in \{0.04, 0.08, 0.2\}$.

- The temperature $T \in \{0.3, 0.6, 0.9\}$ controls clustering, where lower temperatures produce stronger clustering. Higher temperatures reduce clustering (seen through the angular coordinates) and increase the randomness of connectivity, thus reducing the generated network's hyperbolicity (nodes connect more by random rather than following the underlying hyperbolic geometry).

Figure 7 visualizes synthetic nPSO networks with nodes colored by CND score (red: high, blue: low) and sized by degree. The visualization clearly shows that high CND scores correspond to nodes with highcentrality, hubs located near the center of the hyperbolic disk. This relationship is most evident for $\gamma = 2$, where the skewed degree distribution creates a clear distinction between central hubs and peripheral nodes. For $\gamma = 3$, the trend persists but is less pronounced due to fewer super-hubs, consistent with the network's reduced hyperbolicity. These results provide strong visual evidence that CND effectively identifies structurally important nodes in the hyperbolic latent space.

To quantitatively support our claim, we evaluate how well the latent geometry estimators approximate the true hyperbolic distances. We use the hyperbolic distance correlation (HD-correlation) metric, the Pearson correlation between all pairwise geometrical shortest path distances in the networks' original hyperbolic space and the weighted shortest path distances using the latent-geometry estimators as edge weights (Muscoloni et al., 2017). The higher this correlation, the better the latent-geometry estimator is able to recover the geometrical distances between pairs of nodes in a network's underlying geometry.

Figure 4 shows a high HD-correlation for both CND and RA2 across all tested nPSO configurations, confirming that these measures used in our dismantling framework are effective latent geometry estimators. This is further supported by the statistical significance reported in Table 20.

The Pearson correlation is visualized in Figure 8 for different parameters, visualizing how well the distance approximation changes as the network becomes less hyperbolic. As expected, for $\gamma = 2$, the correlation decreases for both estimators with increasing temperature (i.e., reduced clustering and hyperbolicity). For the less hyperbolic $\gamma = 3$ networks, this decreasing trend persists for CND but not for RA2. This suggests that CND remains a robust estimator of the latent geometry even when

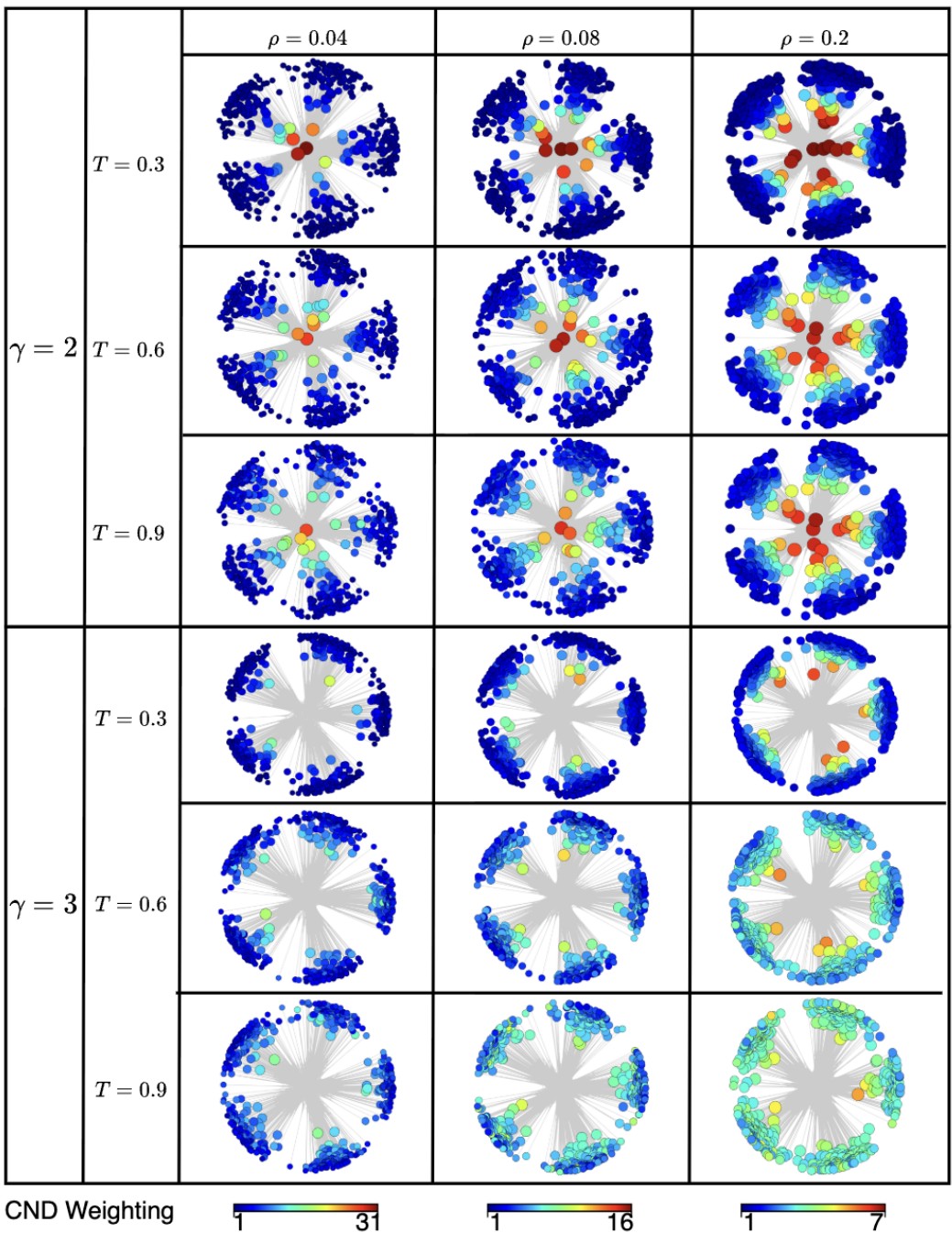

Figure 7: nPSO model networks visualized in the hyperbolic space. Fixed parameters are the number of nodes, N=500, and the number of communities, C=5. Nodes are colored according to their CND measure, where red represents higher CND scores and blue lower ones. Ranges of CND values are reported in the color bar and are different for each density level. Node sizes are positively correlated with their degree.

hyperbolic structure is less pronounced, whereas RA2's performance is more dependent on strongly hyperbolic conditions, consistent with our dismantling experiments.

We also conducted experiments considering only existing links, correlating their estimated weights with the true geometrical shortest path distances in the hyperbolic space (Figures 21 and 22). The results confirm that both CND and RA2 are effective latent geometry estimators, as the link weights strongly correlate with the true distances.

This visual and quantitative evidence demonstrates our LGD-NA measures' ability to accurately estimate the geometric distance between nodes. Consequently, the node aggregation step in our LGD-NA framework can successfully identify nodes that connect distant regions in the latent space.

Table 4: Pearson correlation between all the pairwise geometrical shortest path distances of the network nodes in the original nPSO model and in the reconstructed hyperbolic space (HD-correlation) (Muscoloni et al., 2017). Mean values over 10 seeds are reported, with a color gradient where green corresponds to values approaching 1 and red to values approaching -1. The power-law exponent $\gamma$ represents the scale-freeness found in real-world networks. networks. $\rho$ is the density of the networks. The temperature $T$ controls the level of clustering (lower temperatures yield stronger clustering). Fixed parameters are the number of nodes, $N = 500$, and the number of communities, $C = 5$. Standard Error of the Mean (SEM) and Fisher p-value are found in Table 20.

| N=500, C=5 | | | $\rho$=0.04 | $\rho$=0.08 | $\rho$=0.2 |
|---|---|---|---|---|---|
| $\gamma$=2 | CND | T=0.3 | 0.722 | 0.792 | 0.846 |
| | | T=0.6 | 0.693 | 0.768 | 0.801 |
| | | T=0.9 | 0.633 | 0.765 | 0.777 |
| | RA2 | T=0.3 | 0.521 | 0.532 | 0.521 |
| | | T=0.6 | 0.524 | 0.484 | 0.308 |
| | | T=0.9 | 0.498 | 0.460 | 0.303 |
| $\gamma$=3 | CND | T=0.3 | 0.584 | 0.624 | 0.645 |
| | | T=0.6 | 0.510 | 0.579 | 0.590 |
| | | T=0.9 | 0.452 | 0.552 | 0.597 |
| | RA2 | T=0.3 | 0.685 | 0.714 | 0.783 |
| | | T=0.6 | 0.722 | 0.780 | 0.805 |
| | | T=0.9 | 0.688 | 0.795 | 0.755 |

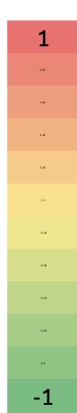

## D  TOPOLOGICAL CENTRALITY MEASURES

Table 5: Comparison of topological centrality measures and the associated time complexity for dynamic dismantling using each centrality measure. *Information Locality* denotes the type of structural information required to assign a score to each node. *Time Complexity* denotes the time complexity for dynamic dismantling using each centrality measure on sparse graphs, without reinsertion. $N$: number of nodes. $m$: number of links.

| Measure | Author | Year | Type | Information Locality | Time Complexity |
|---|---|---|---|---|---|
| Degree | | | Degree-based | Local | $\mathcal{O}(N \log N)$ |
| Eigenvector | Bonacich (1972) | 1972 | Walks-based | Global | $\mathcal{O}(N(N + m))$ |
| Node Betweenness (NBC) | Freeman (1977) | 1977 | Shortest path-based | Global | $\mathcal{O}(N(Nm))$ |
| PageRank (PR) | Page et al. (1999) | 1999 | Random walk-based | Global | $\mathcal{O}(N(N + m))$ |
| Resilience | Zhang et al. (2020) | 2020 | Resilience-based | Global | $\mathcal{O}(N(N + m)$ |
| Domirank | Engsig et al. (2024) | 2024 | Fitness-based | Global | $\mathcal{O}(N(N + m)$ |
| Fitness | Servedio et al. (2025) | 2025 | Fitness-based | Global | $\mathcal{O}(N(N + m)$ |

**Time Complexity.** We analyze the time complexity of dynamic dismantling (excluding reinsertion) for the topological centrality measures used in our experiments, summarized in Table 5. As before, the analysis assumes $N$ dismantling steps in the worst-case scenario. For degree, the score update after each removal is local and can be done in $\mathcal{O}(\log N)$ time using a binary heap. For NBC, we use Brandes' algorithm (Brandes, 2001), which computes betweenness centrality in $\mathcal{O}(Nm)$ time per

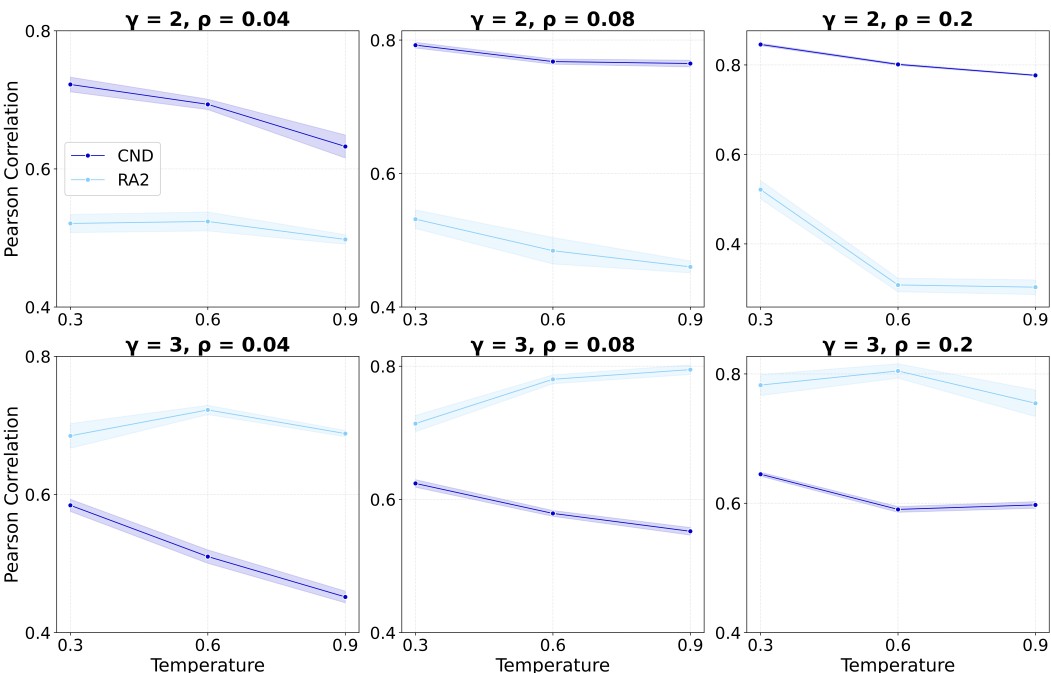

Figure 8: Pearson correlation between all the pairwise geometrical shortest path distances of the network nodes in the original nPSO model and in the reconstructed hyperbolic space (HD-correlation) (Muscoloni et al., 2017). Mean values over 10 seeds are reported, with the shaded area the Standard Error of the Mean (SEM). The power-law exponent $\gamma$ represents the scale-freeness found in real-world networks. networks. $\rho$ is the density of the networks.The temperature $T$ controls the level of clustering (lower temperatures yield stronger clustering). Fixed parameters are the number of nodes, $N = 500$, and the number of communities, $C = 5$.

step for unweighted networks. Eigenvector, PageRank, Resilience, Domirank, and Fitness all rely on matrix-vector multiplications, which has a time complexity of $\mathcal{O}(m)$. We also add the term $N$, which represents the overhead of looping over nodes to update or normalize the resulting vector at each iteration. This leads to a total per-step cost of $\mathcal{O}(N+m)$. We also omit the constant $k$ for Eigenvector, PageRank, and Fitness centrality, which represents the number of iterations these methods perform. In practice, reaching full convergence to a single optimal solution is often computationally infeasible; this is why a fixed number of $k$ iterations is typically defined.

## E  STATISTICAL AND MACHINE LEARNING NETWORK DISMANTLING.

Table 6: Comparison of dismantling algorithms (Artime et al., 2024). *Information Locality* denotes the type of structural information required to assign a score to each node. *Dynamicity* indicates whether scores are recomputed after each removal. *Reinsertion* specifies whether the algorithm includes a reinsertion step after dismantling. *Time Complexity* denotes the time complexity of the method on sparse graphs, without reinsertion. $N$: number of nodes. $m$: number of links. $h$: number of attention heads. $T$: maximal diameter of the trees in the forest for BPD and MS. $\epsilon$ is a small constant used in spectral partitioning operations. *Included* states whether the method was run in our experiments; if not, a brief reason is provided.

| Algorithm | Type | Author | Year | Information Locality | Dynamicity | Reinsertion | Time Complexity | Included |
|---|---|---|---|---|---|---|---|---|
| Collective Influence (CI) | Influence maximization | Morone et al. (2016) | 2016 | Local | Dynamic | Yes | $\mathcal{O}(N \log N)$ | Yes |
| Belief propagation-guided decimation (BPD) | Message passing-based decycling | Mugisha & Zhou (2016) | 2016 | Global | Dynamic | Optional | $\mathcal{O}(mT)$ | No - Code missing |
| Min-Sum (MS) | Message passing-based decycling | Braunstein et al. (2016) | 2016 | Global | Dynamic | Yes | $\mathcal{O}(mT) + \mathcal{O}(N(\log N + T))$ | Yes |
| Generalized Network Dismantling (GND) | Spectral partitioning | Ren et al. (2019) | 2019 | Global | Dynamic | Optional | $\mathcal{O}(N \log^{2+\varepsilon} N)$ | Yes |
| CoreHD | Degree-based decycling | Zdeborová et al. (2016) | 2016 | Global | Dynamic | Yes | $\mathcal{O}(N)$ | Yes |
| Explosive Immunization (EI) | Explosive percolation | Clusella et al. (2016) | 2016 | Global | Dynamic | No | $\mathcal{O}(N \log N)$ | Yes |
| FINDER | Machine learning | Fan et al. (2020b) | 2020 | Global | Dynamic | Optional | $\mathcal{O}(N(1 + \log N) + m)$ | No - Code outdated |
| Graph Dismantling Machine (GDM) | Machine learning | Grassia et al. (2021) | 2021 | Global | Static | Optional | $\mathcal{O}(h(N + m))$ | Yes |
| CoreGDM | Machine learning | Grassia & Mangioni (2023) | 2023 | Global | Static | Yes | $\mathcal{O}(h(N + m))$ | Yes |

Table 6 is adapted and extended from Table 1 of Artime et al. (Artime et al., 2024), a recent and comprehensive review which has become a key reference in the field of network dismantling. The majority of these algorithms were included in our experiments, with the exception of BPD and FINDER due to unavailable or outdated code, respectively.

## F  REINSERTION METHODS

Reinsertion was originally introduced in the context of immunization as a reverse process: starting from a fully dismantled network, nodes are reinserted one by one, each time selecting the node whose addition causes the smallest increase in the largest connected component (LCC) (Schneider et al., 2012). This reversed sequence then defines an effective dismantling order. In subsequent studies, reinsertion has been used as a post-processing step to improve dismantling outcomes (Artime et al., 2024): the network is first dismantled by a given method, and nodes are reinserted until the LCC reaches the dismantling threshold. This reduces the dismantling cost while preserving the original attack target.

In this work, dismantling cost is defined as the number of nodes removed from the network. The reinsertion step aims to directly minimize this cost by reintroducing nodes that were initially removed but found to be unnecessary for achieving the dismantling objective. It's important to note that while reinsertion reduces the number of physical removals, it does introduce a higher computational

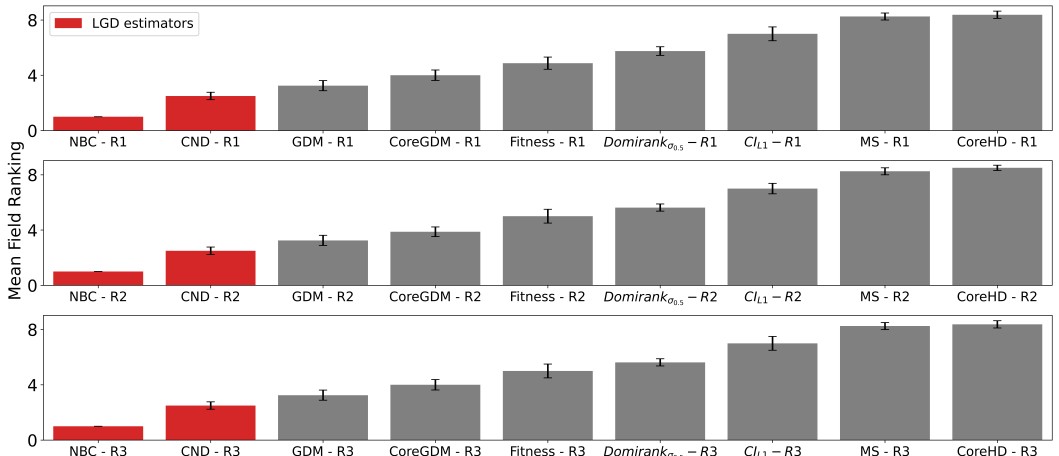

Figure 9: Mean field ranking for a subset of the best-performing methods from each category with each reinsertion method (R1, R2, R3) ($n = 1{,}237$). Methods based on latent geometry are shown in red. All LGD and topological centrality measures use dynamic dismantling. Error bars indicate the standard error of the mean (SEM). Method acronyms are defined in Tables 3, 5, and 6.

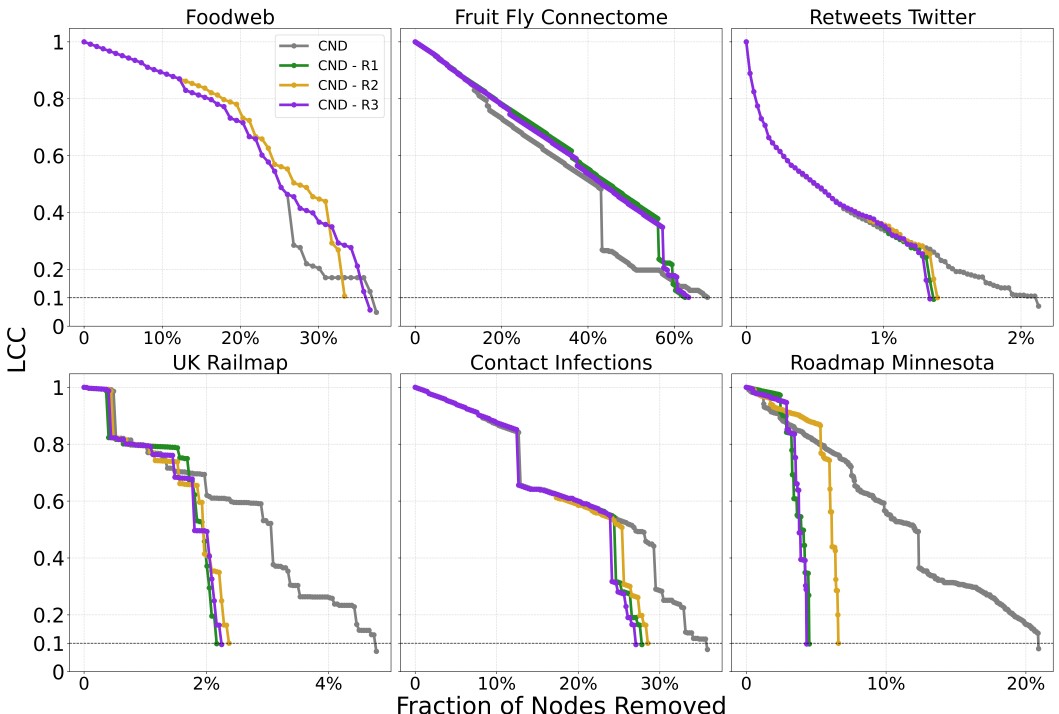

Figure 10: Dynamic dismantling process on example networks comparing CND with and without reinsertion. The plot shows the normalized size of the largest connected component (LCC) as a function of the fraction of nodes removed, with a target LCC threshold of 10%. The final evaluation metric is the Area Under the Curve (AUC) of the LCC trajectory.

cost as it's a post-processing step performed after the initial dismantling. However, the primary objective is to minimize this physical intervention, as in many real-world scenarios, the logistical and financial implications of physically removing network components (e.g., infrastructure) far outweigh the computational resources expended during the optimization phase. This is why we compare all methods with and without the reinsertion step.

Several reinsertion criteria have been proposed: Braunstein et al. (2016) select the node that ends up in the smallest resulting component after reinsertion; Morone et al. (2016) choose the node that reconnects the fewest components; Mugisha & Zhou (2016) select the node that causes the smallest LCC increase. See Table 7 for a full comparison.

Reinsertion can greatly enhance dismantling performance. However, recent work shows that this step can overpower the dismantling algorithm itself, allowing weak methods to appear effective when paired with reinsertion (Fan et al., 2020a). To address this, we enforce two constraints to ensure fair comparisons and prevent reinsertion from dominating the dismantling process:

1. Reinsertion must stop once the LCC exceeds the dismantling threshold. Recomputing a new dismantling order by reinserting all nodes is not allowed.

2. Ties in the reinsertion criterion must be broken by reversing the dismantling order: nodes removed later are prioritized.

These rules ensure that reinsertion complements rather than overrides the dismantling process, preserving the integrity of the original method.

In our experiments, we implement three reinsertion methods, adapted from prior work, here we explain which part of their method we change for our experiments. Those changes are marked with an asterisk (*) in Table 7.:

- **R1** (Braunstein et al., 2016): We replace their original tiebreak (smallest node index) with reverse dismantling order.

- **R2** (Morone et al., 2016): We apply the LCC stopping condition. Originally, all nodes are reinserted to compute a new dismantling sequence.

- **R3** (Mugisha & Zhou, 2016): We apply reverse dismantling order as the tiebreak, as no rule is defined in their paper, and their code is unavailable.

R3 is the most similar to the reverse immunization method proposed by Schneider et al. (2012), where nodes are added back one by one based on minimal LCC growth. In their original method, ties are broken by selecting the node with the fewest connections to already reinserted nodes; if multiple candidates remain, one is chosen at random.

We note that reinsertion typically reduces the number of removals but does not always lead to a lower AUC. Since the trajectory of the LCC changes with reinsertion, the dismantling process may reach the threshold faster, improving AUC. However, this is not guaranteed, as we see in the first two subplots of Figure 10 for the Foodweb and Fruit Fly Connectome networks. The methods with reinsertion arrive at the dismantling threshold in fewer number of removals, but the change in the LCC curve results in a worse final AUC.

We also see that the reduction in AUC is not proportional to the reduction in the number of removals, as seen in Figures 11 and 12 for CND. Indeed, reinsertion, by definition, reinserts nodes that were ultimately unnecessary for the dismantling process to reach its target.

A significant limitation in previous literature is the lack of differentiation between algorithms that inherently include reinsertion and those that do not, leading to inconsistent comparisons. To ensure a strictly fair evaluation, we standardized two critical control variables across all experiments: the tie-breaking mechanism for the order of reinsertion and the stopping criteria. Furthermore, rather than arbitrarily assigning a reinsertion strategy, we evaluated every method under the three reinsertion methods. We report the best performance for each method, ensuring that the results reflect the maximum potential of the dismantling strategy rather than an inconsistent application of reinsertion.

**Ranking Stability.** Across all tested reinsertion methods, the mean-field ranking remains the same: NBC consistently outranks CND, which in turn outranks GDM. This order holds true both when

comparing specific fixed reinsertion methods and when selecting the best-performing method for each dismantling method. However, we observe a nuanced interaction between the dismantling algorithms and their best reinsertion strategy: the optimal reinsertion method varies (R2 is optimal for NBC, R3 for CND, and R1 for GDM). For all other algorithms, though, R1 is the most effective reinsertion strategy."

**Time Complexity.** We report the total time complexity of each reinsertion method over the full reinsertion process in Table 7, assuming all dismantled nodes are considered for reinsertion for every step and that all nodes are reinserted. Candidates for reinsertion are denoted as $r$. As a result, we multiply the per-step cost of updating for each method by the total number of reinsertion candidates. $k_{\max}$ is the maximum number of components a node can connect to, equal to the maximum degree in the original graph, and $C'$ is the maximum size of any connected component during the reinsertion phase. For R1, the candidate node that ends up in the smallest resulting component is selected. Reinserting a node may merge up to $k_{\max}$ components, each of size at most $C'$, requiring an update of at most $k_{\max} \cdot C'$ nodes. These updates are tracked in a binary heap of size $r$, where at maximum $k_{\max} \cdot C'$ nodes have to be updated, giving a cost of $\log(k_{\max} \cdot C')$ per update. The per-step cost is therefore $\mathcal{O}(k_{\max} \cdot C' \log(k_{\max} \cdot C'))$. R2 selects the node that connects the fewest existing components. Unlike R1, it requires inspecting not only the components merged by the candidate node, but also the neighbors of the affected neighbors. This increases the complexity by a factor of $k_{\max}$, resulting in a per-step time complexity of $\mathcal{O}(k_{\max}^2 \cdot C' \log(k_{\max}^2 \cdot C'))$. R3 evaluates each candidate by explicitly computing the resulting LCC size after reinsertion. Each evaluation requires a graph traversal to recompute connected components, which takes $\mathcal{O}(N + m)$ time on sparse graphs. This has to be done for each reinsertion candidate, at every step, so $\mathcal{O}(r^2(N + m))$,

Table 7: Comparison of reinsertion methods. *Criteria* defines the criterion for selecting which node to reinsert. *Tiebreak* specifies how ties are resolved. *LCC Condition* indicates whether all dismantled nodes are reinserted or if reinsertion stops once the predefined LCC threshold is reached. *Time Complexity* denotes the time complexity of each reinsertion method on sparse graphs, for the whole reinsertion process. *N*: number of nodes. *m*: number of links. *r*: set of reinsertion candidates. $k_{\max}$: maximum degree in the original graph $G$. $C'$: maximum size of any connected component during the reinsertion phase. *Used In* lists the methods that use each method, in bold, the dismantling method that originally proposed that reinsertion method. An asterisk (*) marks components of the reinsertion method that were modified in our study, as detailed in Appendix F.

| Name | Author | Year | Criteria | Tiebreak | LCC Condition | Time Complexity | Used In |
|------|--------|------|----------|----------|---------------|-----------------|---------|
| R1 | Braunstein et al. (2016) | 2016 | Node that ends up in the smallest component | Reverse dismantling order* | Yes | $\mathcal{O}(r(k_{\max} \cdot C' \cdot \log(k_{\max} \cdot C')))$ | **MS**, CoreGDM, CoreHD, GDM, GND |
| R2 | Morone et al. (2016) | 2016 | Node that connects to the fewest clusters | Reverse dismantling order | Yes* | $\mathcal{O}(k_{\max}^2 \cdot C' \cdot \log(k_{\max}^2 \cdot C'))$ | **CI** |
| R3 | Mugisha & Zhou (2016) | 2016 | Node that causes the smallest increase in LCC size | Reverse dismantling order* | Yes | $\mathcal{O}(r^2(N + m))$ | **BPD** |

# G DYNAMIC & STATIC DISMANTLING

In static dismantling, node scores are computed once at the beginning and are then removed in descending order of importance until the dismantling threshold is reached. In contrast, dynamic dismantling recomputes the scores after each removal. As shown in Figure 13, with CND given as an example, dynamic dismantling consistently outperforms static dismantling across all fields. Dynamic variants achieve lower AUC and fewer removals in every case, confirming the advantage of score recomputation.

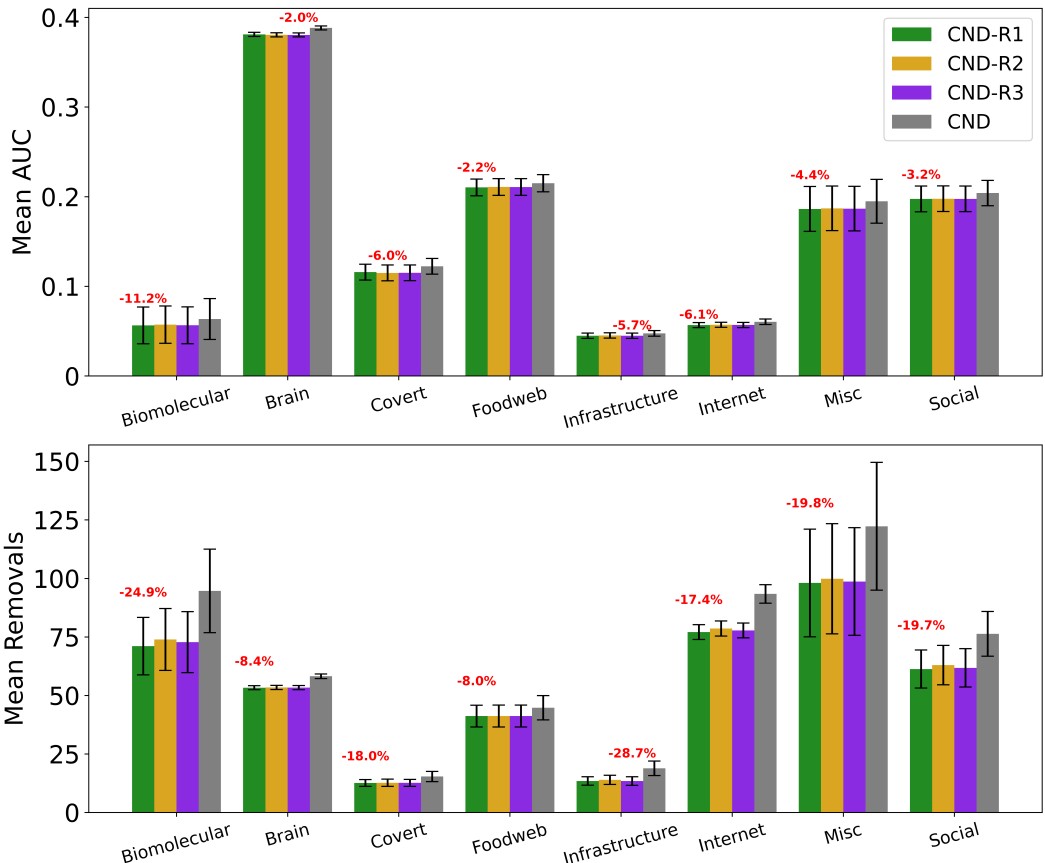

Figure 11: Mean AUC and number of removals by field for CND without reinsertion and with each reinsertion method (R1, R2, R3) ($n = 1{,}237$ for all methods). Error bars represent the standard error of the mean (SEM). Red text indicates the percentage improvement achieved by using the best-performing reinsertion method for each field. Quantitative results for the AUC and removals improvement from each reinsertion methods are reported in Table 8.

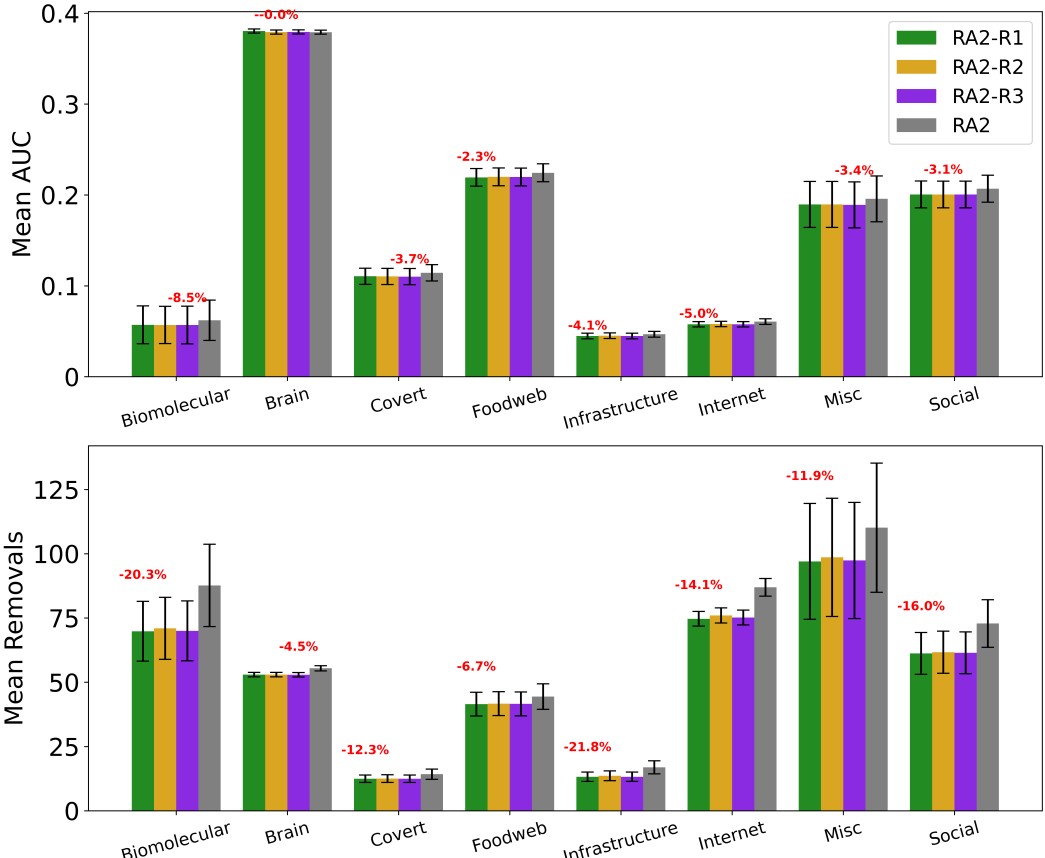

Figure 12: Mean AUC and number of removals by field for RA2 without reinsertion and with each reinsertion method (R1, R2, R3) ($n = 1{,}237$ for all methods). Error bars represent the standard error of the mean (SEM). Red text indicates the percentage improvement achieved by using the best-performing reinsertion method for each field. Quantitative results for the AUC and removals improvement from each reinsertion methods are reported in Table 8.

Table 8: Percentage improvement for the mean AUC and mean number of removals for each reinsertion method over the baseline for CND and RA2 ($n = 1{,}237$). In bold the method that improves the baseline the most, by field.

| AUC | CND | | | | AUC | RA2 | | |
|---|---|---|---|---|---|---|---|---|
| | R1 | R2 | R3 | | | R1 | R2 | R3 |
| Biological | **11.2%** | 9.9% | 11.0% | | Biological | 8.0% | 8.3% | **8.5%** |
| Connectome | 1.9% | 2.0% | **2.0%** | | Connectome | -0.3% | **0.0%** | -0.1% |
| Covert | 5.3% | **6.0%** | 6.0% | | Covert | 3.4% | 3.5% | **3.8%** |
| Foodweb | **2.2%** | 2.0% | 2.0% | | Foodweb | **2.3%** | 2.0% | 2.1% |
| Infrastructure | 5.6% | 5.0% | **5.7%** | | Infrastructure | **4.1%** | 3.4% | 4.0% |
| Internet | **6.1%** | 5.5% | 5.8% | | Internet | **5.0%** | 4.3% | 4.9% |
| Misc | **4.4%** | 4.0% | 4.2% | | Misc | 3.2% | 3.1% | **3.4%** |
| Social | **3.2%** | 3.1% | 3.2% | | Social | 3.1% | **3.1%** | 3.1% |

| Removals | CND | | | | Removals | RA2 | | |
|---|---|---|---|---|---|---|---|---|
| | R1 | R2 | R3 | | | R1 | R2 | R3 |
| Biological | **24.9%** | 21.9% | 23.1% | | Biological | **20.3%** | 19.1% | 20.1% |
| Connectome | **8.4%** | 8.2% | 8.3% | | Connectome | 4.4% | 4.4% | **4.5%** |
| Covert | **18.0%** | 17.4% | 17.7% | | Covert | **12.3%** | 12.1% | **12.3%** |
| Foodweb | 8.0% | 7.9% | **8.0%** | | Foodweb | **6.7%** | 6.2% | 6.4% |
| Infrastructure | 28.7% | 26.2% | **28.7%** | | Infrastructure | **21.9%** | 19.4% | 21.8% |
| Internet | **17.5%** | 15.8% | 16.7% | | Internet | **14.1%** | 12.6% | 13.5% |
| Misc | **19.8%** | 18.3% | 19.3% | | Misc | **11.9%** | 10.5% | 11.6% |
| Social | **19.7%** | 17.5% | 19.1% | | Social | **16.0%** | 15.3% | 15.7% |

Table 9: Full summary statistics of the ATLAS networks used in this study, averaged by field: number of subfields and networks, average number of nodes $\langle N \rangle$, number of edges $\langle E \rangle$, density $\langle \rho \rangle$, mean degree $\langle\langle d \rangle\rangle$, characteristic path length $\langle \ell \rangle$, assortativity $\langle r \rangle$ (Newman, 2002), transitivity $\langle T \rangle$, mean local clustering coefficient $\langle\langle \text{Loc. CC} \rangle\rangle$, maximum $k$-core $\langle k_{\max} \rangle$, average $k$-core $\langle\langle k \rangle\rangle$, LCP-corr $\langle LCP_{corr} \rangle$ (Cannistraci et al., 2013), and modularity $\langle Q \rangle$ (Newman, 2004)

| Field | Biomolecular | Brain | Covert | Foodweb | Infrastructure | Internet | Misc | Social | **Total** |
|---|---|---|---|---|---|---|---|---|---|
| Subfields | 5 | 1 | 2 | 1 | 7 | 1 | 8 | 7 | **32** |
| Networks | 27 | 529 | 89 | 71 | 314 | 206 | 38 | 201 | **1,475** |
| $\langle N \rangle$ | 2,997 | 97 | 107 | 117 | 664 | 5,708 | 2,880 | 3,267 | |
| $\langle E \rangle$ | 11,855 | 1,535 | 266 | 1,087 | 1,332 | 19,601 | 19,921 | 53,977 | |
| $\langle \rho \rangle$ | 0.01 | 0.34 | 0.17 | 0.16 | 0.07 | 0.01 | 0.07 | 0.11 | |
| $\langle\langle d \rangle\rangle$ | 6.7 | 28.3 | 5.7 | 15.2 | 4.9 | 7.5 | 14.1 | 26.9 | |
| $\langle \ell \rangle$ | 4.4 | 1.7 | 3 | 2.2 | 9.9 | 3.4 | 3.5 | 3.5 | |
| $\langle r \rangle$ | -0.21 | -0.03 | -0.15 | -0.28 | -0.52 | -0.22 | -0.07 | -0.05 | |
| $\langle T \rangle$ | 0.06 | 0.55 | 0.39 | 0.19 | 0.06 | 0.11 | 0.22 | 0.29 | |
| $\langle\langle Loc.CC \rangle\rangle$ | 0.13 | 0.63 | 0.46 | 0.22 | 0.11 | 0.31 | 0.34 | 0.36 | |
| $\langle k_{\max} \rangle$ | 10.6 | 20.1 | 5.9 | 12.8 | 4.9 | 25 | 21.6 | 25.7 | |
| $\langle\langle k \rangle\rangle$ | 3.6 | 17.5 | 4.2 | 9.2 | 3 | 4 | 8.3 | 15.4 | |
| $\langle LCP_{corr} \rangle$ | 0.66 | 0.97 | 0.76 | 0.67 | 0.15 | 0.94 | 0.85 | 0.77 | |
| $\langle Q \rangle$ | 0.59 | 0.25 | 0.48 | 0.26 | 0.46 | 0.5 | 0.49 | 0.5 | |

Table 10: Number and size of real-world networks tested by dismantling algorithms. $N$ denotes the number of nodes, $E$ the number of edges.

| Algorithm | Year | Networks | $N_{max}$ | $E_{max}$ |
|---|---|---|---|---|
| Collective Influence (CI) (Morone et al., 2016) | 2016 | 2 | 14M | 51M |
| CoreHD (Zdeborová et al., 2016) | 2016 | 12 | 1.7M | 11M |
| Explosive Immunization (EI) (Clusella et al., 2016) | 2016 | 5 | 50K | 344K |
| Min-Sum (MS) (Braunstein et al., 2016) | 2016 | 2 | 1.1M | 2.9M |
| Generalized Network Dismantling (GND) (Ren et al., 2019) | 2019 | 10 | 5K | 17K |
| Resilience Centrality (Zhang et al., 2020) | 2020 | 4 | 1K | 14K |
| Graph Dismantling Machine (GDM) (Grassia et al., 2021) | 2021 | 57 | 1.4M | 2.8M |
| CoreGDM (Grassia & Mangioni, 2023) | 2023 | 15 | 79K | 468K |
| Domirank Centrality (Engsig et al., 2024) | 2024 | 6 | 24M | 58M |
| Fitness Centrality (Servedio et al., 2025) | 2025 | 5 | 297 | 4K |
| **LGD-NA** | **2025** | **1,475** | **23K** | **507K** |

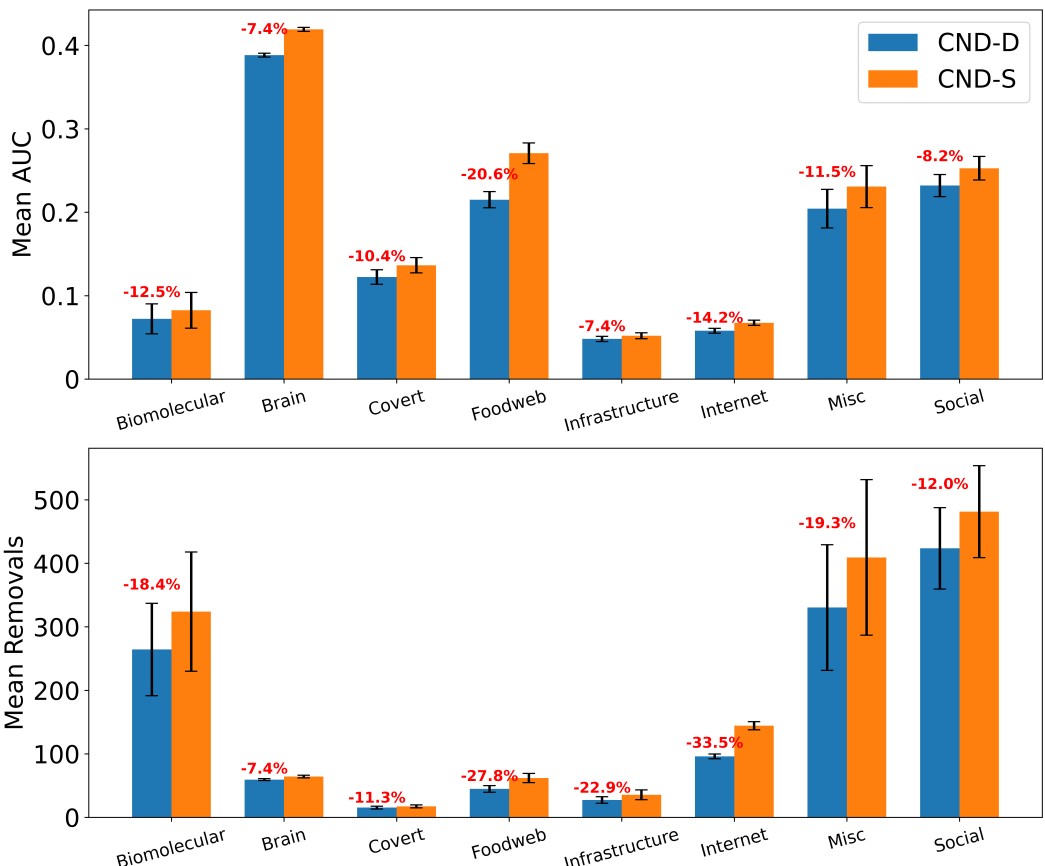

Figure 13: Mean AUC and number of removals for dynamic and static CND ($n = 1{,}296$. Error bars represent the standard error of the mean (SEM). Red text indicates the percentage improvement achieved by using dynamic over static variants.

Table 11: Percentage improvement for the mean AUC and mean number of removals for dynamic CND over static CND ($n = 1,296$), by field.

|  | AUC | Removals |
|---|---|---|
| **Biomolecular** | 12.53% | 18.42% |
| **Brain** | 7.37% | 7.43% |
| **Covert** | 10.35% | 11.30% |
| **Foodweb** | 20.57% | 27.81% |
| **Infrastructure** | 7.36% | 22.95% |
| **Internet** | 14.20% | 33.47% |
| **Misc** | 11.47% | 19.27% |
| **Social** | 8.22% | 11.99% |

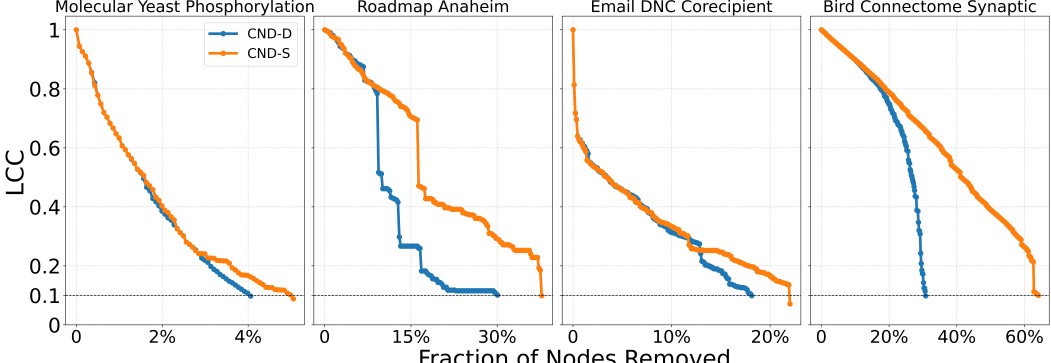

Figure 14: Dismantling process on example networks comparing dynamic and static CND. The plot shows the normalized size of the largest connected component (LCC) as a function of the fraction of nodes removed, with a target LCC threshold of 10%. Performance is evaluated using the Area Under the Curve (AUC) of the LCC trajectory.

# H    ENGINEERING NETWORK ROBUSTNESS

Here, we would like to comment on the feasibility of our suggested network modifications in practical scenarios. In the case of PPI networks, recent advances in structural modeling of molecules using AlphaFold 3 (AF3) (Abramson et al., 2024) have reduced the longstanding limitations in testing and engineering arbitrary proteins. Coupling our dismantling predictions with AF3 could impact drug repositioning and drug design. For example, in the case of antibiotic-resistant bacteria, knowledge of how to dismantle the bacterial PPI network could be used to identify which proteins to target with repositioned, modified, or newly designed antibiotics. Meanwhile, to minimize side effects of newly designed drugs that target critical proteins in bacterial PPI networks, our reinforcement strategy based on common-neighbor generation could be applied to predict which protective bindings to promote in the human PPI network, thereby reducing the destructive impact of a drug on a critical human protein whose impairment could cause side effects. Finally, the application of the proposed common-neighbor reinforcement strategy to increase the robustness of flight and shipping networks is straightforward, as it can indicate which critical nodes should be reinforced by adding links between their adjacent nodes.

We validate our reinforcement strategy explained in Section 4.6 on three types of real-world networks considered critical: a human PPI network (biological), a flight map network (transportation with social and geographic constraints), and a shipping trade network (transportation with economic and geographic constraints). We select the top 1% of highest-scoring nodes according to the chosen measure (NBC or CND) and randomly add either 1% or 10% of the potential links between their respective adjacent nodes, thereby modifying the network topology according to the explainable rule discovered via CND.

Table 12: Area Under the dismantling Curve (AUC) for NBC and CND on the original network and "reinforced" networks, by adding 1% and 10% of the links between common neighbors of the top 1% of nodes. In parentheses the increase in the AUC compared to the original network, representing the reduction in the dismantling effectiveness.

| Human PPI | 0% Links | 1% Links | 10% Links |
|---|---|---|---|
| NBC-baseline | 0.051 | | |
| NBC-reinforced | | 0.093 (+84%) | 0.159 (+214%) |
| CND-baseline | 0.055 | | |
| CND-reinforced | | 0.098 ((+79%)) | 0.198 ((+259%)) |

| Flight Map US | 0% Links | 1% Links | 10% Links |
|---|---|---|---|
| NBC-baseline | 0.042 | | |
| NBC-reinforced | | 0.067 (+61%) | 0.122 (+193%) |
| CND-baseline | 0.055 | | |
| CND-reinforced | | 0.098 ((+95%)) | 0.172 ((+241%)) |

| Trade Shipping | 0% Links | 1% Links | 10% Links |
|---|---|---|---|
| NBC-baseline | 0.138 | | |
| NBC-reinforced | | 0.236 (+71%) | 0.240 (+74%) |
| CND-baseline | 0.220 | | |
| CND-reinforced | | 0.298 ((+36%)) | 0.350 ((+59%)) |

Our results in Figure 4 and Table 13 also confirm that engineering robustness translates into functional gains across the four real networks studied. Regarding Fault Tolerance, in the Drosophila Connectome and Flight Map, the analysis informs the design of fault-tolerant neuromorphic circuits and identifies critical hubs where reinforcement prevents systemic transport failure. For Security and Communications, in the adversarial systems (Terrorist Cell and School Contact Network), we can calculate the theoretical robustness ceiling, by accounting for unobserved links (e.g., dormant ties in

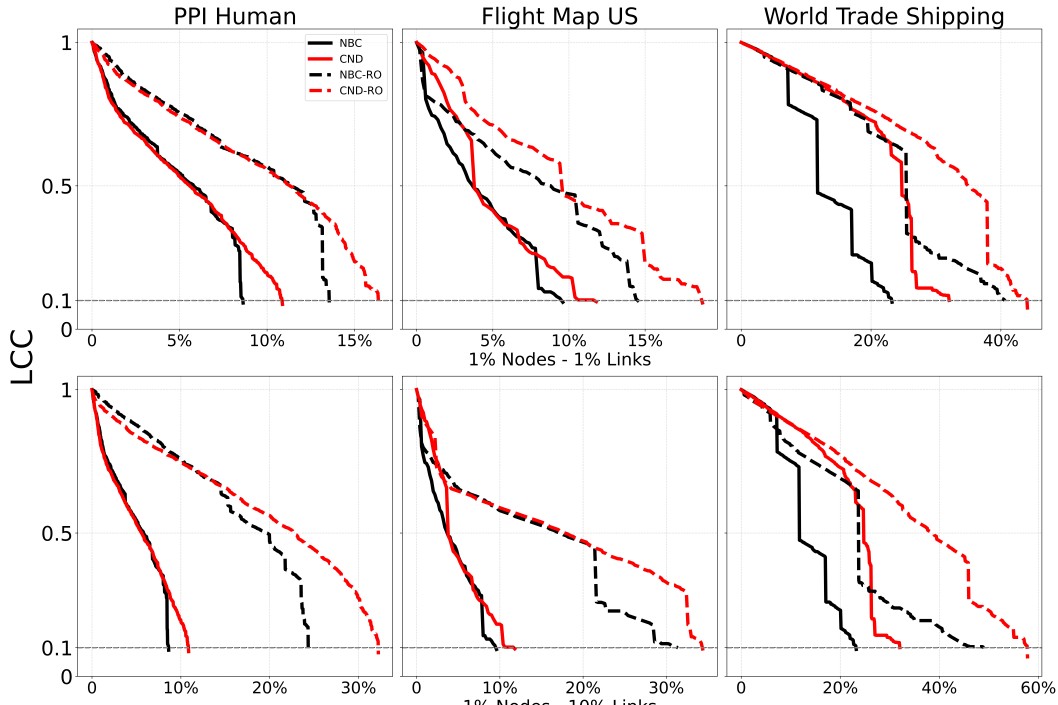

Figure 15: Dismantling curve on the original (solid) and reinforced networks (dashed). The plot shows the normalized size of the largest connected component (LCC) as a function of the fraction of nodes removed, with a target LCC threshold of 10%. Performance is evaluated using the Area Under the Curve (AUC) of the LCC trajectory.

covert networks) and quantify the margin of error required to successfully disrupt communications or secure the network against epidemics despite incomplete data.

## I   FUNCTIONAL METRICS AND REAL-WORLD APPLICATION DETAILS

We now provide the specific experimental setup for the real-world functional experiments described in Section I. Unlike topological metrics (e.g., LCC size), these experiments measure the functional performance of the systems under dismantling.

**Drosophila Connectome:**   We utilize the central brain connectome of Drosophila melanogaster (FlyWire), comprising over 125,000 neurons and 50 million synaptic connections. To assess functional performance, we employ a Leaky Integrate-and-Fire (LIF) spiking computational model, as established by Shiu et al. (2024). We focus on the sugar-sensing gustatory circuit (which results in a smaller subnetwork of 377 nodes and 13,671 edges), a critical pathway for feeding initiation. We simulate the activation of sugar-sensing Gustatory Receptor Neurons (GRNs), with a stimulation frequency of 100 Hz. We use the default parameters provided by Shiu et al. (2024), with the trial duration adjusted to 100 ms for the sake of time. The performance metric is the Motor Neuron Firing Rate, MN9. A dismantling attack is considered successful if the removal of specific interneurons prevents the propagation of the signal from the sensory GRNs to the motor neurons, causing the firing rate to drop.

**Paris/Brussels Terrorist Cell**   We analyze the network of the terrorist cell responsible for the November 2015 Paris attacks and the 2016 Brussels bombings (Gutfraind & Genkin, 2017), with 77 nodes and 271 edges. In adversarial networks, the ability of the leadership to communicate with the rest of the operational network, and vice versa, is crucial. As a result, we define Commander Reach as the percentage of network operatives (nodes) that retain a valid communication path to at least one of the three identified cell commanders.

Table 13: Network-specific evaluation metric and the original LCC AUC performance metric for four real-world networks, for NBC, CND, and RA2. $N$ denotes the number of nodes, $E$ the number of edges. Reinforced are the those reinforced by adding 10% of the links between common neighbors of the top 5% of nodes according to each method, as detailed in Appendix H. In red the improvement in robustness compared to the baseline network, assuming that a higher value indicates worse dismantling performance, meaning the network is more robust. Specific functional metrics are detailed in Appendix I. In bold the best method for each metric and network.

| Drosophila Connectome | LCC AUC | | | Sugar Firing Rate (Freq. = 100 Hz) AUC | | |
|---|---|---|---|---|---|---|
| N=377, E=13,671 | Baseline | Reinforced | | Baseline | Reinforced | |
| NBC | **0.410** | 0.465 | *+14%* | **0.030** | **0.139** | *+363%* |
| CND | 0.454 | **0.461** | *+1%* | 0.093 | 0.180 | *+93%* |
| RA2 | 0.459 | 0.480 | *+5%* | 0.042 | 0.153 | *+261%* |

| Paris/Brussels Terrorist Cell | LCC AUC | | | Commander's Reach AUC | | |
|---|---|---|---|---|---|---|
| N=77, E=273 | Baseline | Reinforced | | Baseline | Reinforced | |
| NBC | 0.123 | **0.264** | *+114%* | 0.182 | 0.305 | *+67%* |
| CND | 0.125 | 0.274 | *+120%* | 0.149 | **0.182** | *+22%* |
| RA2 | **0.114** | **0.264** | *+133%* | **0.095** | 0.274 | *+188%* |

| RyanAir Flight Map | LCC AUC | | | Global Efficiency AUC | | |
|---|---|---|---|---|---|---|
| N=128, E=601 | Baseline | Reinforced | | Baseline | Reinforced | |
| NBC | 0.149 | **0.322** | *+116%* | 0.040 | **0.111** | *+179%* |
| CND | 0.143 | 0.338 | *+136%* | **0.039** | 0.121 | *+207%* |
| RA2 | **0.142** | 0.336 | *+136%* | 0.040 | 0.116 | *+189%* |

| School Contact | LCC AUC | | | Final Outbreak Size AUC | | |
|---|---|---|---|---|---|---|
| N=327, E=5,818 | Baseline | Reinforced | | Baseline | Reinforced | |
| NBC | **0.359** | **0.417** | *+16%* | 0.351 | 0.366 | *+4%* |
| CND | 0.420 | 0.452 | *+8%* | 0.288 | 0.339 | *+17%* |
| RA2 | 0.418 | 0.457 | *+9%* | **0.285** | **0.312** | *+9%* |

**Ryanair Flight Map:** This network represents the flight routes of Ryanair in Europe (Cardillo et al., 2013), with 128 nodes and 601 edges. Nodes represent airports, and edges represent direct flight connections. We measure the functional integrity of the transport system using Global Efficiency ($E_{glob}$), rather than the Average Shortest Path Length ($APL$), because: $APL$ is mathematically undefined (or diverges to infinity) when a network fragments into disconnected components, which inevitably occurs during dismantling. Global Efficiency avoids this divergence by averaging the inverse geodesic distances.

$$E_{glob} = \frac{1}{N(N-1)} \sum_{i \neq j} \frac{1}{d_{ij}}$$

Where $N$ is the number of airports and $d_{ij}$ is the shortest path length between airport $i$ and $j$. If no path exists, $\frac{1}{d_{ij}} = 0$. This metric correctly quantifies the remaining communication capacity of a fragmented network.

**School Contact Network:** We utilize a contact network collected from a French high school (Mastrandrea et al., 2015), with 327 nodes and 5,818 edges, where nodes represent students and edges represent close-proximity physical contacts capable of disease transmission. We simulate the spread of an infectious disease using a classic Susceptible-Exposed-Infectious-Recovered (SEIR) compartmental model (Anderson & May, 1991). Unlike basic SIR models, SEIR includes an "Exposed" state to account for the latency period typical of real-world pathogens. The simulation begins with 5% of nodes infected; over 200 discrete time steps, individuals progress from Susceptible (S) to Exposed (E) (based on contact with infected neighbors and rate $\beta = 0.01$), then to Infected (I) (based on latency rate $\alpha = 0.01$), and finally to Recovered $R$ (based on recovery rate $\gamma = 0.01$). These uniform parameters were selected to establish a generic baseline, ensuring that observed variations in outbreak size are attributable to network topology rather than pathogen-specific characteristics. We average the results of 50 independent simulations. The functional integrity of the network is measured by the Final Outbreak Size, defined as the total percentage of the population that was infected and subsequently recovered. The size is normalized with the size of the largest connected component, from which the epidemic starts from.

**Engineering Network Robustness Protocol:**  To validate our method for "engineering robustness", we also reinforce these networks as defined in Section 4.7, choosing the top 5% of nodes and adding 10% of links, and rerun the dismantling process under the exact same conditions.

Table 13 shows the quantitative results for our original dismantling metric, LCC AUC, alongside the functional metrics defined above. Notably, both metrics yield highly similar rankings of the top methods, further validating our choice of LCC AUC as a robust evaluation standard."

## J  GPU ACCELERATION

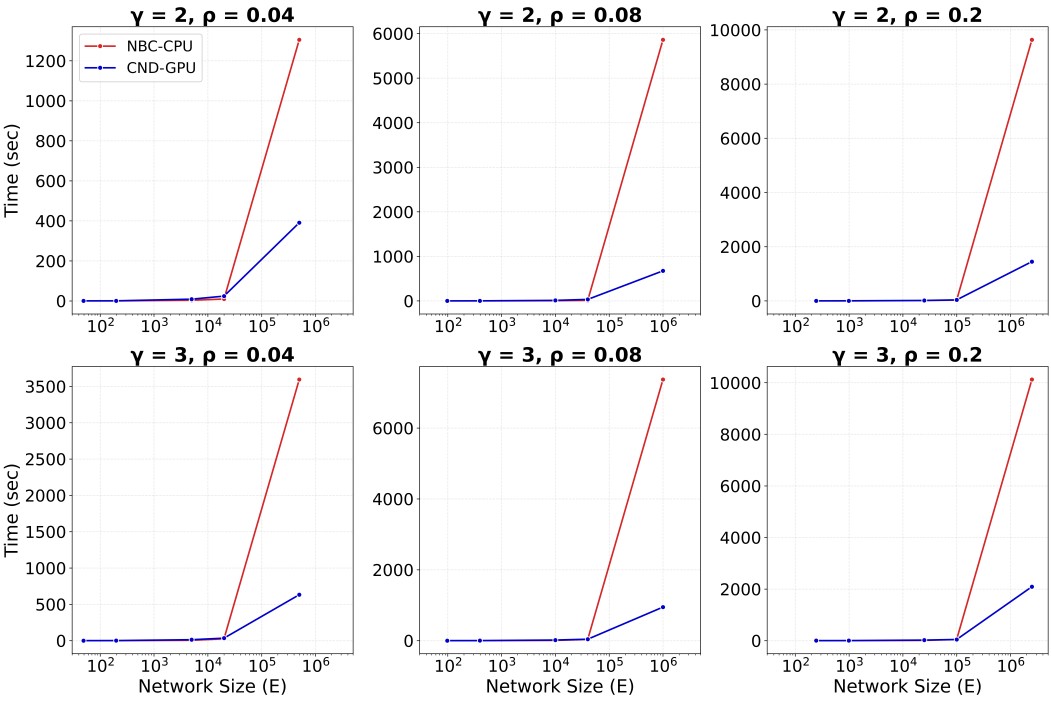

Figure 16: Running time (in seconds) comparison between CND on GPU and NBC on CPU on synthetic nPSO networks, as a function of network size in terms of edges. Experiments were conducted with network sizes ranging from 10 to 2,499,500 edges with densities ($\rho$) of 4%, 8%, and 20%, using fixed temperature $T = 0.3$ and community counts scaled by network size ($C = 2$ for $N \in \{10, 50, 100\}$, $C = 5$ for $N \in \{500, 1,000\}$, $C = 10$ for $N = 5,000$). See Figure 37 for quantitative results.

Table 14: Average runtime (in seconds) and standard error of the mean (SEM) by field and method for dynamic dismantling. Evaluated on networks of up to 23,000 nodes and 507,000 edges ($n = 1,475$). In bold the fastest method per field.$\langle N \rangle$ denotes average number of nodes and $\langle E \rangle$ number of edges.

| Field | $\langle N \rangle$ | $\langle E \rangle$ | CND-CPU | CND-GPU | NBC-CPU | RA2-CPU | RA2-GPU |
|---|---|---|---|---|---|---|---|
| Biomolecular | 2,997 | 11,855 | 1,688.9 ± 553.2 | 37.7 ± 9.6 | 174.5 ± 74.2 | 3,699.8 ± 1155.6 | **29.4 ± 7.2** |
| Brain | 97 | 1,535 | 7 ± 1.7 | 4.2 ± 0.6 | **0.2 ± 0.1** | 7.7 ± 2.6 | 2.4 ± 0.1 |
| Covert | 107 | 266 | 7.3 ± 5.5 | 0.9 ± 0.2 | **0.2 ± 0.1** | 19.8 ± 15.4 | 0.6 ± 0.1 |
| Foodweb | 117 | 1,087 | 24.2 ± 7 | 2.4 ± 0.3 | **0.2 ± 0.1** | 25.4 ± 10.8 | 1.9 ± 0.2 |
| Infrastructure | 664 | 1,332 | 2,610.8 ± 921 | 34.9 ± 11.6 | **11.6 ± 4.5** | 2,441.8 ± 861.2 | 27 ± 9.5 |
| Internet | 5,708 | 19,601 | 6,149.2 ± 853.2 | 34.2 ± 3 | 138.6 ± 15.3 | 9,801.8 ± 1176.6 | **31.9 ± 2.8** |
| Misc | 2,880 | 19,921 | 3,641.8 ± 1771.9 | 54.9 ± 15.9 | 439.3 ± 170 | 5,065.1 ± 1990.8 | **53.2 ± 21.2** |
| Social | 3,267 | 53,977 | 8,322.5 ± 1287.6 | **149.4 ± 20.1** | 4,840.6 ± 1008.5 | 12,474.4 ± 2146 | 161 ± 44 |

Since the difference in running time between the three LGD-NA methods is not relevant, neither for CPU nor GPU, we report the running time of the original RA2 in the main text (Figure 3) and the CND in the Appendix (Figure 18). When comparing CND and NBC, on the largest network,

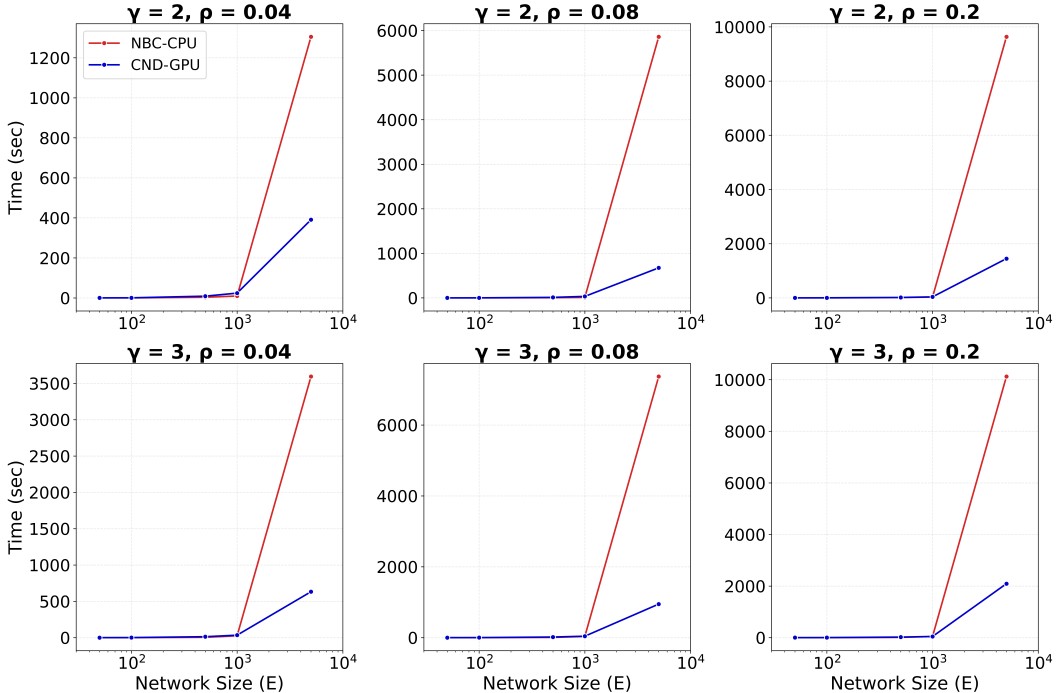

Figure 17: Running time (in seconds) comparison between CND on GPU and NBC on CPU on synthetic nPSO networks, as a function of network size in terms of nodes. Experiments were conducted with network sizes ranging from 10 to 5,000 nodes with densities ($\rho$) of 4%, 8%, and 20%, using fixed temperature $T = 0.3$ and community counts scaled by network size ($C = 2$ for $N \in \{10, 50, 100\}$, $C = 5$ for $N \in \{500, 1,000\}$, $C = 10$ for $N = 5,000$). See Figure 37 for quantitative results.

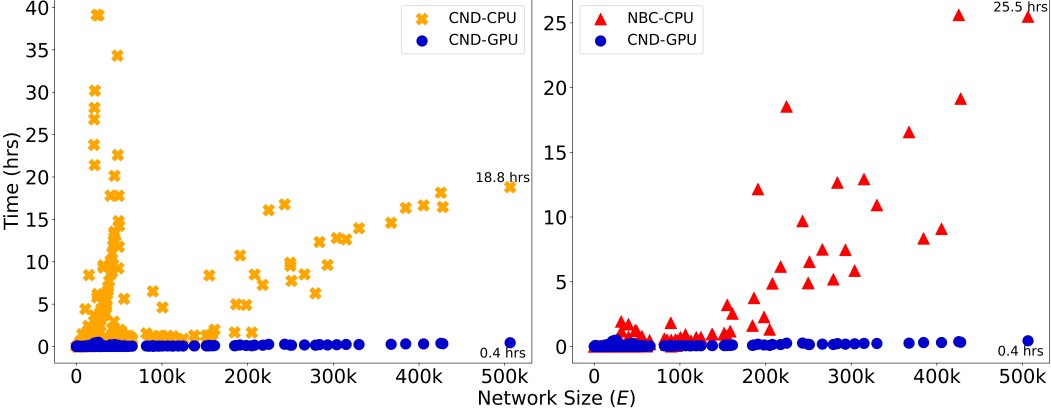

Figure 18: Runtime (in hours) is plotted against network size, measured by the number of edges, $E$, for dynamic dismantling. The annotated time indicates the runtime for the largest network. Evaluated on networks of up to 23,000 nodes and 507,000 edges ($n = 1,475$). See Figure 14 for quantitative results.

Table 15: Runtime (in seconds) for NBC run on CPU (graph-tool) and GPU (cuGraph) on a subset of networks. $N$ denotes the number of nodes and $E$ the number of edges.

|  | N | E | NBC-GPU | NBC-CPU |
|---|---|---|---|---|
| **Foodweb Blackrock** | 86 | 375 | 4.7 | 0.04 |
| **Phonecall 2012** | 193 | 1,030 | 20.7 | 0.08 |
| **Rat Transcription 2010** | 524 | 1,081 | 37.9 | 0.10 |
| **Roadmap Winnipeg** | 1,040 | 1,595 | 848.3 | 0.41 |

GPU-accelerated CND is over 46 times faster than its CPU counterpart and also over 63 times faster than NBC running on CPU.

To empirically validate these runtime advantages in a controlled setting, we conducted additional experiments using the nPSO model (Muscoloni & Cannistraci, 2018b) with network sizes ranging from 10 to 5,000 nodes, and 10 to 2,499,500 edges, with densities of 4%, 8%, and 20%. We keep the temperature (lower temperature yields higher clustering) fixed ($T = 0.3$) and adjust the number of communities to suit the size of the network ($C = 2$ for $N \in \{10, 50, 100\}$, $C = 5$ for $N \in \{500, 1,000\}$, $C = 10$ for $N \in \{5,000\}$). Our results demonstrate that GPU-accelerated LGD-NA methods begin to show running time advantages over NBC-CPU when networks exceed approximately 1,000 nodes (see Figure 17) or 100,000 edges (see Figure 16). This threshold aligns with our observations in real-world networks (see Figures 3 and 3, and Table 14), where GPU methods achieve superior running times for larger-scale networks such as biomolecular, internet, and social networks, while offering no runtime benefit for smaller networks where CPU implementations remain efficient.

Section 4.5 details how we implement the RA-based measures for GPU, using matrix multiplication. We end up with the following formula for RA2:

$$\mathbf{RA2} = \frac{1 + \mathbf{E}_{\mathrm{L2}} + \mathbf{E}_{\mathrm{L2}}^{\top} + \mathbf{E}_{\mathrm{L2}} \circ \mathbf{E}_{\mathrm{L2}}^{\top}}{1 + \mathbf{CN}_{\mathrm{L2}}}$$

and for CND:

$$\mathbf{CND} = \frac{1}{1 + \mathbf{CN}_{\mathrm{L2}}}$$

In our experiments, the CPU-based RA2 and CND implementation uses Python's `NumPy` (implemented in C) while the GPU implementation uses Python's `CuPy` (implemented in C++/CUDA).

As mentioned earlier, we report only the CPU running time for NBC, as its GPU implementation did not yield any speedup. While some studies report GPU implementations of NBC with improved performance (Fan et al., 2017; Shi & Zhang, 2011; Pande & Bader, 2011; McLaughlin & Bader, 2018; Sariyüce et al., 2013; Bernaschi et al., 2016), these are often limited by hardware-specific optimizations, data-specific assumptions (e.g., small-world, social, or biological networks), and using heuristics that are tailored to specific settings rather than offering general solutions. Moreover, publicly available code is rare, making these approaches difficult to reproduce or integrate. Overall, NBC is not naturally suited for GPU implementation, as it does not rely on matrix multiplication, but is based on computing shortest path counts between all node pairs. In our experiments, the CPU-based NBC implementation from Python's `graph_tool` (implemented in C++), based on Brandes' algorithm (Brandes, 2001) with time complexity $\mathcal{O}(Nm)$ for unweighted graphs, outperformed the GPU version from Python's `cuGraph` (implemented for C++/CUDA).

It is important to note that our LGD-NA implementation inherently utilizes a hybrid workflow: the sequential dismantling logic is managed by the CPU, while the expensive latent geometry estimations (relying on matrix multiplication) are offloaded to the GPU. This architecture is highly effective for LGD-NA but is not applicable to NBC. For NBC, the core computational burden is the calculation of all-pairs shortest paths, a task that does not lend itself well to GPU computations, meaning a hybrid pipeline yields no significant performance gain, and thus solely runs on CPU.

# K  NBC Approximators

Table 16: Average runtime (in seconds), AUC, and number of removals with their associated standard error of the mean (SEM) by field and method for dynamic dismantling. Evaluated where $N_{min} = 2,235$, $N_{max} = 9,885$, $E_{min} = 10,075$, $E_{max} = 506,437$, ($n = 157$). In bold the best method per field, by runtime, AUC, and number of removals. $\langle N \rangle$ denotes average number of nodes and $\langle E \rangle$ number of edges.

| Field | $\langle N \rangle$ | $\langle E \rangle$ | Method | Running time (sec) | AUC | Removals |
|---|---|---|---|---|---|---|
| Biomolecular | 5,528 | 26,804 | CND | **86 ± 17** | 0.087 ± 0.016 | 830 ± 134 |
| | | | NBC | 453 ± 171 | 0.082 ± 0.017 | 708 ± 124 |
| | | | NBC-20 | 437 ± 183 | 0.081 ± 0.016 | **705 ± 124** |
| | | | NBC-log | 365 ± 81 | **0.08 ± 0.015** | 733 ± 130 |
| Infrastructure | 5,803 | 15,068 | CND | 203 ± 113 | 0.099 ± 0.019 | 1,324 ± 575 |
| | | | **NBC** | **76 ± 42** | **0.021 ± 0.006** | **241 ± 13** |
| | | | NBC-20 | 105 ± 43 | 0.022 ± 0.006 | 249 ± 16 |
| | | | NBC-log | 141 ± 29 | 0.022 ± 0.006 | 266 ± 24 |
| Internet | 6,623 | 31,292 | CND | **29 ± 2** | 0.02 ± 0 | 263 ± 9 |
| | | | NBC | 196 ± 28 | **0.016 ± 0.001** | **171 ± 6** |
| | | | NBC-20 | 49 ± 5 | **0.016 ± 0.001** | **171 ± 6** |
| | | | NBC-log | 119 ± 6 | **0.016 ± 0.001** | 172 ± 6 |
| Misc | 5,444 | 53,289 | CND | **121 ± 26** | 0.134 ± 0.03 | 1,114 ± 229 |
| | | | NBC | 1,220 ± 402 | **0.106 ± 0.028** | **876 ± 215** |
| | | | NBC-20 | 655 ± 338 | **0.106 ± 0.028** | 889 ± 217 |
| | | | NBC-log | 491 ± 117 | 0.108 ± 0.028 | 929 ± 228 |
| Social | 5,778 | 156,918 | CND | **405 ± 45** | 0.325 ± 0.019 | 2,805 ± 218 |
| | | | NBC | 14,961 ± 2,782 | 0.277 ± 0.016 | **2,527 ± 206** |
| | | | NBC-20 | 1,368 ± 147 | **0.274 ± 0.016** | 2,543 ± 207 |
| | | | NBC-log | 1,363 ± 118 | 0.279 ± 0.017 | 2,745 ± 230 |

While the high computational cost of Node Betweenness Centrality (NBC) has motivated the development of numerous approximators (Bader et al., 2007; Bergamini & Meyerhenke, 2015; Haghir Chehreghani, 2013; Riondato & Kornaropoulos, 2014), comparing against them is challenging due to the scarcity of standardized, publicly available code and the complexity of their sampling algorithms, which are often performant only for specific domains or incompatible with disconnected graph structures.

To address this, we implemented two standard randomized pivoting strategies for approximation. NBC-20 estimates betweenness centrality using a random sample of 20% of the nodes. NBC-log uses a random sample of $10 * \log_2(N)$ nodes. NBC-20 prioritizes accuracy by always scanning a fixed slice of the network (20%), whereas NBC-log prioritizes speed by scanning a much smaller, logarithmically scaled subset that grows very slowly as the network gets larger. We evaluated these baselines against the exact NBC and our CND method on a subset of 157 networks selected for their size, where exact NBC calculation begins to become computationally expensive.

Table 16 reports the dismantling performance (AUC) and total runtime, averaged by field. First, we see that the NBC approximators perform comparably to the exact NBC, even occasionally outperforming it. However, this performance increase of NBC approximators should not be overstated due to the smaller sample size of this experiment. Second, while NBC approximators are significantly faster than exact NBC, CND remains faster than both approximation methods in almost all domains. A notable exception occurs in the Infrastructure field. Here, the usually slowest method NBC is actually the fastest in terms of total runtime because it takes a significantly lower number of removals to dismantle the network. Consequently, even though CND is faster per step, the NBC-based methods result in a lower total runtime simply because the dismantling threshold is reached much earlier.

Finally, it is critical to distinguish the theoretical foundations of these approaches. Existing NBC approximators focus on accelerating the estimation of a global metric. In contrast, LGD-NA leverages purely local topological information to directly estimate pairwise distances in the latent metric space. This distinction allows LGD-NA to bypass the need for global knowledge or more complex sampling

strategies. Although approximation techniques improve NBC's speed, their reliance on sampling global paths is inherently less efficient than our strictly local approach, as we validate in Table 16. Furthermore, sampling global information remains vulnerable to missing data and adversarial noise. The strength of LGD-NA, therefore, lies in its ability to achieve high dismantling performance by directly utilizing local geometric insights, rather than attempting to approximate a computationally intensive global metric.

## L    APPLICATION TO DIRECTED NETWORKS

While network dismantling has primarily targeted undirected networks Artime et al. (2024), many critical real-world systems, such as neural circuits, social media platforms, and financial transaction systems, are inherently directed. Although our LGD-NA framework is designed for undirected network, we explore its applicability to directed graphs. Research on directed network dismantling is relatively underexplored. Directed Node Entanglement (DNE) (Wu et al., 2025) generalizes the network density matrix to specifically capture and disrupt directed information flow within a system. Ma et al. (2022) utilizes the non-backtracking matrix to identify and remove the minimum set of nodes connecting distinct edge modules. Dismantling on Signed Networks based on Evolutionary Deep Reinforcement Learning (DSEDR) (Ou et al., 2024) is an evolutionary deep reinforcement learning approach designed to dismantle signed networks by optimizing a novel objective function. Embedding-based Signed Network Dismantling (ESND) (Xie et al., 2025) combines giant component detection, network embedding, and node clustering to identify critical nodes.

We evaluate our LGD-NA framework on directed graphs using two approaches: applying the original method (treating the graph as undirected) and a directed variant where node scores are computed aggregating only outgoing links. For comparison, we implement Directed Node Entanglement (DNE) as a baseline specific to directed networks. We exclude other methods—such as the set-based approach of Ma et al. (2022) and the complex, multi-step frameworks of DSEDR and ESND—to maintain a focused comparison on computationally efficient, node-ranking strategies.

Table 17: LCC AUC for the three LGD-NA estimators treating a directed network as undirected, their directed variant, and DNE, a specific directed network dismantling method. In bold the best performing method per network. $N$ denotes the number of nodes and $E$ the number of edges.

|  | College Messages $N=1,899, E=20,296$ | Drosophila Connectome $N=377, E=13,671$ | Email EU $N=986, E=24,929$ |
|---|---|---|---|
| **CND** | **0.1426** | **0.4470** | **0.2710** |
| **CND-Directed** | 0.1537 | 0.4512 | 0.3064 |
| **RA2** | 0.1427 | 0.4529 | 0.2824 |
| **RA2-Directed** | 0.1466 | 0.4507 | 0.2955 |
| **RA2$_{num}$** | 0.1431 | 0.4625 | 0.2873 |
| **RA2$_{num}$-Directed** | 0.1485 | 0.4653 | 0.2984 |
| **DNE** | 0.4069 | 0.4923 | 0.4311 |

We focus on three directed networks from different fields. College Messages (Social) (Panzarasa et al., 2009) represents private messages on a social network at UC Irvine (1,899 nodes, 20,296 edges). Drosophila Connectome (Brain) (Shiu et al., 2024), focuses on the sugar-sensing gustatory circuit (377 nodes, 13,671 edges). Email EU (Social)(Paranjape et al., 2017) are anonymized internal emails from a European research institution (986 nodes, 24,929 edges).

Table 17 shows two key results. First, for our LGD-NA framework, treating directed networks as undirected yields superior dismantling performance over our directed variant. Second, even when applied in this undirected way, LGD-NA outperforms the directed-specific DNE baseline, demonstrating its effectiveness on directed networks.

Finally, our framework is predicated on the association between network topology and a latent geometry. For directed networks, this relationship is less established, with only preliminary studies (Allard et al., 2024). Therefore, designing asymmetric network measures that account for directional flow remains an open challenge, yet a promising direction for future research.

## M  EXPERIMENTAL SETUP

**Baseline Topological Centrality Measures.**  We selected centrality measures to cover diverse categories: shortest path-based (NBC), degree-based (degree), walk-based (eigenvector), random walk-based (PageRank), resilience-based (Resilience), and fitness-based (Domirank and Fitness centrality). We also tested closeness and load centrality, but both performed worse than NBC and rely on the same shortest-path principle; thus, we retained NBC. Similarly, Katz centrality underperformed compared to eigenvector centrality and is also based on spectral properties of the adjacency matrix. For DomiRank, we tested three values for the numerator in the $\sigma$ parameter formula: 0.1, 0.5, and 0.9. While the original study sometimes performs a grid search to find the optimal $\sigma$ per network, this is not feasible for our large-scale evaluation. Instead, we selected a representative range and found that $\sigma_{num} = 0.5$ yielded the best performance, and we report that value. The parameter $\sigma$ controls the balance between local degree-based dominance and global network-structure-based competition. As $\sigma \to 0$, the scores approximate degree centrality. As $\sigma$ increases, nodes are evaluated in increasingly competitive environments, where centrality depends more on non-local structural dominance than individual connectivity. For Fitness centrality, we capped the number of iterations at $k = 100$. Without this cap, the method took a prohibitively long time to converge, especially on large networks.

**Baseline Dismantling Methods.**  We selected the best-performing and most widely adopted dismantling algorithms from the literature (Artime et al., 2024). As mentioned earlier, we did not include BPD and FINDER in our experiments due to unavailable and outdated code, respectively. For Collective Influence (CI), we tested values $\ell = 1, 2, 3$, where $\ell$ defines the radius of a ball centered at node $i$, and $\partial B(i, \ell)$ is the frontier at distance $\ell$ (i.e., nodes exactly $\ell$ hops away). We found that $\ell = 1$ performed best across our benchmarks and report this setting, while $\ell = 3$ performed the worst. For Explosive Immunization (EI), we evaluated both scores $\sigma^{(1)}$ and $\sigma^{(2)}$. The $\sigma^{(1)}$ score targets high-connectivity nodes to rapidly fragment the network early on. The $\sigma^{(2)}$ score aims to prevent large cluster merges near the percolation threshold by avoiding the connection of separate components. We found that $\sigma^{(1)}$ consistently outperformed $\sigma^{(2)}$, and thus we use it in our final experiments. For eigenvector centrality, we capped the number of power iterations at $k = 100$ to avoid long or unbounded runtimes, since convergence can be very slow in large networks. For PageRank, we used a convergence tolerance of $\epsilon = 10^{-6}$, as the algorithm runs until the change in scores falls below this threshold.

**LGD-NA Measures.**  Our analysis of pure win rates (draws are excluded) in Table 18 reveals distinct domain-specific strengths. CND achieves the highest win rate in Biomolecular, Foodweb, Infrastructure, Internet, and Social networks. In contrast, RA2 is the preferred method for Brain and Covert networks. Notably, $RA2_{num}$ does not emerge as the top-performing method in any of the tested domains.

**Robustness to Threshold Variations.**  As shown in Table 19, the mean field rankings of the methods are broadly consistent across removal thresholds of 10%, 25%, and 50%. While permutations occur, the dominance of NBC and CND is consistent across different thresholds, confirming that our conclusions are not artifacts of the 10% threshold.

**Note on Reinsertion.**  We did not apply reinsertion techniques on all networks included in the initial dismantling experiments. In some cases, certain methods performed so poorly that applying reinsertion became prohibitively slow. To ensure consistency, we excluded these networks from the reinsertion analysis for all methods. Specifically, we imposed a cutoff: networks were excluded if any method required more than 800 node removals to reach the dismantling threshold. Based on this criterion, 59 networks were excluded.

Table 18: Pure win rate (draws excluded), for LGD-NA measures, without reinsertion ($n = 1{,}296$). In bold the method with the highest win rate per field.

|  | CND | RA2 | $RA2_{num}$ |
|---|---|---|---|
| **Biomolecular** | **52%** | 30% | 17% |
| **Brain** | 28% | **68%** | 4% |
| **Covert** | 16% | **42%** | 41% |
| **Foodweb** | **67%** | 16% | 16% |
| **Infrastructure** | **65%** | 16% | 19% |
| **Internet** | **52%** | 46% | 2% |
| **Misc** | 41% | **45%** | 14% |
| **Social** | **51%** | 29% | 20% |

Table 19: Mean field ranking with standard error of the mean (SEM), for different threshold levels ($n = 1{,}296$). In bold the best method per threshold.

| Threshold | 10% | 25% | 50% |
|---|---|---|---|
| **NBC** | **1 ± 0** | **1 ± 0** | **1 ± 0** |
| **CND** | 3.625 ± 0.56 | 3.5 ± 0.5 | 3.25 ± 0.25 |
| **RA2** | 4 ± 0.93 | 3.875 ± 0.79 | 5.125 ± 0.77 |
| **$RA2_{num}$** | 4.875 ± 0.72 | 5.5 ± 0.76 | 8.375 ± 0.73 |
| **GDM** | 5 ± 0.5 | 5.25 ± 0.56 | 4.875 ± 0.58 |
| **CoreGDM-NR** | 6.125 ± 0.52 | 6.25 ± 0.62 | 7.25 ± 0.86 |
| **GND** | 8.875 ± 2.07 | 7.75 ± 2.09 | 5.875 ± 1.83 |
| **Degree** | 8.875 ± 0.64 | 9 ± 0.63 | 9.75 ± 0.8 |
| **Domirank $\sigma_{0.5}$** | 8.875 ± 0.64 | 8.625 ± 0.6 | 8.75 ± 0.59 |
| **PR** | 9.125 ± 1.08 | 8.375 ± 1.05 | 6.5 ± 0.96 |
| **Fitness** | 9.5 ± 1.34 | 8.625 ± 1.03 | 7.375 ± 1.02 |
| **Resilience** | 11.125 ± 0.4 | 11.625 ± 0.42 | 12.375 ± 0.42 |
| **$CI_{L1}$** | 13 ± 0.63 | 12.625 ± 0.42 | 13.125 ± 0.44 |
| **EI $\sigma_1$** | 13.625 ± 1.29 | 15.25 ± 0.45 | 15.5 ± 0.38 |
| **Eigenvector** | 14.25 ± 0.31 | 14.375 ± 0.32 | 14.5 ± 0.46 |
| **MS** | 15.125 ± 0.61 | 15.375 ± 0.56 | 14.625 ± 1.27 |
| **CoreHD-NR** | 16 ± 0.5 | 16 ± 0.38 | 14.75 ± 1.44 |

**LCC AUC as the evaluation metric.**   We employ LCC AUC as our primary evaluation metric because it provides a unified standard for comparing methods across 32 distinct complex system domains. As the established standard in the vast majority of dismantling studies (Artime et al., 2024), it allows for direct comparisons between diverse networks from disparate fields and different dismantling algorithms. While dynamical metrics offer specific insights, simulating 'live' system dynamics for every network is computationally unfeasible and conceptually inconsistent given the broad scope of domains. Crucially, our functional analysis in Section 4.7 demonstrates that rankings based on LCC AUC align closely with domain-specific functional metrics, validating LCC fragmentation as a reliable proxy for functional disruption. Note that a lower number of removals does not always imply a lower AUC. Our AUC metric rewards methods that fragment the network early, even if they require more steps to reach the dismantling threshold. As shown in Figure 19, we show cases where a method that reaches the threshold with more removals can still achieve a lower AUC, due to earlier damage to the network structure.

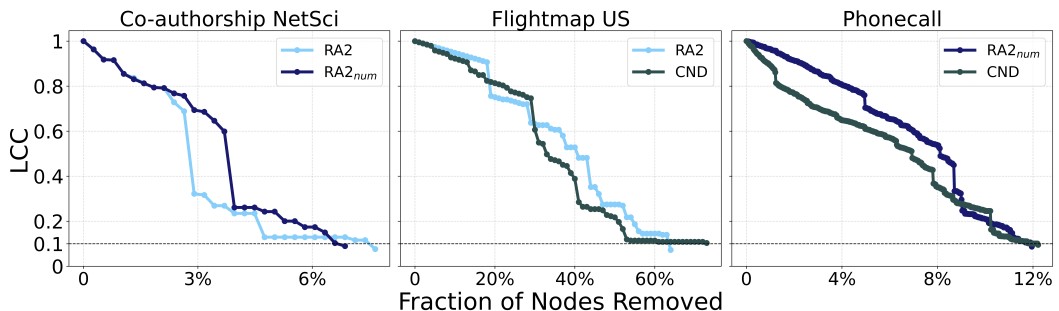

Figure 19: Dynamic dismantling process on example networks comparing CND, RA2 and $RA2_{num}$, showing a lower removal number does not necessarily mean a lower Area Under the Curve (AUC) of the LCC trajectory. The plot shows the normalized size of the largest connected component (LCC) as a function of the fraction of nodes removed, with a target LCC threshold of 10%.

## N   TECHNICAL IMPLEMENTATION AND REPRODUCIBILITY

**Code.**   All our methods are implemented in our codebase, available in the supplementary material. We implement RA2, $RA2_{num}$, CND, NBC, as well as degree and eigenvector centrality. We also adapt and integrate the original implementations of Domirank centrality and Fitness centrality. Since no code was available for Resilience centrality, we implemented it ourselves based on the original description in the paper. Instructions for running these methods and reproducing the experiments are included in the supplementary material. We also provide a representative network from the ATLAS dataset for testing purposes. For GDM, CoreGDM, GND, CI, EI, MS, and CoreHD, we use the publicly available code from Artime et al. (2024)'s review.

**Computational Resources.**   All experiments were conducted on a machine equipped with an AMD Ryzen Threadripper PRO 3995WX CPU (64 cores), 251 GiB of RAM, and a single NVIDIA RTX A4000 GPU with 16 GiB of memory. All code was implemented in Python, with dependencies and library versions specified in the supplementary material to ensure full reproducibility.

**Quantitative Results.**   All results are in Tables 23, 24, 25, 26, 27, 28, 29, 30, 31, 32, 33, 34, and 35 in the Appendix.

## O   DISCUSSION, LIMITATIONS, AND FUTURE WORK

**Missing or Manipulated Data.**   The robustness of our LGD-NA methods to missing or manipulated information, such as missing neighbor data or adversarial modifications to neighborhood structure, is a crucial question for practical applications of any network analysis method, especially those relying on local information. Our LGD-NA methods, by design, are inherently more robust to global missing or manipulated information compared to methods that require a complete global view of the

network (e.g., exact NBC). Since our methods rely solely on local neighborhood information, random missing data across the network would primarily affect only the scores of directly impacted nodes, rather than propagating errors throughout the entire graph. Similarly, adversarial modifications would need to target specific local neighborhoods to significantly alter a node's dismantling score, making large-scale, coordinated attacks more challenging than for global metrics. However, we acknowledge that direct adversarial manipulation of a node's immediate neighborhood could indeed impact its calculated score.

**Weighted Networks.** While many real-world systems, such as citation networks, neural synaptic connections, transportation and trade networks, contain directed interactions, our LGD-NA measures focus on unweighted topologies for two key reasons. First, in practical dismantling contexts, weight data is often dynamic, temporal, or unavailable, whereas topology is more robust. Second, current state-of-the-art dismantling algorithms focus on unweighted graphs (Artime et al., 2024), making fair comparisons impossible. Consequently, we consider the extension of latent-geometry approaches to weighted graphs as a promising avenue for future work.

**Limitations.** A limitation of this study is the mismatch between theoretical and observed runtimes, which can vary across methods. These differences stem from factors such as the programming language used, hardware acceleration, and implementation-level optimizations. However, all experiments were run on the same CPU and GPU to ensure a fair comparison, and we made a strong effort to optimize all methods both in terms of runtime and dismantling performance. For example, we tested different parameters for Domirank, CI, and EI, and evaluated multiple variants of shortest path-based and spectral partitioning-based centrality measures. Furthermore, we were unable to test on extremely large networks due to hardware constraints and the high computational cost of running a broad set of dismantling methods. However, we are confident that our results would generalize to larger networks, given the diversity of the 1,475 networks tested, spanning a wide range of domains and sizes from very small to large. Another limitation relates to the parameter tuning required by some baseline methods, especially machine learning-based approaches and Domirank. Due to the scale of our experimental setup—both in the number and size of networks—we were unable to perform extensive tuning. Although targeted tuning could enhance performance for specific methods on individual networks, it would compromise consistency across the wide range of complex systems domains considered. In contrast, LGD-NA requires no parameter tuning and consistently achieves strong, generalizable performance across all tested networks.

**Future Work.** Future research could further explore latent geometry, particularly how to effectively combine local and global information in dismantling strategies. Improving the scalability of matrix-based computations, especially for very large and sparse networks, is another important direction. There is also a need for more cost-efficient dynamic dismantling strategies that reduce the overhead of recomputing scores after every node removal without significantly sacrificing performance. In addition, edge dismantling remains a relatively underexplored area compared to node-based dismantling, and it would be valuable to investigate whether latent geometry-driven principles can also guide the efficient removal of links in complex networks. Targeting edges can be just as important as targeting nodes, and in many real-world systems, such as transportation networks (railroads, roads, subways, or shipping trade routes), edge removal may represent the more realistic and sensible threat scenario, making it highly relevant for dismantling strategies. Finally, our work can also be of interest to Explainable AI (XAI) for models like GNNs and Reservoir Computers (RC). By identifying critical (dismantling) or unimportant (percolation) nodes in a network, we can conduct "perturbation experiments" to assess their impact on an ML model's performance. This process reveals which parts of the graph are most crucial for the model's predictions or computations, effectively opening the model's "black box" and providing crucial insights into its internal decision-making, robustness, and vulnerabilities.

## P   ETHICS STATEMENT

The research on network dismantling presented in this paper has a potential for dual use. The techniques developed to identify and exploit network vulnerabilities could theoretically be used to design targeted attacks on critical systems, such as communication, transportation, or power grid networks. However, it can also be used for defensive strategies. A thorough understanding of network

vulnerabilities is a prerequisite for designing robust systems. To directly address the dual-use concern, we demonstrate the constructive potential of our work by presenting a novel technique for proactively engineering network robustness using our latent geometry-based methods (see Section 4.6). This application moves beyond vulnerability analysis to provide an explainable framework for modifying a network to enhance its resilience. Finally, by openly publishing our theoretical foundations, source code, and comprehensive evaluations, we aim to ensure that the benefits of this research, namely the ability to secure critical networks, are accessible to all. We believe the societal benefit of advancing defensive capabilities significantly outweighs the risk of potential misuse.

## Q    REPRODUCIBILITY STATEMENT

To ensure full reproducibility, we have made our source code publicly available, including detailed instructions on how to replicate all experiments. The codebase includes an implementation of our LGD-NA framework (illustrated in Figure 1), the exact formulas used (detailed in Appendix A), and an example network for demonstration. The code is compatible with both CPU and GPU environments and also provides the necessary tools to engineer network robustness as described in this work. The baseline methods were implemented using the code from the review by Artime et al. (2024). The exact topological measures of all networks used in our study are provided in Appendix 9. Further details regarding the experimental setup, including hardware specifications, are described in Appendix M and  N.

## R    CLAIM OF THE LLM USAGE

We used LLM-based tools to improve the language and flow; the principles, core logic, and innovations are entirely the authors'.

## SUPPLEMENTARY MATERIAL

Table 20: Pearson correlation between between all the pairwise hyperbolic distances of the network nodes in the original nPSO model and in the reconstructed hyperbolic space (HD-correlation) (Muscoloni et al., 2017). Mean values over 10 seeds and Standard Error of the Mean (SEM) are reported, and the Fisher p-value in parentheses. The power-law exponent $\gamma$ represents the scale-freeness found in real-world networks. networks. $\rho$ is the density of the networks. Fixed parameters are the number of nodes, $N = 500$, and the number of communities, $C = 5$. The temperature $T$ controls the level of clustering (lower temperatures yield stronger clustering).

| N=500, C=5 | | | $\rho$=0.04 | $\rho$=0.08 | $\rho$=0.2 |
|---|---|---|---|---|---|
| $\gamma$=2 | CND | T=0.3 | 0.722 ± 0.005 (0.000) | 0.792 ± 0.002 (0.000) | 0.846 ± 0.001 (0.000) |
| | | T=0.6 | 0.693 ± 0.004 (0.000) | 0.768 ± 0.002 (0.000) | 0.801 ± 0.001 (0.000) |
| | | T=0.9 | 0.633 ± 0.008 (0.000) | 0.765 ± 0.003 (0.000) | 0.777 ± 0.001 (0.000) |
| | RA2 | T=0.3 | 0.521 ± 0.007 (0.000) | 0.532 ± 0.007 (0.000) | 0.521 ± 0.010 (0.000) |
| | | T=0.6 | 0.524 ± 0.007 (0.000) | 0.484 ± 0.010 (0.000) | 0.308 ± 0.008 (0.000) |
| | | T=0.9 | 0.498 ± 0.003 (0.000) | 0.460 ± 0.004 (0.000) | 0.303 ± 0.008 (0.000) |
| $\gamma$=3 | CND | T=0.3 | 0.584 ± 0.004 (0.000) | 0.624 ± 0.003 (0.000) | 0.645 ± 0.001 (0.000) |
| | | T=0.6 | 0.510 ± 0.005 (0.000) | 0.579 ± 0.002 (0.000) | 0.590 ± 0.002 (0.000) |
| | | T=0.9 | 0.452 ± 0.004 (0.000) | 0.552 ± 0.003 (0.000) | 0.597 ± 0.003 (0.000) |
| | RA2 | T=0.3 | 0.685 ± 0.009 (0.000) | 0.714 ± 0.006 (0.000) | 0.783 ± 0.008 (0.000) |
| | | T=0.6 | 0.722 ± 0.003 (0.000) | 0.780 ± 0.003 (0.000) | 0.805 ± 0.006 (0.000) |
| | | T=0.9 | 0.688 ± 0.002 (0.000) | 0.795 ± 0.004 (0.000) | 0.755 ± 0.010 (0.000) |

Table 21: Pearson correlation between estimated link weights from CND and RA2 versus true geometric distances in nPSO networks. Mean values over 10 seeds are reported, with a color gradient where green corresponds to values approaching 1 and red to values approaching -1. The power-law exponent $\gamma$ represents the scale-freeness found in real-world networks. networks. $\rho$ is the density of the networks. The temperature $T$ controls the level of clustering (lower temperatures yield stronger clustering). Fixed parameters are the number of nodes, $N = 500$, and the number of communities, $C = 5$. Standard Error of the Mean (SEM) and Fisher p-value are found in Table 22.

| N=500, C=5 | | | $\rho$=0.04 | $\rho$=0.08 | $\rho$=0.2 |
|---|---|---|---|---|---|
| $\gamma$=2 | CND | T=0.3 | 0.534 | 0.641 | 0.664 |
| | | T=0.6 | 0.602 | 0.675 | 0.719 |
| | | T=0.9 | 0.649 | 0.690 | 0.746 |
| | RA2 | T=0.3 | -0.044 | 0.062 | 0.319 |
| | | T=0.6 | 0.066 | 0.239 | 0.562 |
| | | T=0.9 | 0.140 | 0.361 | 0.682 |
| $\gamma$=3 | CND | T=0.3 | 0.329 | 0.370 | 0.394 |
| | | T=0.6 | 0.543 | 0.512 | 0.473 |
| | | T=0.9 | 0.607 | 0.553 | 0.510 |
| | RA2 | T=0.3 | 0.301 | 0.388 | 0.530 |
| | | T=0.6 | 0.441 | 0.542 | 0.625 |
| | | T=0.9 | 0.473 | 0.588 | 0.669 |

Table 22: Pearson correlation between estimated link weights from CND and RA2 versus true geometric distances in nPSO networks. Mean values over 10 seeds is reported and Standard Error of the Mean (SEM) are reported, and the Fisher p-value in parentheses. The power-law exponent $\gamma$ represents the scale-freeness found in real-world networks. networks. $\rho$ is the density of the networks. Fixed parameters are the number of nodes, $N = 500$, and the number of communities, $C = 5$. The temperature $T$ controls the level of clustering (lower temperatures yield stronger clustering).

| N=500, C=5 | | | $\rho$=0.04 | $\rho$=0.08 | $\rho$=0.2 |
|---|---|---|---|---|---|
| $\gamma$=2 | CND | T=0.3 | 0.534 ± 0.006 (0.000) | 0.641 ± 0.004 (0.000) | 0.664 ± 0.001 (0.0) |
| | | T=0.6 | 0.602 ± 0.003 (0.000) | 0.675 ± 0.005 (0.000) | 0.719 ± 0.001 (0.0) |
| | | T=0.9 | 0.649 ± 0.002 (0.000) | 0.690 ± 0.004 (0.000) | 0.746 ± 0.001 (0.0) |
| | RA2 | T=0.3 | -0.044 ± 0.004 (0.000) | 0.062 ± 0.003 (0.000) | 0.319 ± 0.003 (0.0) |
| | | T=0.6 | 0.066 ± 0.004 (0.000) | 0.239 ± 0.003 (0.000) | 0.562 ± 0.002 (0.0) |
| | | T=0.9 | 0.140 ± 0.005 (0.000) | 0.361 ± 0.004 (0.000) | 0.682 ± 0.002 (0.0) |
| $\gamma$=3 | CND | T=0.3 | 0.329 ± 0.005 (0.000) | 0.370 ± 0.003 (0.000) | 0.394 ± 0.006 (0.0) |
| | | T=0.6 | 0.543 ± 0.006 (0.000) | 0.512 ± 0.002 (0.000) | 0.473 ± 0.004 (0.0) |
| | | T=0.9 | 0.607 ± 0.003 (0.000) | 0.553 ± 0.001 (0.000) | 0.510 ± 0.003 (0.0) |
| | RA2 | T=0.3 | 0.301 ± 0.006 (0.000) | 0.388 ± 0.003 (0.000) | 0.530 ± 0.003 (0.0) |
| | | T=0.6 | 0.441 ± 0.003 (0.000) | 0.542 ± 0.003 (0.000) | 0.625 ± 0.003 (0.0) |
| | | T=0.9 | 0.473 ± 0.003 (0.000) | 0.588 ± 0.002 (0.000) | 0.669 ± 0.003 (0.0) |

| AUC | $CI_{L1}$ | $CI_{L2}$ | $CI_{L3}$ | CND-D | CND-S | CoreGDM-NR | CoreHD-NR | Degree | Domirank $\sigma_{0.1}$ | Domirank $\sigma_{0.5}$ | Domirank $\sigma_{0.9}$ |
|---|---|---|---|---|---|---|---|---|---|---|---|
| Biomolecular | 0.0794 ± 0.021 | 0.0722 ± 0.018 | 0.0879 ± 0.024 | 0.0721 ± 0.018 | 0.0825 ± 0.021 | 0.0772 ± 0.02 | 0.0951 ± 0.021 | 0.0783 ± 0.021 | 0.0793 ± 0.021 | 0.0786 ± 0.021 | 0.0792 ± 0.02 |
| Brain | 0.4177 ± 0.002 | 0.4367 ± 0.002 | 0.4746 ± 0.002 | 0.3883 ± 0.002 | 0.4192 ± 0.002 | 0.408 ± 0.002 | 0.4244 ± 0.002 | 0.4158 ± 0.002 | 0.4153 ± 0.002 | 0.4147 ± 0.002 | 0.4133 ± 0.002 |
| Covert | 0.1407 ± 0.009 | 0.1949 ± 0.013 | 0.2819 ± 0.013 | 0.1223 ± 0.009 | 0.1364 ± 0.009 | 0.1208 ± 0.009 | 0.1709 ± 0.01 | 0.1282 ± 0.009 | 0.1282 ± 0.009 | 0.1286 ± 0.009 | 0.1268 ± 0.009 |
| Foodweb | 0.2346 ± 0.01 | 0.2428 ± 0.013 | 0.4134 ± 0.009 | 0.215 ± 0.01 | 0.2707 ± 0.012 | 0.2175 ± 0.01 | 0.2472 ± 0.01 | 0.2199 ± 0.01 | 0.2194 ± 0.01 | 0.2156 ± 0.009 | 0.2177 ± 0.009 |
| Infrastructure | 0.0906 ± 0.005 | 0.1347 ± 0.007 | 0.2742 ± 0.007 | 0.0481 ± 0.003 | 0.0519 ± 0.003 | 0.0466 ± 0.003 | 0.0593 ± 0.004 | 0.0491 ± 0.003 | 0.0498 ± 0.003 | 0.0498 ± 0.003 | 0.0495 ± 0.003 |
| Internet | 0.0612 ± 0.003 | 0.0587 ± 0.003 | 0.0907 ± 0.009 | 0.0579 ± 0.003 | 0.0675 ± 0.003 | 0.0597 ± 0.003 | 0.0776 ± 0.003 | 0.0608 ± 0.003 | 0.0613 ± 0.003 | 0.0615 ± 0.003 | 0.0625 ± 0.003 |
| Misc | 0.2127 ± 0.024 | 0.2133 ± 0.024 | 0.2815 ± 0.029 | 0.2043 ± 0.023 | 0.2307 ± 0.025 | 0.2075 ± 0.024 | 0.2338 ± 0.023 | 0.2119 ± 0.023 | 0.2128 ± 0.023 | 0.2122 ± 0.023 | 0.213 ± 0.023 |
| Social | 0.2506 ± 0.014 | 0.2638 ± 0.014 | 0.3045 ± 0.015 | 0.2319 ± 0.013 | 0.2527 ± 0.014 | 0.2399 ± 0.014 | 0.2696 ± 0.013 | 0.2472 ± 0.014 | 0.2458 ± 0.014 | 0.2449 ± 0.014 | 0.2462 ± 0.014 |

| AUC | $EI_{\sigma 1}$ | $EI_{\sigma 2}$ | Eigenvector | Fitness | GDM | GND | MS | NBC | PR | RA2 | $RA2_{num}$ | Resilience |
|---|---|---|---|---|---|---|---|---|---|---|---|---|
| Biomolecular | 0.0782 ± 0.019 | 0.389 ± 0.014 | 0.0828 ± 0.022 | 0.0798 ± 0.021 | 0.0769 ± 0.02 | 0.0749 ± 0.015 | 0.0961 ± 0.021 | **0.0579 ± 0.013** | 0.0793 ± 0.021 | 0.0717 ± 0.018 | 0.0742 ± 0.019 | 0.079 ± 0.021 |
| Brain | 0.4128 ± 0.002 | 0.4791 ± 0.001 | 0.421 ± 0.002 | 0.415 ± 0.002 | 0.4074 ± 0.002 | 0.3957 ± 0.002 | 0.424 ± 0.002 | **0.3617 ± 0.002** | 0.4156 ± 0.002 | 0.3792 ± 0.002 | 0.3979 ± 0.002 | 0.4173 ± 0.002 |
| Covert | 0.1772 ± 0.01 | 0.3636 ± 0.009 | 0.1393 ± 0.009 | 0.1282 ± 0.009 | 0.1144 ± 0.009 | 0.1302 ± 0.01 | 0.1685 ± 0.01 | **0.1104 ± 0.009** | 0.1346 ± 0.009 | 0.1144 ± 0.009 | 0.1141 ± 0.009 | 0.1303 ± 0.01 |
| Foodweb | 0.249 ± 0.009 | 0.3903 ± 0.01 | 0.244 ± 0.01 | 0.2116 ± 0.009 | 0.2167 ± 0.01 | 0.2641 ± 0.012 | 0.2407 ± 0.01 | **0.1948 ± 0.009** | 0.2128 ± 0.01 | 0.2244 ± 0.01 | 0.2243 ± 0.01 | 0.2337 ± 0.011 |
| Infrastructure | 0.0926 ± 0.005 | 0.3873 ± 0.005 | 0.0837 ± 0.004 | 0.0499 ± 0.003 | 0.0455 ± 0.003 | 0.06 ± 0.003 | 0.0581 ± 0.004 | **0.0404 ± 0.003** | 0.0508 ± 0.003 | 0.0472 ± 0.003 | 0.0477 ± 0.003 | 0.0571 ± 0.004 |
| Internet | 0.0666 ± 0.003 | 0.4528 ± 0.001 | 0.0629 ± 0.003 | 0.062 ± 0.003 | 0.0594 ± 0.003 | 0.0723 ± 0.004 | 0.0768 ± 0.003 | **0.049 ± 0.003** | 0.0605 ± 0.003 | 0.0581 ± 0.003 | 0.0593 ± 0.003 | 0.0609 ± 0.003 |
| Misc | 0.2217 ± 0.023 | 0.4449 ± 0.01 | 0.2173 ± 0.023 | 0.2131 ± 0.023 | 0.2071 ± 0.024 | 0.2068 ± 0.023 | 0.2313 ± 0.022 | **0.1712 ± 0.022** | 0.2115 ± 0.023 | 0.2053 ± 0.024 | 0.2073 ± 0.023 | 0.2124 ± 0.024 |
| Social | 0.2525 ± 0.014 | 0.4329 ± 0.006 | 0.2549 ± 0.014 | 0.2448 ± 0.014 | 0.2389 ± 0.014 | 0.2306 ± 0.014 | 0.2683 ± 0.013 | **0.2039 ± 0.013** | 0.2455 ± 0.014 | 0.2348 ± 0.014 | 0.2343 ± 0.014 | 0.2491 ± 0.014 |

Table 23: Mean and standard error of the mean (SEM) for the AUC per field, by dismantling method ($n = 1,296$). In bold the best method per field.

Table 24: Mean and standard error of the mean (SEM) for the number of removals per field, by dismantling method ($n = 1{,}296$). In bold the best method per field.

| Removals | $CI_{L1}$ | $CI_{L2}$ | $CI_{L3}$ | CND-D | CND-S | CoreGDM-NR | CoreHD-NR | Degree | Domirank $\sigma_{0.1}$ | Domirank $\sigma_{0.5}$ | Domirank $\sigma_{0.9}$ |
|---|---|---|---|---|---|---|---|---|---|---|---|
| Biomolecular | 259.1 ± 71 | 247.9 ± 70.2 | 258.3 ± 69.4 | 264.3 ± 72.7 | 323.9 ± 93.9 | 264.7 ± 75.5 | 272.7 ± 71.6 | 272.2 ± 74.2 | 280.3 ± 76.1 | 284.4 ± 76.3 | 295.3 ± 78.1 |
| Brain | 60.7 ± 1.7 | 71.9 ± 2 | 81.4 ± 2.9 | 59.4 ± 1.6 | 64.2 ± 2.1 | 59.4 ± 1.7 | 62.4 ± 1.7 | 60.4 ± 1.7 | 60.5 ± 1.7 | 60.5 ± 1.7 | 60.4 ± 1.7 |
| Covert | 15.9 ± 2 | 21.7 ± 1.8 | 33.4 ± 4.3 | 15.3 ± 2.2 | 17.3 ± 2.4 | 14.3 ± 1.8 | 19.8 ± 4.4 | 14.9 ± 2.1 | 15.2 ± 2.2 | 15.3 ± 2.3 | 15.3 ± 2.3 |
| Foodweb | 46.4 ± 5 | 48.8 ± 4.9 | 89.8 ± 7.6 | 44.8 ± 5.2 | 62 ± 7.2 | 45.2 ± 5.2 | 43.9 ± 4.9 | 44 ± 5.1 | 44.2 ± 5.1 | 43.6 ± 5.1 | 44.1 ± 5.2 |
| Infrastructure | 32.1 ± 4.3 | 40.6 ± 4.8 | 58.1 ± 6.7 | 27.3 ± 5.1 | 35.5 ± 7.7 | 22.7 ± 3.7 | 27.7 ± 4.8 | 28.7 ± 5.3 | 28.1 ± 4.9 | 28.4 ± 5 | 28 ± 4.9 |
| Internet | 89.6 ± 3.3 | 88.9 ± 3.9 | 115.9 ± 7.2 | 96 ± 3.8 | 144.2 ± 6.3 | 90.8 ± 3.4 | 100.7 ± 4 | 95.7 ± 3.8 | 97.8 ± 3.9 | 100.3 ± 4.1 | 102 ± 4.2 |
| Misc | 324.9 ± 98.6 | 317.2 ± 95.9 | 377.3 ± 100.8 | 330.3 ± 98.8 | 409.2 ± 122.5 | 335.4 ± 103.2 | 339.5 ± 99.7 | 333.2 ± 99.6 | 338.2 ± 100.8 | 340.8 ± 101 | 347.7 ± 103.4 |
| Social | 431.4 ± 64.7 | 427 ± 63.5 | 543.7 ± 80 | 423.5 ± 64.2 | 481.2 ± 72.4 | 439.4 ± 68 | 449.5 ± 65.5 | 434.1 ± 64.8 | 436.4 ± 65 | 435 ± 64.9 | 437.5 ± 65.3 |

| Removals | EI $\sigma_1$ | EI $\sigma_2$ | Eigenvector | Fitness | GDM | GND | MS | NBC | PR | RA2 | RA2$_{num}$ | Resilience |
|---|---|---|---|---|---|---|---|---|---|---|---|---|
| Biomolecular | 216.5 ± 65.3 | 1766.3 ± 298.2 | 269.3 ± 72.5 | 298.6 ± 78.1 | 284.3 ± 80.1 | 346.1 ± 113.7 | 269.5 ± 70.5 | 214.3 ± 65.7 | 292.7 ± 78 | 251.6 ± 70.3 | 255.9 ± 71.3 | 264.9 ± 72.8 |
| Brain | 57.4 ± 1.5 | 85.1 ± 3.9 | 61.2 ± 1.8 | 60.8 ± 1.8 | 59.4 ± 1.7 | 65.1 ± 2 | 61.8 ± 1.7 | 55.9 ± 1.4 | 61.4 ± 1.8 | 56.7 ± 1.6 | 56.6 ± 1.7 | 60.5 ± 1.7 |
| Covert | 13.9 ± 1.3 | 82.2 ± 36.6 | 16 ± 2.4 | 15.4 ± 2.3 | 13.9 ± 1.9 | 15.9 ± 1.8 | 20 ± 4.4 | 13.4 ± 1.2 | 15.9 ± 2.3 | 14.2 ± 2 | 13.8 ± 2 | 14.8 ± 2.1 |
| Foodweb | 43.7 ± 4.6 | 84.7 ± 9.4 | 46.3 ± 5 | 43.9 ± 5.2 | 45.2 ± 5.3 | 61.3 ± 7.7 | 44 ± 4.9 | 41.6 ± 4.6 | 44.4 ± 5.2 | 44.4 ± 5 | 43.3 ± 4.8 | 45.3 ± 5 |
| Infrastructure | 18 ± 2.3 | 141.5 ± 26.3 | 29.3 ± 4.3 | 28.5 ± 5 | 23.2 ± 3.9 | 20.9 ± 2.7 | 27.1 ± 4.7 | 15.3 ± 2 | 31.4 ± 5.7 | 23.5 ± 4 | 26.2 ± 4.3 | |
| Internet | 78.2 ± 2.7 | 1095 ± 94.2 | 92 ± 3.4 | 104.1 ± 4.3 | 93.8 ± 3.7 | 118.5 ± 3.8 | 100.3 ± 4 | 74.3 ± 2.3 | 101.2 ± 4 | 89.4 ± 3.3 | 90.2 ± 3.3 | 92.6 ± 3.5 |
| Misc | 281.1 ± 87.3 | 934.8 ± 228 | 330.9 ± 99.3 | 348 ± 103.4 | 343.5 ± 105.5 | 383.8 ± 126.7 | 337 ± 98.9 | 263.6 ± 82.1 | 347.8 ± 102.3 | 312 ± 95.9 | 317.9 ± 96.7 | 325.1 ± 98.4 |
| Social | 389.8 ± 61 | 1049.7 ± 104.2 | 439.4 ± 65 | 441.4 ± 65.6 | 449.5 ± 69.2 | 476.3 ± 73.8 | 446 ± 64.8 | 377.2 ± 59.4 | 440 ± 65.8 | 413.7 ± 63.4 | 415.4 ± 63.8 | 433.6 ± 64.8 |

Table 25: Ranking per field for selected dismantling method ($n = 1{,}296$). In bold the best method per field.

| Ranking | NBC | CND | RA2 | RA2$_{num}$ | GDM | CoreGDM-NR | GND | Degree | Domirank $\sigma_{0.5}$ | PR | Fitness | Resilience | $CI_{L1}$ | EI $\sigma_1$ | Eigenvector | MS | CoreHD-NR |
|---|---|---|---|---|---|---|---|---|---|---|---|---|---|---|---|---|---|
| Biomolecular | 1 | 3 | 2 | 4 | 6 | 7 | 5 | 9 | 10 | 12 | 14 | 11 | 13 | 8 | 15 | 17 | 16 |
| Brain | 1 | 3 | 2 | 5 | 6 | 7 | 4 | 12 | 9 | 11 | 10 | 13 | 14 | 8 | 15 | 16 | 17 |
| Covert | 1 | 6 | 3 | 2 | 4 | 5 | 10 | 7 | 9 | 12 | 8 | 11 | 14 | 17 | 13 | 15 | 16 |
| Foodweb | 1 | 4 | 10 | 9 | 6 | 7 | 17 | 8 | 5 | 3 | 2 | 11 | 12 | 16 | 14 | 13 | 15 |
| Infrastructure | 1 | 6 | 4 | 5 | 2 | 3 | 14 | 7 | 8 | 10 | 9 | 11 | 16 | 17 | 15 | 12 | 13 |
| Internet | 1 | 2 | 3 | 4 | 5 | 6 | 15 | 8 | 11 | 7 | 12 | 9 | 10 | 14 | 13 | 16 | 17 |
| Misc | 1 | 2 | 3 | 6 | 5 | 7 | 4 | 9 | 10 | 8 | 13 | 11 | 12 | 15 | 14 | 16 | 17 |
| Social | 1 | 3 | 5 | 4 | 6 | 7 | 2 | 11 | 9 | 10 | 8 | 12 | 13 | 14 | 15 | 16 | 17 |
| *Mean ± SEM* | *1 ± 0* | *3.625 ± 0.57* | *4 ± 0.93* | *4.875 ± 0.72* | *5 ± 0.50* | *6.125 ± 0.52* | *8.875 ± 2.07* | *8.875 ± 0.64* | *8.875 ± 0.64* | *9.125 ± 1.08* | *9.5 ± 1.34* | *11.125 ± 0.40* | *13 ± 0.63* | *13.625 ± 1.30* | *14.25 ± 0.31* | *15.125 ± 0.61* | *16 ± 0.50* |

Table 26: Mean and standard error of the mean (SEM) for the AUC per field, by dismantling method, for reinsertion method R1, ($n = 1{,}237$). In bold the best method per field.

| AUC | $CI_{L1}$-R1 | CND-R1 | CoreGDM-R1 | CoreHD-R1 | Domirank $\sigma_{0.5}$-R1 | Fitness-R1 | GDM-R1 | MS-R1 | NBC-R1 |
|---|---|---|---|---|---|---|---|---|---|
| Biomolecular | 0.0599 ± 0.022 | 0.0564 ± 0.021 | 0.0606 ± 0.023 | 0.0632 ± 0.021 | 0.0615 ± 0.024 | 0.0606 ± 0.023 | 0.0585 ± 0.022 | 0.0696 ± 0.025 | **0.049 ± 0.018** |
| Brain | 0.4051 ± 0.002 | 0.381 ± 0.002 | 0.3984 ± 0.002 | 0.4045 ± 0.002 | 0.4035 ± 0.002 | 0.4032 ± 0.002 | 0.3988 ± 0.002 | 0.4069 ± 0.002 | **0.3679 ± 0.002** |
| Covert | 0.127 ± 0.009 | 0.1158 ± 0.009 | 0.1168 ± 0.009 | 0.1501 ± 0.01 | 0.1225 ± 0.009 | 0.1219 ± 0.009 | 0.1134 ± 0.009 | 0.1476 ± 0.01 | **0.1048 ± 0.009** |
| Foodweb | 0.227 ± 0.01 | 0.2103 ± 0.009 | 0.2135 ± 0.01 | 0.2371 ± 0.01 | 0.2122 ± 0.009 | 0.2087 ± 0.009 | 0.2124 ± 0.01 | 0.2311 ± 0.009 | **0.1968 ± 0.009** |
| Infrastructure | 0.0577 ± 0.004 | 0.0449 ± 0.003 | 0.0446 ± 0.003 | 0.0521 ± 0.003 | 0.0459 ± 0.003 | 0.0457 ± 0.003 | 0.0436 ± 0.003 | 0.0508 ± 0.003 | **0.041 ± 0.003** |
| Internet | 0.0601 ± 0.003 | 0.0567 ± 0.003 | 0.059 ± 0.003 | 0.0682 ± 0.003 | 0.0599 ± 0.003 | 0.0596 ± 0.003 | 0.0587 ± 0.003 | 0.0673 ± 0.003 | **0.0532 ± 0.003** |
| Misc | 0.1948 ± 0.026 | 0.1863 ± 0.025 | 0.1889 ± 0.026 | 0.2033 ± 0.025 | 0.1929 ± 0.025 | 0.1936 ± 0.025 | 0.1895 ± 0.026 | 0.1996 ± 0.025 | **0.1732 ± 0.025** |
| Social | 0.2137 ± 0.015 | 0.1975 ± 0.014 | 0.2061 ± 0.015 | 0.2228 ± 0.015 | 0.2097 ± 0.015 | 0.2091 ± 0.015 | 0.2052 ± 0.015 | 0.2234 ± 0.015 | **0.1883 ± 0.014** |

Table 27: Mean and standard error of the mean (SEM) for the number of removals per field, by dismantling method, for reinsertion method R1, ($n = 1{,}237$). In bold the best method per field.

| Removals | $CI_{L1}$-R1 | CND-R1 | CoreGDM-R1 | CoreHD-R1 | Domirank $\sigma_{0.5}$-R1 | Fitness-R1 | GDM-R1 | MS-R1 | NBC-R1 |
|---|---|---|---|---|---|---|---|---|---|
| Biomolecular | 70.1 ± 11.8 | 71.1 ± 12.3 | 71.5 ± 11.7 | 69.2 ± 11.3 | 72.6 ± 12.7 | 73.6 ± 12.9 | 71.9 ± 12.1 | 71.1 ± 11.3 | **65.4 ± 10.6** |
| Brain | 54.9 ± 0.9 | 53.3 ± 0.9 | 54.2 ± 0.9 | 54.9 ± 0.9 | 54.8 ± 0.9 | 54.8 ± 0.9 | 54.4 ± 0.9 | 55.2 ± 0.9 | **52.8 ± 0.8** |
| Covert | 13 ± 1.5 | 12.6 ± 1.5 | 12.7 ± 1.5 | 13.8 ± 1.9 | 12.8 ± 1.5 | 12.8 ± 1.6 | 12.6 ± 1.4 | 13.9 ± 1.9 | **12 ± 1.2** |
| Foodweb | 41.8 ± 4.5 | 41.2 ± 4.6 | 41.7 ± 4.6 | 41 ± 4.6 | 41 ± 4.7 | 41 ± 4.7 | 41.7 ± 4.6 | 40.9 ± 4.6 | **39.8 ± 4.4** |
| Infrastructure | 14.9 ± 1.9 | 13.4 ± 1.8 | 13.3 ± 1.9 | 13.7 ± 2 | 13.9 ± 2 | 13.7 ± 1.9 | 13.6 ± 1.9 | 13.8 ± 2 | **12.3 ± 1.6** |
| Internet | 75.4 ± 2.8 | 77.1 ± 3.1 | 76.6 ± 3 | 75.6 ± 2.7 | 79.6 ± 3.3 | 80.1 ± 3.4 | 78.2 ± 3.2 | 75.3 ± 2.7 | **70 ± 2.4** |
| Misc | 98.4 ± 22.7 | 98 ± 23 | 99.5 ± 23.3 | 99.8 ± 23.4 | 99.9 ± 23.4 | 101.3 ± 23.7 | 99.9 ± 23.6 | 99.2 ± 23.1 | **92.3 ± 21.8** |
| Social | 62.4 ± 8.3 | 61.3 ± 8.1 | 62.2 ± 8.3 | 62.5 ± 8.3 | 63.1 ± 8.3 | 63.3 ± 8.4 | 63.8 ± 8.5 | 62.7 ± 8.4 | **58 ± 7.7** |

Table 28: Ranking per field for selected dismantling method, for reinsertion method R1, ($n = 1{,}237$). In bold the best method per field.

| Ranking | NBC-R1 | CND-R1 | GDM-R1 | CoreGDM-R1 | Fitness-R1 | Domirank $\sigma_{0.5}$-R1 | $CI_{L1}$-R1 | MS-R1 | CoreHD-R1 |
|---|---|---|---|---|---|---|---|---|---|
| Biomolecular | 1 | 2 | 3 | 5 | 6 | 7 | 4 | 9 | 8 |
| Brain | 1 | 2 | 4 | 3 | 5 | 6 | 8 | 9 | 7 |
| Covert | 1 | 3 | 2 | 4 | 5 | 6 | 7 | 8 | 9 |
| Foodweb | 1 | 3 | 5 | 6 | 2 | 4 | 7 | 8 | 9 |
| Infrastructure | 1 | 4 | 2 | 3 | 5 | 6 | 9 | 7 | 8 |
| Internet | 1 | 2 | 3 | 4 | 5 | 6 | 7 | 8 | 9 |
| Misc | 1 | 2 | 4 | 3 | 6 | 5 | 7 | 8 | 9 |
| Social | 1 | 2 | 3 | 4 | 5 | 6 | 7 | 9 | 8 |
| *Mean ± SEM* | *1 ± 0* | *2.5 ± 0.27* | *3.25 ± 0.37* | *4 ± 0.38* | *4.875 ± 0.44* | *5.75 ± 0.31* | *7 ± 0.50* | *8.25 ± 0.25* | *8.375 ± 0.26* |

Table 29: Mean and standard error of the mean (SEM) for the AUC per field, by dismantling method, for reinsertion method R2, ($n = 1{,}237$). In bold the best method per field.

| AUC | Cl$_{L1}$-R2 | CND-R2 | CoreGDM-R2 | CoreHD-R2 | Domirank $\sigma_{0.5}$-R2 | Fitness-R2 | GDM-R2 | MS-R2 | NBC-R2 |
|---|---|---|---|---|---|---|---|---|---|
| Biomolecular | 0.062 ± 0.024 | 0.0572 ± 0.021 | 0.0613 ± 0.023 | 0.0664 ± 0.022 | 0.0626 ± 0.024 | 0.064 ± 0.025 | 0.0612 ± 0.023 | 0.0725 ± 0.026 | **0.0485 ± 0.017** |
| Brain | 0.406 ± 0.002 | 0.3805 ± 0.002 | 0.3991 ± 0.002 | 0.4084 ± 0.002 | 0.4045 ± 0.002 | 0.4043 ± 0.002 | 0.3996 ± 0.002 | 0.4097 ± 0.002 | **0.3659 ± 0.002** |
| Covert | 0.1272 ± 0.009 | 0.1149 ± 0.009 | 0.1167 ± 0.009 | 0.1514 ± 0.01 | 0.1226 ± 0.009 | 0.1217 ± 0.009 | 0.1131 ± 0.009 | 0.1484 ± 0.01 | **0.1036 ± 0.009** |
| Foodweb | 0.2273 ± 0.01 | 0.2108 ± 0.009 | 0.2138 ± 0.01 | 0.2377 ± 0.01 | 0.2124 ± 0.009 | 0.2089 ± 0.009 | 0.213 ± 0.01 | 0.2319 ± 0.009 | **0.1957 ± 0.009** |
| Infrastructure | 0.0582 ± 0.004 | 0.0451 ± 0.003 | 0.0449 ± 0.003 | 0.0529 ± 0.003 | 0.0463 ± 0.003 | 0.0462 ± 0.003 | 0.0438 ± 0.003 | 0.0514 ± 0.003 | **0.0409 ± 0.003** |
| Internet | 0.0607 ± 0.003 | 0.0571 ± 0.003 | 0.0596 ± 0.003 | 0.0697 ± 0.003 | 0.0605 ± 0.003 | 0.0603 ± 0.003 | 0.0593 ± 0.003 | 0.0687 ± 0.003 | **0.0529 ± 0.003** |
| Misc | 0.1961 ± 0.026 | 0.187 ± 0.025 | 0.1902 ± 0.026 | 0.2054 ± 0.025 | 0.1946 ± 0.025 | 0.1954 ± 0.025 | 0.1908 ± 0.026 | 0.2032 ± 0.025 | **0.1711 ± 0.025** |
| Social | 0.2148 ± 0.015 | 0.1977 ± 0.014 | 0.207 ± 0.015 | 0.227 ± 0.015 | 0.2113 ± 0.015 | 0.2107 ± 0.015 | 0.2065 ± 0.015 | 0.2275 ± 0.015 | **0.1852 ± 0.014** |

Table 30: Mean and standard error of the mean (SEM) for the number of removals per field, by dismantling method, for reinsertion method R2, ($n = 1{,}237$). In bold the best method per field.

| Removals | Cl$_{L1}$-R2 | CND-R2 | CoreGDM-R2 | CoreHD-R2 | Domirank $\sigma_{0.5}$-R2 | Fitness-R2 | GDM-R2 | MS-R2 | NBC-R2 |
|---|---|---|---|---|---|---|---|---|---|
| Biomolecular | 71.5 ± 11.9 | 73.9 ± 13.2 | 73.4 ± 12.4 | 74.1 ± 12.2 | 76.2 ± 13.6 | 78.7 ± 14.1 | 76.9 ± 13.9 | 75.6 ± 12.3 | **65.3 ± 10.6** |
| Brain | 55.1 ± 0.9 | 53.4 ± 0.9 | 54.4 ± 0.9 | 55.7 ± 0.9 | 55.1 ± 0.9 | 55.1 ± 0.9 | 54.7 ± 1 | 55.8 ± 0.9 | **52.8 ± 0.8** |
| Covert | 13 ± 1.6 | 12.7 ± 1.5 | 12.8 ± 1.5 | 14.2 ± 2.1 | 12.8 ± 1.6 | 12.9 ± 1.7 | 12.8 ± 1.6 | 14.3 ± 2.2 | **11.9 ± 1.2** |
| Foodweb | 42.1 ± 4.7 | 41.2 ± 4.7 | 42.1 ± 4.8 | 41.2 ± 4.6 | 41.1 ± 4.7 | 41.2 ± 4.8 | 42.1 ± 4.8 | 41 ± 4.6 | **39.9 ± 4.4** |
| Infrastructure | 15.8 ± 2.1 | 13.9 ± 1.9 | 14 ± 2 | 14.5 ± 2.2 | 14.5 ± 2.1 | 14.6 ± 2.2 | 14 ± 2.1 | 14.6 ± 2.2 | **12.3 ± 1.6** |
| Internet | 76.7 ± 2.9 | 78.6 ± 3.2 | 78 ± 3.1 | 77.9 ± 3 | 81.5 ± 3.4 | 82.6 ± 3.5 | 79.9 ± 3.3 | 77.5 ± 3 | **70.1 ± 2.4** |
| Misc | 101.9 ± 23.9 | 99.8 ± 23.5 | 102.3 ± 24.2 | 104.2 ± 24.5 | 102.8 ± 24.1 | 106 ± 24.8 | 103.1 ± 24.5 | 104 ± 24.3 | **92.8 ± 21.9** |
| Social | 64 ± 8.6 | 62.9 ± 8.4 | 64.4 ± 8.7 | 65.8 ± 8.7 | 65.7 ± 8.7 | 65.9 ± 8.7 | 65.8 ± 8.9 | 66.1 ± 8.7 | **57.6 ± 7.7** |

Table 31: Ranking per field for selected dismantling method, for reinsertion method R2, ($n = 1{,}237$). In bold the best method per field.

| Ranking | NBC-R2 | CND-R2 | GDM-R2 | CoreGDM-R2 | Fitness-R2 | Domirank $\sigma_{0.5}$-R2 | Cl$_{L1}$-R2 | MS-R2 | CoreHD-R2 |
|---|---|---|---|---|---|---|---|---|---|
| Biomolecular | 1 | 2 | 3 | 4 | 7 | 6 | 5 | 9 | 8 |
| Brain | 1 | 2 | 4 | 3 | 5 | 6 | 7 | 9 | 8 |
| Covert | 1 | 3 | 2 | 4 | 5 | 6 | 7 | 8 | 9 |
| Foodweb | 1 | 3 | 5 | 6 | 2 | 4 | 7 | 8 | 9 |
| Infrastructure | 1 | 4 | 2 | 3 | 5 | 6 | 9 | 7 | 8 |
| Internet | 1 | 2 | 3 | 4 | 5 | 6 | 7 | 8 | 9 |
| Misc | 1 | 2 | 4 | 3 | 6 | 5 | 7 | 8 | 9 |
| Social | 1 | 2 | 3 | 4 | 5 | 6 | 7 | 9 | 8 |
| *Mean ± SEM* | *1 ± 0* | *2.5 ± 0.27* | *3.25 ± 0.37* | *3.875 ± 0.35* | *5 ± 0.50* | *5.625 ± 0.26* | *7 ± 0.38* | *8.25 ± 0.25* | *8.5 ± 0.19* |

Table 32: Mean and standard error of the mean (SEM) for the AUC per field, by dismantling method, for reinsertion method R3, ($n = 1{,}237$). In bold the best method per field.

| AUC | Cl$_{L1}$-R3 | CND-R3 | CoreGDM-R3 | CoreHD-R3 | Domirank $\sigma_{0.5}$-R3 | Fitness-R3 | GDM-R3 | MS-R3 | NBC-R3 |
|---|---|---|---|---|---|---|---|---|---|
| Biomolecular | 0.0603 ± 0.022 | 0.0565 ± 0.021 | 0.0611 ± 0.023 | 0.063 ± 0.021 | 0.0617 ± 0.024 | 0.0619 ± 0.024 | 0.0594 ± 0.022 | 0.0702 ± 0.025 | **0.0484 ± 0.017** |
| Brain | 0.4052 ± 0.002 | 0.3804 ± 0.002 | 0.3986 ± 0.002 | 0.405 ± 0.002 | 0.4037 ± 0.002 | 0.4033 ± 0.002 | 0.3992 ± 0.002 | 0.4071 ± 0.002 | **0.3662 ± 0.002** |
| Covert | 0.127 ± 0.009 | 0.115 ± 0.009 | 0.1169 ± 0.009 | 0.1503 ± 0.01 | 0.1225 ± 0.009 | 0.1219 ± 0.009 | 0.1131 ± 0.009 | 0.148 ± 0.01 | **0.1038 ± 0.009** |
| Foodweb | 0.2273 ± 0.01 | 0.2107 ± 0.009 | 0.2136 ± 0.01 | 0.2371 ± 0.01 | 0.2122 ± 0.009 | 0.2088 ± 0.009 | 0.2126 ± 0.01 | 0.2311 ± 0.009 | **0.1963 ± 0.009** |
| Infrastructure | 0.0582 ± 0.004 | 0.0448 ± 0.003 | 0.0447 ± 0.003 | 0.0522 ± 0.003 | 0.046 ± 0.003 | 0.0458 ± 0.003 | 0.0436 ± 0.003 | 0.0509 ± 0.003 | **0.0409 ± 0.003** |
| Internet | 0.0602 ± 0.003 | 0.0569 ± 0.003 | 0.0591 ± 0.003 | 0.0684 ± 0.003 | 0.06 ± 0.003 | 0.0599 ± 0.003 | 0.059 ± 0.003 | 0.0675 ± 0.003 | **0.053 ± 0.003** |
| Misc | 0.1946 ± 0.026 | 0.1866 ± 0.025 | 0.1892 ± 0.026 | 0.2038 ± 0.025 | 0.1935 ± 0.025 | 0.1945 ± 0.025 | 0.1897 ± 0.026 | 0.2 ± 0.025 | **0.1711 ± 0.025** |
| Social | 0.214 ± 0.015 | 0.1975 ± 0.014 | 0.2063 ± 0.015 | 0.2245 ± 0.015 | 0.2101 ± 0.015 | 0.2094 ± 0.015 | 0.2059 ± 0.015 | 0.2253 ± 0.015 | **0.1857 ± 0.014** |

Table 33: Mean and standard error of the mean (SEM) for the number of removals per field, by dismantling method, for reinsertion method R3, ($n = 1{,}237$). In bold the best method per field.

| Removals | Cl$_{L1}$-R3 | CND-R3 | CoreGDM-R3 | CoreHD-R3 | Domirank $\sigma_{0.5}$-R3 | Fitness-R3 | GDM-R3 | MS-R3 | NBC-R3 |
|---|---|---|---|---|---|---|---|---|---|
| Biomolecular | 70.6 ± 11.8 | 72.8 ± 13 | 72.5 ± 11.8 | 70.2 ± 11.6 | 73.4 ± 12.8 | 76.2 ± 13.6 | 73.4 ± 12.5 | 71.6 ± 11.4 | **65.3 ± 10.6** |
| Brain | 54.9 ± 0.9 | 53.4 ± 0.9 | 54.2 ± 0.9 | 55 ± 0.9 | 54.9 ± 0.9 | 54.9 ± 0.9 | 54.6 ± 1 | 55.2 ± 0.9 | **52.8 ± 0.8** |
| Covert | 13 ± 1.5 | 12.6 ± 1.5 | 12.8 ± 1.5 | 13.9 ± 2 | 12.8 ± 1.5 | 13 ± 1.7 | 12.6 ± 1.4 | 14 ± 1.9 | **11.9 ± 1.2** |
| Foodweb | 42.1 ± 4.6 | 41.2 ± 4.7 | 41.7 ± 4.6 | 41 ± 4.6 | 41.1 ± 4.7 | 41.2 ± 4.7 | 41.9 ± 4.7 | 40.9 ± 4.6 | **39.8 ± 4.4** |
| Infrastructure | 15.4 ± 2 | 13.4 ± 1.8 | 13.6 ± 2 | 13.6 ± 2 | 14.2 ± 2 | 14 ± 2 | 13.7 ± 2 | 14 ± 2.1 | **12.3 ± 1.6** |
| Internet | 75.6 ± 2.8 | 77.8 ± 3.2 | 77.2 ± 3 | 76 ± 2.7 | 80.2 ± 3.3 | 81.1 ± 3.4 | 79 ± 3.2 | 75.4 ± 2.7 | **70.1 ± 2.4** |
| Misc | 98.8 ± 23.1 | 98.7 ± 23 | 100.1 ± 23.6 | 100.8 ± 23.3 | 100.6 ± 23.4 | 104.3 ± 24.5 | 100.9 ± 23.8 | 99.6 ± 23.2 | **92.6 ± 21.8** |
| Social | 62.7 ± 8.3 | 61.8 ± 8.2 | 63 ± 8.5 | 63.8 ± 8.4 | 64 ± 8.5 | 64.3 ± 8.6 | 65 ± 8.7 | 63.8 ± 8.4 | **57.5 ± 7.6** |

Table 34: Ranking per field for selected dismantling method, for reinsertion method R3, ($n = 1{,}237$). In bold the best method per field.

| Ranking | NBC-R3 | CND-R3 | GDM-R3 | CoreGDM-R3 | Fitness-R3 | Domirank $\sigma_{0.5}$-R3 | $Cl_{L1}$-R3 | MS-R3 | CoreHD-R3 |
|---|---|---|---|---|---|---|---|---|---|
| Biomolecular | **1** | 2 | 3 | 5 | 7 | 6 | 4 | 9 | 8 |
| Brain | **1** | 2 | 4 | 3 | 5 | 6 | 8 | 9 | 7 |
| Covert | **1** | 3 | 2 | 4 | 5 | 6 | 7 | 8 | 9 |
| Foodweb | **1** | 3 | 5 | 6 | 2 | 4 | 7 | 8 | 9 |
| Infrastructure | **1** | 4 | 2 | 3 | 5 | 6 | 9 | 7 | 8 |
| Internet | **1** | 2 | 3 | 4 | 5 | 6 | 7 | 8 | 9 |
| Misc | **1** | 2 | 4 | 3 | 6 | 5 | 7 | 8 | 9 |
| Social | **1** | 2 | 3 | 4 | 5 | 6 | 7 | 9 | 8 |
| *Mean ± SEM* | *1 ± 0* | *2.5 ± 0.27* | *3.25 ± 0.37* | *4 ± 0.38* | *5 ± 0.50* | *5.625 ± 0.26* | *7 ± 0.50* | *8.25 ± 0.25* | *8.375 ± 0.26* |

Table 35: Ranking per field for selected dismantling method with their best-performing reinsertion method ($n = 1{,}237$). In bold the best method per field.

| Ranking | NBC-R2 | CND-R3 | GDM-R1 | CoreGDM-R1 | Fitness-R1 | Domirank $\sigma_{0.5}$-R1 | $Cl_{L1}$-R1 | MS-R1 | CoreHD-R1 |
|---|---|---|---|---|---|---|---|---|---|
| Biomolecular | **1** | 2 | 3 | 5 | 6 | 7 | 4 | 9 | 8 |
| Brain | **1** | 2 | 4 | 3 | 5 | 6 | 8 | 9 | 7 |
| Covert | **1** | 3 | 2 | 4 | 5 | 6 | 7 | 8 | 9 |
| Foodweb | **1** | 3 | 5 | 6 | 2 | 4 | 7 | 8 | 9 |
| Infrastructure | **1** | 4 | 2 | 3 | 5 | 6 | 9 | 7 | 8 |
| Internet | **1** | 2 | 3 | 4 | 5 | 6 | 7 | 8 | 9 |
| Misc | **1** | 2 | 4 | 3 | 6 | 5 | 7 | 8 | 9 |
| Social | **1** | 2 | 3 | 4 | 5 | 6 | 7 | 9 | 8 |
| *Mean ± SEM* | *1 ± 0* | *2.5 ± 0.27* | *3.25 ± 0.37* | *4 ± 0.38* | *4.875 ± 0.44* | *5.75 ± 0.31* | *7 ± 0.50* | *8.25 ± 0.25* | *8.375 ± 0.26* |

Table 36: Average AUC by field for top two performing methods: NBC and CND, under different reinsertion methods ($n = 1{,}237$) and without reinsertion ($n = 1{,}296$). In bold the best method per field and reinsertion method.

| AUC | Baseline | | R1 | | R2 | | R3 | |
|---|---|---|---|---|---|---|---|---|
| | CND | NBC | CND-R1 | NBC-R1 | CND-R2 | NBC-R2 | CND-R3 | NBC-R3 |
| **Biomolecular** | 0.072 | 0.058 | 0.056 | 0.049 | 0.057 | **0.048** | 0.057 | **0.048** |
| **Brain** | 0.388 | **0.362** | 0.381 | 0.368 | 0.380 | 0.366 | 0.380 | 0.366 |
| **Covert** | 0.122 | 0.110 | 0.116 | 0.105 | 0.115 | **0.104** | 0.115 | **0.104** |
| **Foodweb** | 0.215 | **0.195** | 0.210 | 0.197 | 0.211 | 0.196 | 0.211 | 0.196 |
| **Infrastructure** | 0.048 | **0.040** | 0.045 | 0.041 | 0.045 | 0.041 | 0.045 | 0.041 |
| **Internet** | 0.058 | **0.049** | 0.057 | 0.053 | 0.057 | 0.053 | 0.057 | 0.053 |
| **Misc** | 0.204 | **0.171** | 0.186 | 0.173 | 0.187 | **0.171** | 0.187 | **0.171** |
| **Social** | 0.232 | 0.204 | 0.198 | 0.188 | 0.198 | **0.185** | 0.198 | 0.186 |

Table 37: Running time (in seconds) comparison between CND on GPU and NBC on CPU on synthetic nPSO networks. Experiments were conducted with network sizes ranging from 10 to 5,000 nodes and densities ($\rho$) of 4%, 8%, and 20%, using fixed temperature $T = 0.3$ and community counts scaled by network size ($C = 2$ for $N \in \{10, 50, 100\}$, $C = 5$ for $N \in \{500, 1,000\}$, $C = 10$ for $N = 5,000$). In bold the fastest method per network. $N$ denotes number of nodes and $E$ number of edges.

| γ | C | N | E | ρ | CND-GPU | NBC-CPU |
|---|---|---|---|---|---|---|
| 2 | 2 | 10 | 10 | 0.22 | 0.020 | **0.011** |
| 2 | 2 | 10 | 20 | 0.44 | 0.008 | **0.003** |
| 2 | 2 | 10 | 30 | 0.67 | 0.007 | **0.004** |
| 2 | 2 | 50 | 49 | 0.04 | 0.19 | **0.13** |
| 2 | 2 | 50 | 98 | 0.08 | 0.37 | **0.20** |
| 2 | 2 | 50 | 245 | 0.20 | 0.80 | **0.13** |
| 2 | 2 | 100 | 198 | 0.04 | 0.50 | **0.16** |
| 2 | 2 | 100 | 396 | 0.08 | 1.23 | **0.23** |
| 2 | 2 | 100 | 990 | 0.20 | 3.10 | **0.52** |
| 2 | 5 | 500 | 4,990 | 0.04 | 8.78 | **3.26** |
| 2 | 5 | 500 | 9,980 | 0.08 | 11.62 | **4.75** |
| 2 | 5 | 500 | 24,950 | 0.20 | 15.76 | **12.83** |
| 2 | 5 | 1,000 | 19,980 | 0.04 | 23.93 | **9.53** |
| 2 | 5 | 1,000 | 39,960 | 0.08 | 33.47 | **10.30** |
| 2 | 5 | 1,000 | 99,900 | 0.20 | 37.50 | **30.22** |
| 2 | 10 | 5,000 | 499,900 | 0.04 | **390.67** | 1,304.82 |
| 2 | 10 | 5,000 | 999,800 | 0.08 | **674.40** | 5,860.19 |
| 2 | 10 | 5,000 | 2,499,500 | 0.20 | **1,448.55** | 9,632.88 |
| 3 | 2 | 10 | 10 | 0.22 | 0.003 | **0.002** |
| 3 | 2 | 10 | 20 | 0.44 | 0.004 | **0.002** |
| 3 | 2 | 10 | 30 | 0.67 | 0.005 | **0.002** |
| 3 | 2 | 50 | 49 | 0.04 | 0.23 | **0.07** |
| 3 | 2 | 50 | 98 | 0.08 | 0.67 | **0.13** |
| 3 | 2 | 50 | 245 | 0.20 | 1.13 | **0.16** |
| 3 | 2 | 100 | 198 | 0.04 | 1.12 | **0.11** |
| 3 | 2 | 100 | 396 | 0.08 | 1.81 | **0.22** |
| 3 | 2 | 100 | 990 | 0.20 | 2.77 | **0.40** |
| 3 | 5 | 500 | 4,990 | 0.04 | 12.97 | **4.64** |
| 3 | 5 | 500 | 9,980 | 0.08 | 17.18 | **6.69** |
| 3 | 5 | 500 | 24,950 | 0.20 | 21.37 | **9.38** |
| 3 | 5 | 1,000 | 19,980 | 0.04 | 34.24 | **25.47** |
| 3 | 5 | 1,000 | 39,960 | 0.08 | 40.73 | **35.08** |
| 3 | 5 | 1,000 | 99,900 | 0.20 | **44.92** | 47.46 |
| 3 | 10 | 5,000 | 499,900 | 0.04 | **632.02** | 3,593.83 |
| 3 | 10 | 5,000 | 999,800 | 0.08 | **946.71** | 7,367.02 |
| 3 | 10 | 5,000 | 2,499,500 | 0.20 | **2,090.94** | 10,122.08 |

