# OpenReview forum: "Latent Geometry-Driven Network Automata for Complex Network Dismantling"
_ICLR.cc/2026/Conference — ICLR 2026 Poster_

### Official Review · Reviewer_MAH5 · 2025-10-20

**Soundness:** 3
**Presentation:** 3
**Contribution:** 3
**Rating:** 4
**Confidence:** 3

**Summary:**

The paper proposes Latent Geometry-Driven Network Automata (LGD-NA) for network dismantling—sequentially removing nodes to fragment a graph. The key idea is to estimate latent geometric distances on the graph using local, training-free automata rules, convert these to edge dissimilarities, sum per-node to obtain a geometric strength score, and perform dynamic dismantling (recompute scores after each removal). The study evaluates on an ATLAS of 1,475 real-world networks across 32 domains, by AUC of the LCC curve until 10% LCC, and reports top mean-field ranks for the latent-geometry family vs. centrality, message-passing and ML baselines. It also provides a GPU implementation and a robustness-engineering experiment.

**Strengths:**

1. Latent-geometry-driven dismantling, realized via local automata rules (RA2; CND ablation) that use only first-hop structure, is a clean and useful angle.
2. The paper is clearly written. Pipeline, metrics (AUC to 10% LCC), and reinsertion constraints are explicit.
3. The proposed work is evaluated on large-scale set, including 1,475 networks across 32 domains.

**Weaknesses:**

1. Paper alternates between “LGD-NA outperforms all” and “NBC achieves better dismantling but is slower.”
2. Results are for undirected, unweighted graphs; many domains are directed or weighted (transport flows, trade). Please consider adding experiments (or a clearly stated limitation) for those cases.
3. Missing tuning/compute budgets and accelerated-NBC baselines.
4. Reinsertion and robustness-engineering analyses need stronger controls.

**Questions:**

1. NBC vs LGD-NA performance kind of unclear. Please provide a table with absolute AUC and mean-field rank side-by-side for NBC, RA2, CND under identical dynamic/reinsertion settings, and list fields where one strictly dominates.
2. Please show results for at least one directed (e.g., trade) and one weighted (e.g., power grid loads) domain, in order to make the paper more convincing.
3. I am kind of interested in the sensitivity analysis of the proposed method. For 3–4 strong methods, please plot AUC with no reinsertion vs 3 reinsertion policies and report ΔAUC. So that we can confirm rankings are stable or not.

---

> ### Author Response · Authors · 2025-11-21
> **Answer to Reviewer MAH5 1/3**
>
> We thank the Reviewer for their insightful and positive feedback, and for highlighting the simplicity and strength of our LGD-NA framework.
>
> **1. Paper alternates between “LGD-NA outperforms all” and “NBC achieves better dismantling but is slower.”**
>
> We thank the reviewer for identifying the ambiguity in our wording regarding the top-performing algorithm. We intended to convey that LGD-NA outperforms all other dedicated dismantling algorithms (GDM, CoreGDM, GND, Collective Influence, Explosive Immunization, Min-Sum, CoreHD). As for centrality metrics used for dismantling, Node Betweenness Centrality (NBC) performs the best overall, with LGD-NA following closely in second place. The main limitation of using NBC in real-world scenarios is its inherent inefficiency (O(VE) time complexity and cannot be accelerated by GPUs), making our GPU-optimized LGD-NA framework the most practical solution for large-scale networks.
>
> We now resolve this ambiguity by revising Lines 408 and 508 to explicitly specify "dismantling algorithms." Furthermore, we refined the fourth and last paragraph of Subsection 4.4  to read as follows:
>
> *“LGD-NA consistently outperforms all other non-latent geometry-driven dismantling algorithms, including those relying on spectral Laplacian-based and machine learning. The only measure that still outperforms LGD-NA is the Node Betweenness Centrality (NBC) metric (which is also latent-geometry-driven), applied to dynamic dismantling. These results strongly demonstrate the practical reliability of our latent geometry-driven dismantling framework, LGD-NA.”*
>
>
> **2. Results are for undirected, unweighted graphs; many domains are directed or weighted (transport flows, trade). Please consider adding experiments (or a clearly stated limitation) for those cases.**
>
> We thank the Reviewer for this insightful suggestion. We agree that extending the analysis to directed and weighted graphs is a crucial consideration for specific domains like transport or trade networks. Following the Reviewer's recommendation, we have included a paragraph discussing the limitations in regard to directed and weighted networks in Appendix O.
>
> *"Weighted and Directed Networks: While many real-world systems, such as citation networks (directed), neural synaptic connections (weighted), or transportation and trade networks (directed and weighted), contain directed or weighted interactions, our LGD-NA measures focus on unweighted, undirected topologies for two key reasons. First, in practical dismantling contexts, weight data is often dynamic, temporal, or unavailable, whereas topology is more robust. Second, current state-of-the-art dismantling algorithms focus on undirected, unweighted graphs (Artime et al., 2024), making fair comparisons impossible. Consequently, we consider the extension of latent-geometry approaches to directed and weighted graphs as a promising avenue for future work."*
>
> **3. Missing tuning/compute budgets**
>
> If tuning budget refers to the cost of hyperparameter search, we wish to highlight that our LGD-NA model is parameter-free and requires no tuning, offering a significant advantage over ML-based baselines. We explicitly added this *"parameter-free"* distinction on Line 234 and Line 265.
>
> If the compute budget refers to running time, we agree that a direct comparison is necessary. We have added the exact runtimes in seconds for the CPU versions of NBC, RA2, and CND and the GPU versions of RA2 and CND, in Table 14 as well as hardware specifications in Appendix N.
>
> If we have misinterpreted the Reviewer's specific definition of tuning and compute budgets, we are happy to provide additional details upon further clarification.

---

> ### Author Response · Authors · 2025-11-21
> **Answer to Reviewer MAH5 2/3**
>
> **4. Missing accelerated-NBC baselines.**
>
> We acknowledge the reviewer’s expectation to compare against NBC approximators. As a result, we have conducted new experiments comparing our methods against a simple NBC approximation strategy. We have documented these findings in a new section, Appendix K: NBC Approximators, and added Table 16 to report the specific results.
>
> *“While the high computational cost of Node Betweenness Centrality (NBC) has motivated the development of numerous approximators (Bader et al., 2007; Bergamini & Meyerhenke, 2015; Haghir Chehreghani, 2013; Riondato & Kornaropoulos, 2014), comparing against them is challenging due to the scarcity of standardized, publicly available code and the complexity of their sampling algorithms, which are often performant only for specific domains or incompatible with disconnected graph structures.*
>
> *To address this, we implemented two standard randomized pivoting strategies for approximation. NBC-20 estimates betweenness centrality using a random sample of 20% of the nodes. NBC-log uses a random sample of 10 x log2(N) nodes. NBC-20 prioritizes accuracy by always scanning a fixed slice of the network (20%), whereas NBC-log prioritizes speed by scanning a much smaller, logarithmically scaled subset that grows very slowly as the network gets larger. We evaluated these baselines against the exact NBC and our CND method on a subset of 157 networks selected for their size, where exact NBC calculation begins to become computationally expensive.*
>
> *Table 16 reports the dismantling performance (AUC) and total runtime, averaged by field. First, we see that the NBC approximators perform comparably to the exact NBC, even occasionally outperforming it. However, this performance increase of NBC approximators should not be overstated due to the smaller sample size of this experiment. Second, while NBC approximators are significantly faster than exact NBC, CND remains faster than both approximation methods in almost all domains. A notable exception occurs in the Infrastructure field. Here, the usually slowest method NBC is actually the fastest in terms of total runtime because it takes a significantly lower number of removals to dismantle the network. Consequently, even though CND is faster per step, the NBC-based methods result in a lower total runtime simply because the dismantling threshold is reached much earlier.*
>
> *Finally, it is critical to distinguish the theoretical foundations of these approaches. Existing NBC approximators focus on accelerating the estimation of a global metric. In contrast, LGD-NA leverages purely local topological information to directly estimate pairwise distances in the latent metric space. This distinction allows LGD-NA to bypass the need for global knowledge or more complex sampling strategies. Although approximation techniques improve NBC's speed, their reliance on sampling global paths is inherently less efficient than our strictly local approach, as we validate in Table 16. Furthermore, sampling global information remains vulnerable to missing data and adversarial noise. The strength of LGD-NA, therefore, lies in its ability to achieve high dismantling performance by directly utilizing local geometric insights, rather than attempting to approximate a computationally intensive global metric.”*
>
> Regarding GPU-based NBC, we did not include a GPU baseline for the following technical and practical reasons, as mentioned in Subsection 4.5 and Appendix J:
>
> - We tested the GPU implementation of NBC provided by cuGraph and found it to be significantly slower than the highly optimized C++ implementation in graph-tool (CPU).
> - This makes sense because, unlike our LGD-NA method, which leverages matrix multiplication, an operation ideally suited for GPU.
> - NBC relies on counting all-pairs shortest paths. This operation is inherently sequential and cannot be parallelized effectively on GPUs.
>
> To clarify this for the reader, we have added the following paragraph to Subsection 4.5:
>
> *“We report only the CPU running time for NBC, as its GPU implementation did not yield any speedup (see Table 15). While some studies report GPU implementations of NBC with improved performance (Fan et al., 2017; Shi & Zhang, 2011; Pande & Bader, 2011; McLaughlin & Bader, 2018; Sariyuce et al., 2013; Bernaschi et al., 2016), these are often limited by hardware-specific optimizations, data-specific assumptions (e.g., small-world, social, or biological networks), and the use of heuristics that are tailored to specific settings rather than offering general solutions. Moreover, publicly available code is rare, making these approaches difficult to reproduce or integrate. Overall, NBC is not naturally suited for GPU implementation, as it does not rely on matrix multiplication, but is based on computing shortest path counts between all node pairs.”*

---

> ### Author Response · Authors · 2025-11-21
> **Answer to Reviewer MAH5 3/3**
>
> For the Reviewer's interest, we also added the dismantling time for selected networks for NBC on CPU and GPU (hardware specifications can be found in Appendix N), in Table 15.
>
> As mentioned previously, we have also added the exact runtimes for the CPU versions of NBC, RA2, and CND and the GPU versions of RA2 and CND, in Table 14.
>
>
> **5. Reinsertion analyses need stronger controls.**
>
> We wish to clarify the extent of the controls implemented in our reinsertion experiments. We believe our experimental setup is more rigorous than what has been done in previous dismantling studies. To make this explicit, we have added the following paragraph to Appendix F:
>
> *"A significant limitation in previous literature is the lack of differentiation between algorithms that inherently include reinsertion and those that do not, leading to inconsistent comparisons. To ensure a strictly fair evaluation, we standardized two critical control variables across all experiments: the tie-breaking mechanism for the order of reinsertion and the stopping criteria. Furthermore, rather than arbitrarily assigning a reinsertion strategy, we evaluated every method under the three reinsertion methods. We report the best performance for each method, ensuring that the results reflect the maximum potential of the dismantling strategy rather than an inconsistent application of reinsertion."*
>
> **6. robustness-engineering analyses need stronger controls.**
>
> We clarify that our robustness analysis was conducted under controlled conditions to ensure validity. We evaluated the top two performing methods (NBC and CND) across three networks from different fields, adding 1% and 10% of links between common neighburs to the top 1% of nodes. The experimental setup and evaluation metrics were kept identical for all these experiments, which are the exact same as those used for all the other experiments.
>
> **7. NBC vs LGD-NA performance is kind of unclear. Please provide a table with absolute AUC and mean-field rank side-by-side for NBC, RA2, CND under identical dynamic/reinsertion settings,**
>
> We agree with the Reviewer that we could have made the location of all our quantitative results clearer in the main text and we have added the following explicit reference to the first paragraph of Subsection 4.4:
>
> *"Main results are visualized in Figure 2, and full quantitative results, including side-by-side comparisons of absolute AUC and mean-field ranks for all methods and fields, are reported in Tables 23 through 36 in the Appendix."*
>
> **8. and list fields where one strictly dominates.**
>
> We agree that breaking down performance dominance by field provides valuable insight. We will add the following specific analysis to Subsection 4.4:
>
> *"NBC strictly dominates as the top-ranking method across all fields. However, among the second-best performers, the LGD-NA methods lead in the majority of domains: CND ranks second in Internet networks, RA2 in Biomolecular and Brain networks, and RA2num in Covert networks. The only fields where non-LGD-NA methods rank second are Foodweb (Fitness Centrality), Infrastructure (GDM), and Social networks (GND)."*
>
> **9. I am kind of interested in the sensitivity analysis of the proposed method. For 3–4 strong methods, please plot AUC with no reinsertion vs 3 reinsertion policies and report ΔAUC. So that we can confirm rankings are stable or not.**
>
> We fully agree with the Reviewer that a side-by-side sensitivity analysis of strong methods is essential to verify the stability of rankings. To address this, we have expanded our Appendix F to ensure a complete comparison between our top-performing variants. While we previously visualized the LCC curves and AUC metrics for CND (Figures 10, 11), we have now followed the Reviewer’s recommendation and added the corresponding plots for RA2 (our second-best LGD-NA variant) as Figure 12.
>
> - https://anonymous.4open.science/r/15012-3E88/figure_10.png
> - https://anonymous.4open.science/r/15012-3E88/figure_11.png
> - https://anonymous.4open.science/r/15012-3E88/figure_12.png
>
> Second, we confirm that the method rankings are indeed stable. We have added the following analysis to Appendix F:
>
> *"Ranking Stability: Across all tested reinsertion methods, the mean-field ranking remains the same: NBC consistently outranks CND, which in turn outranks GDM. This order holds true both when comparing specific fixed reinsertion methods and when selecting the best-performing method for each dismantling method. However, we observe a nuanced interaction between the dismantling algorithms and their best reinsertion strategy: the optimal reinsertion method varies (R2 is optimal for NBC, R3 for CND, and R1 for GDM). For all other algorithms, though, R1 is the most effective reinsertion strategy.”*
>
> We once again thank the Reviewer for their valuable feedback and we ensured that the points raised are now fully addressed and clarified in the final manuscript. We welcome any further questions or feedback regarding our response.

---

> ### Author Response · Authors · 2025-11-21
> **Table of results**
>
> For brevity and clarity, we report selected excerpts of the new tables below. The full versions are available in the revised manuscript.
>
> - **Point 3**
>
> **Table 14: Average runtime (in seconds) by field and method for dynamic dismantling. Evaluated on networks of up to 23,000 nodes and 507,000 edges (n = 1,475). Average number of nodes N and number of edges E by field. In bold the fasted method.**
>
> Field | N$_{mean}$|E$_{mean}$|CND-CPU | CND-GPU |NBC-CPU |RA2-CPU |RA2-GPU |
> | :--- |:---:|:---:|:---:|:---:| :---:|:---:|:---:|
> Biomolecular |2,997 |11,855 |1,688.9|37.7 |174.5 |3,699.8 |**29.4** |
> Brain |97 |1,535 |7  |4.2 |**0.2** | 7.7 |2.4 |
> Covert |107 |266 |7.3 |0.9 |**0.2** |19.8 |0.6 |
> Foodweb |117 |1,087 |24.2 |2.4 |**0.2** |25.4 |1.9 |
> Infrastructure |664 |1,332 |2,610.8 |34.9 |**11.6** |2,441.8 |27 |
> Internet |5,708 |19,601 |6,149.2 |34.2 |138.6 |9,801.8 |**31.9**|
> Misc |2,880 |19,921 |3,641.8|54.9 |439.3 |5,065.1 |**53.2** |
> Social |3,267 |53,977 |8,322.5 |**149.4**|4,840.6 |12,474.4 |161 |
>
> - **Point 4**
>
> **Table 15: Runtime (in seconds) for NBC run on CPU (graph-tool) and GPU (cuGraph) on a subset of networks. Number of nodes N and number of edges E. In bold the fasted method.**
>
> ||N|E|NBC-GPU|NBC-CPU|
> | :--- |:---:|:---:|:---:|:---:|
> Foodweb Blackrock|86|375|4.7|**0.04**|
> Phonecall 2012|193|1,030|20.7|**0.08**|
> Rat Transcription 2010|524|1,081|37.9|**0.10**|
> Roadmap Winnipeg|1,040|1,595|848.3|**0.41**|
>
> - **Point 4**
>
> **Table 16: Average runtime (in seconds) and LCC AUC by field and method for dynamic dismantling. Evaluated where Nmin = 2, 235, Nmax = 9, 885, Emin = 10, 075, Emax = 506,437, (n = 157). In bold the fastest method per field. Average number of nodes N and number of edges E by field.**
>
> Field|N$_{mean}$|E$_{mean}$|Method|Running time (sec)|AUC
> | :--- |:---:|:---:|:---:|:---:|:---:|
> Biomolecular|5,528|26,804|NBC-20|437 |0.081
> ||||NBC|453|0.082
> ||||**CND**|**86**|0.087
> Infrastructure|5,803|15,068|**NBC**|**76**|0.021
> ||||NBC-20|105|0.022
> ||||CND|203|0.099
> Internet|6,623|31,292|NBC-20|49|0.016
> ||||NBC|196|0.016
> ||||**CND**|**29** |0.02
> Misc|5,444|53,289|NBC|1,220|0.106
> ||||NBC-20|655|0.106
> ||||**CND**|**121**|0.134
> Social|5,778|156,918|NBC-20|1,368 |0.274
> ||||NBC|14,961|0.277
> ||||**CND**|**405**|0.325
>
> - **Point 7:**
> **Tables 18/21/24/29: Mean field ranking for top three methods under different reinsertion methods. In bold the best method.**
>
> Method|Baseline|R1|R2|R3
> | :--- |:---:|:---:|:---:|:---:|
> NBC|**1**|**1**|**1**|**1**|
> CND|3.625|2.5|2.5|2.5
> RA2|4|3.25|3.25|3.25
>
> - **Point 9:**
>
> **Table 8: Percentage improvement for the mean AUC and mean number of removals for each reinsertion method over the baseline for CND and RA2 (n = 1,237). In bold the method that improves the baseline the most, by field.**
>
> |**CND**|R1|R2|R3
> | :--- |:---:|:---:|:---:|
> Biological|**11.2%**|9.9%|11.0%|
> Connectome|1.9%|**2.0%**|**2.0%**|
> Covert|5.3%|**6.0%**|**6.0%**|
> Foodweb|**2.2%**|2.0%|2.0%|
> Infrastructure|5.6%|5.0%|**5.7%**|
> Internet|**6.1%**|5.5%|5.8%|
> Misc|**4.4%**|4.0%|4.2%|
> Social|**3.2%**|3.1%|**3.2%**|
>
> |**RA2**|R1|R2|R3
> | :--- |:---:|:---:|:---:|
> Biological|8.0%|8.3%|**8.5%**|
> Connectome|-0.3%|**0.0%**|-0.1%|
> Covert|3.4%|3.5%|**3.8%**|
> Foodweb|**2.3%**|2.0%|2.1%|
> Infrastructure|**4.1%**|3.4%|4.0%|
> Internet|**5.0%**|4.3%|4.9%|
> Misc|3.2%|3.1%|**3.4%**|
> Social|**3.1%**|**3.1%**|**3.1%**|

---

> > ### Comment · Reviewer_MAH5 · 2025-11-26
> >
> > Dear Author,
> >
> > I would like to thank you for the rebuttal. My concerns regarding the terminology consistency, accelerated-NBC baselines, AUC and mean-field rank and so on are addressed. Yet I am still curious about the motivation behind not working on directed graphs. Since this is identified as a research gap, the community is much happier and more welcome to the research on this topic, rather than making more SOTA models on the undirected graphs. I would like to ask the authors for the deeper motivation.
> >
> > Best

---

> ### Author Response · Authors · 2025-11-28
> **Answer to Reviewer MAH5 about directed networks**
>
> We thank the Reviewer for the constructive discussion and for confirming that our previous clarifications were satisfactory.
>
> Regarding the focus on undirected graphs, we agree that a deeper justification is necessary. We have added **Appendix L: Application to Directed Networks** to provide empirical and theoretical evidence supporting this choice.
>
> The new Appendix L shows that our framework, even when applied by simply treating directed networks as undirected, outperforms a dedicated directed network dismantling algorithm (DNE) on several real-world directed networks. This strong empirical result demonstrates the robustness and general effectiveness of our latent geometry approach.
>
> Furthermore, we clarify that the theoretical link between topology and latent geometry in directed networks is still an emerging area. Designing new asymmetric measures for directed graphs is a clear and promising direction for future research.
>
> *“Appendix L: Application to Directed Networks*
>
> *While network dismantling has primarily targeted undirected networks (Artime et al., 2024), many critical real-world systems, such as neural circuits, social media platforms, and financial transaction systems, are inherently directed. Although our LGD-NA framework is designed for undirected networks, we explore its applicability to directed graphs. Research on directed network dismantling is relatively underexplored. Directed Node Entanglement (DNE) (Wu et al., 2025) generalizes the network density matrix to specifically capture and disrupt directed information flow within a system. Ma et al. (2022) utilizes the non-backtracking matrix to identify and remove the minimum set of nodes connecting distinct edge modules. Dismantling on Signed Networks based on Evolutionary Deep Reinforcement Learning (DSEDR) (Ou et al., 2024) is an evolutionary deep reinforcement learning approach designed to dismantle signed networks by optimizing a novel objective function. Embedding-based Signed Network Dismantling (ESND) (Xie et al., 2025) combines giant component detection, network embedding, and node clustering to identify critical nodes.*
>
> *We evaluate our LGD-NA framework on directed graphs using two approaches: applying the original method (treating the graph as undirected) and a directed variant where node scores are computed aggregating only outgoing links, to model information flow. For comparison, we implement Directed Node Entanglement (DNE) as a baseline specific to directed networks. We exclude other methods—such as the set-based approach of Ma et al. (2022) and the complex, multi-step frameworks of DSEDR and ESND—to maintain a focused comparison on computationally efficient, node-ranking strategies.*
>
> *We focus on three directed networks from different fields. College Messages (Social) (Panzarasa, 2009) represents private messages on a social network at UC Irvine (1,899 nodes, 20,296 edges). Drosophila Connectome (Brain) (Shiu et al., 2024), focuses on the sugar-sensing gustatory circuit (377 nodes, 13,671 edges). Email EU (Social) (Paranjape, 2017) are anonymized internal emails from a European research institution (986 nodes, 24,929 edges).*
>
> *Table 17 shows two key results. First, for our LGD-NA framework, treating directed networks as undirected yields superior dismantling performance over our directed variant. Second, even when applied in this undirected way, LGD-NA outperforms the directed-specific DNE baseline, demonstrating its effectiveness on directed networks.*
>
> *Finally, our framework is predicated on the association between network topology and latent geometry. For directed networks, this relationship is less established, with only preliminary studies (Allard et al., 2024). Therefore, designing asymmetric network measures that account for directional flow remains an open challenge, yet a promising direction for future research.“*
>
> **- Table 17:  LCC AUC for the three LGD-NA estimators treating a directed network as undirected, their directed variant, and DNE, a specific directed network dismantling method. In bold the best performing method per network.**
>
> ||College Messages|Drosophila Connectome|Email EU|
> | :--- |:---:|:---:|:---:|
> CND|**0.1426**|**0.4470**|**0.2710**|
> CND-Directed|0.1537|0.4512|0.3064
> RA2|0.1427|0.4529|0.2824
> RA2-Directed|0.1466|0.4507|0.2955
> RA2$_{num}$|0.1431|0.4625|0.2873
> RA2$_{num}$-Directed|0.1485|0.4653|0.2984
> DNE|0.4069|0.4923|0.4311

---

> > ### Comment · Reviewer_MAH5 · 2025-11-28
> >
> > Dear Authors,
> >
> > I would like to thank you for the extra experiments on the directed graphs. In the experiment, the proposed method shows surprisingly good performance on directed graphs. I believe this addition makes the method come with more contributions. Although I am still curious about the inferior performance of directed variants, I think this would be a promising direction for future work. I will raise my rating to 6 once I can do that, as currently I cannot update my recommendation score. (That is strange.)
> >
> > Cheers

---

> > > ### Author Response · Authors · 2025-12-03
> > > **Final Answer to Reviewer MAH5**
> > >
> > > We thank the Reviewer for their positive feedback on our answer regarding directed networks and their willingness to increase their score. We would also like to draw the attention of the Reviewer to some additional runtime experiments we ran, that can add to the Reviewer’s questions around GPU acceleration.
> > >
> > > We used the nPSO model in a controlled environment to determine when GPU acceleration provided a meaningful advantage. Our results show that CND on GPU begins to outperform NBC on CPU over approximately 1,000 nodes or 100,000 edges. We have added Figures 16 (https://anonymous.4open.science/r/15012-3E88/figure_16.png ) and 17 (https://anonymous.4open.science/r/15012-3E88/figure_17.png), and Table 37 to Appendix J to visualize these findings. Appendix J now contains the following paragraph:
> > >
> > > *“To empirically validate these runtime advantages in a controlled setting, we conducted additional experiments using the nPSO model (Muscoloni & Cannistraci, 2018) with network sizes ranging from 10 to 5,000 nodes and densities of 4\%, 8\%, and 20\%. We keep the temperature (lower temperature yields higher clustering) fixed ($T=0.3$) and adjust the number of communities to suit the size of the network ($C=2$ for $N \in \{10, 50, 100\}$, $C=5$ for $N \in \{500, 1,000\}$, $C=10$ for $N \in \{5,000\}$). Our results demonstrate that GPU-accelerated LGD-NA methods begin to show running time advantages over NBC-CPU when networks exceed approximately 1,000 nodes (see Figure 17) or 100,000 edges (see Figure 16). This  threshold aligns with our observations in real-world networks (see Figures 3 and 18, and Table 14), where GPU methods achieve superior running times for larger-scale networks such as biomolecular, internet, and social networks, while offering no runtime benefit for smaller networks where CPU implementations remain efficient.”*
> > >
> > > Finally, we have uploaded all the referenced figures and tables at the following link, for quick and easy access for the benefit of the Reviewer: https://anonymous.4open.science/r/15012-3E88.

---

### Official Review · Reviewer_a3SW · 2025-10-28

**Soundness:** 2
**Presentation:** 3
**Contribution:** 2
**Rating:** 2
**Confidence:** 4

**Summary:**

Latent Geometry-Driven Network Automata Framework
The LGD-NA framework introduces a novel approach to network dismantling by leveraging local network automata rules to estimate effective link distances. ​

LGD-NA utilizes local automata rules to approximate node distances, enhancing dismantling efficiency. ​
It identifies critical nodes and captures latent manifold information for effective network dismantling. ​
The framework outperforms existing algorithms, including machine learning methods, across 1,475 real-world networks. ​
A common-neighbor-based rule achieves near state-of-the-art performance, demonstrating the effectiveness of minimal local information. ​
GPU acceleration enables efficient scaling to large networks, significantly reducing running times.

The paper empirically validates the approach.

**Strengths:**

Contributions:
 1.  Latent Geometry-Driven (LGD) dismantling, where methods estimate effective node distances on a network’s latent manifold to expose critical structural information.

2.  LGD-NA framework uses local network automata rules to approximate these geometric distances; a node’s summed distance to its neighbors estimates how critical it is for dismantling.

3.  simple common neighbors-based rule, which we term Common Neighbor Dissimilarity (CND), is highly effective, achieving performance close to the state-of-the-art method, NBC.

4.  comprehensive experimental validation on an ATLAS of 1,475 real-world networks across 32 complex systems domains, the largest and most diverse collection to date, showing that LGD-NA consistently outperforms all other existing dismantling algorithms, including machine learning methods.

**Weaknesses:**

I have issues with the objectives of a framework for LGD dismantling: what is the purpose of knowing about dismantling?

--fault tolerance: will a system fail?

--communications: will communications be disrupted?

--security: can a system's security be compromised?

The metrics used are totally abstract, and I would like to see a real-world application that shows the significance of your results. At present, it is an extension of small-world theory with no clear application.




The paper makes overly general claims about manifolds and the applications of this approach that must be scaled back, e.g, "network geometry captures essential structural and dynamical properties of complex systems". There is also the strong claim: "a novel strategy to engineer network robustness". The latter claim is unsubstantiated. Robustness is undefined; you are engineering graph-theoretical properties, not a precise notion of robustness.

Originally, small-world graph research showed that systems possess shared graph metrics. Now, you are using dismantling methods, but for what purpose? You need to show how dismantling impacts system performance. All you do is apply small-world theory with an application driven framework, and you MUST look at the application.

You need to be more precise about your use of manifolds and manifold theory. In complex systems, a latent manifold is a hidden, often lower-dimensional structure that captures the essential dynamics or configurations of the system. These manifolds are not directly observable but can be inferred from data using techniques like:

--Nonlinear dimensionality reduction (e.g., t-SNE, UMAP, Isomap)

--Autoencoders and variational autoencoders (VAEs)

--Diffusion maps or spectral embeddings

--Manifold learning in dynamical systems (e.g., Koopman operator theory)

Graph metrics like small-worldness, degree heterogeneity, clustering coefficient, and community structure are NOT manifolds themselves, but they characterize the topology of networks that may be embedded in or arise from latent manifolds.

--Community structure can hint at stratification or clustering on a manifold.
--Small-worldness suggests short geodesic distances on a latent space.
--Degree heterogeneity may reflect curvature or singularities in the manifold.

 Manifolds in complex systems can be:

--Non-Euclidean: Curved, with nontrivial Riemannian metrics.

--Stratified: Composed of multiple manifolds of different dimensions glued together (e.g., hybrid systems with discrete modes).
--Singular: Containing points where the manifold structure breaks down (e.g., bifurcations, phase transitions).

Multi-scale: Different dynamics dominate at different scales, requiring nested or hierarchical manifolds.

So while these metrics are topological descriptors, they can be indirect indicators of underlying manifold geometry.

For example, a system that can operate in multiple modes may have different structural properties important per mode. This corresponds to a system (in your sense) having different small-world metrics representing different modes. Typically, people use persistent homology to show structural consistency across dynamics---what you compute is different from this.

**Questions:**

Your notion of NETWORK ROBUSTNESS is theoretical only. Please define it precisely. How does NETWORK ROBUSTNESS apply to the 3 applications pointed out above, in a precise sense: fault tolerance, communications,  security? I am looking for validation of the strong claim: "a novel strategy to engineer network robustness".

---

> ### Author Response · Authors · 2025-11-25
> **Answer to Reviewer a3SW 1/6**
>
> We sincerely thank the Reviewer for the insightful assessment. We fully agree with their emphasis on the necessity of linking the dismantling task to concrete real-world applications. Motivated by these points, we have expanded our analysis with additional experiments on real-world cases that explicitly demonstrate the utility of our framework in scenarios such as fault tolerance, communication, and security. These new results also substantiate our approach as a viable strategy for engineering network robustness, that we have more rigorously defined, overall significantly strengthening the paper's contribution.
>
> **1. I have issues with the objectives of a framework for LGD dismantling: what is the purpose of knowing about dismantling? --fault tolerance: will a system fail? --communications: will communications be disrupted? --security: can a system's security be compromised? and I would like to see a real-world application that shows the significance of your results. At present, it is an extension of small-world theory with no clear application. Originally, small-world graph research showed that systems possess shared graph metrics. Now, you are using dismantling methods, but for what purpose? You need to show how dismantling impacts system performance. All you do is apply small-world theory with an application driven framework, and you MUST look at the application.**
>
>
> We fully agree with the Reviewer that the objectives of dismantling must be clearly defined and grounded in practical applications. To demonstrate the significance of our results, we have conducted additional experiments on four real-world networks (the Drosophila Connectome, a Paris/Brussels Terrorist Cell, a Ryanair Flight Map, and a School Contact network for epidemic spread) that specifically address the three domains highlighted: fault tolerance, communications, and security.
>
> Accordingly, we have revised the introduction to explicitly state these objectives. Following the first paragraph, we have added:
>
> *"The task of dismantling serves a dual purpose. It determines whether a system is robust and how to reinforce desirable networks, for example preventing system failure in a flight network or security compromises in internet infrastructure. Conversely, it reveals how to disrupt undesirable systems, severing communications in terrorist cells or halting the spread of an epidemic.“*
>
> To provide concrete validation, we have added a new subsection (Section 4.7) to the main text and added Figure 4 (https://anonymous.4open.science/r/15012-3E88/figure_04.png). This subsection validates the effectiveness of our LGD-NA framework for dismantling using domain-specific performance indicators across four real-world networks.
>
> These experiments address the three specific applications:
>
> - **Fault Tolerance**: Drosophila Connectome (metric: sensory neuron firing rate) and Ryanair Flight Map (metric: Global Efficiency).
>
> - **Communications**: Paris/Brussels Terrorist Cell (metric: Commander Reachability).
>
> - **Security**: School Contact Network (metric: SEIR epidemic model outbreak size).
>
> Below is the new subsection added to the manuscript.
>
> *“ 4.7. Real-World Applications: Fault Tolerance, Security, and Communications*
>
> *To demonstrate the practical utility of LGD-NA, we evaluate its performance on four distinct real-world systems using domain-specific functional metrics (full experimental details in Appendix I).*
>
> *- Drosophila Connectome (Shiu et al., 2024) (Fault Tolerance): We utilize a Spiking Neural Network (SNN) model of the sugar-sensing circuit. The metric is the sensory neuron firing rate required to trigger the proboscis extension response.*
>
> *- Paris/Brussels Terrorist Cell (Gutfraind and Genkin, 2017) (Security & Communications): We analyze the network responsible for the 2015 Paris and 2016 Brussels attacks. The metric is Commander Reach, defined as the percentage of operatives able to communicate with at least one of the three key commanders.*
>
> *- Ryanair Flight Map (Cardillo et al., 2013) (Fault Tolerance): A transportation network where we measure Global Efficiency ($E_{glob}$).*
>
> *- School Contact Network (Mastrandrea et al., 2015) (Security/Epidemics): We simulate an epidemic using an SEIR model (Anderson and May, 1991). The metric is the Final Outbreak Size.*

---

> ### Author Response · Authors · 2025-11-25
> **Answer to Reviewer a3SW 2/6**
>
> *Our results in Figure 4 show that dismantling strategies effectively degrade the functional performance across all four systems. In the Drosophila Connectome and Paris/Brussels Terrorist Cell, we observe particularly sharp drops in performance metrics after removing only a small fraction of nodes (~5%). We observe a more gradual deterioration in the global efficiency of the Ryanair Flight Map and the viral spread within the School Contact network. This functional collapse is particularly significant for the two adversarial scenarios (Paris/Brussels Terrorist Cell and School Contact Network): it confirms that LGD-NA is effective for security and communication disruption, efficiently suppressing epidemic outbreaks and isolating hostile leadership with minimal intervention.*
>
> *We subsequently applied our strategy for engineering network robustness to these four scenarios, demonstrating its effectiveness. As shown in Table 13, the reinforced networks are significantly harder to dismantle, achieving robustness gains of up to 363%. This increased resilience is evident across both our original topological metric (LCC AUC) and the domain-specific functional metrics defined for each case.*
>
> *For the Drosophila Connectome, the analysis informs the resilient and redundant design of fault-tolerant neuromorphic circuits by mimicking its biological wiring (Suarez et al., 2021, Hame et al., 2021). In the Ryanair Flight Map, it identifies specific hubs where reinforcement prevents systemic failure. Finally, for adversarial networks (Paris/Brussels Terrorist Cell and Epidemic), our robustness analysis serves a diagnostic purpose when faced with incomplete data. Since social networks, and especially covert ones, often contain unobserved links (e.g., dormant ties or unreported contacts), calculating an empirical robustness ceiling allows us to estimate the margin of error required for successful security operations with partial observability.”*
>
> Then we add the following explanation of our experiments as a new Appendix section:
>
> *“Appendix I: Functional Metrics and Real-World Application Details*
>
> *We now provide the specific experimental setup for the real-world functional experiments described in Subsection 4.7. Unlike topological metrics (e.g., LCC size), these experiments measure the functional performance of the systems under dismantling.*
>
> ***Drosophila Connectome**: We utilize the central brain connectome of Drosophila melanogaster (FlyWire), comprising over 125,000 neurons and 50 million synaptic connections. To assess functional performance, we employ a Leaky Integrate-and-Fire (LIF) spiking computational model, as established by Shiu et al. (2024). We focus on the sugar-sensing gustatory circuit (which results in a smaller subnetwork of 377 nodes and 13,671 edges), a critical pathway for feeding initiation. We simulate the activation of sugar-sensing Gustatory Receptor Neurons (GRNs), with a stimulation frequency of 100 Hz. We use the default parameters provided by Shiu et al. (2024), with the trial duration adjusted to 100 ms for the sake of time. The performance metric is the Motor Neuron Firing Rate, MN9. A dismantling attack is considered successful if the removal of specific interneurons prevents the propagation of the signal from the sensory GRNs to the motor neurons, causing the firing rate to drop.*
>
> ***Paris/Brussels Terrorist Cell**: We analyze the network of the terrorist cell responsible for the November 2015 Paris attacks and the 2016 Brussels bombings (Gutfraind and Genkin, 2017), with 77 nodes and 271 edges. In adversarial networks, the ability of the leadership to communicate with the rest of the operational network, and vice versa, is crucial. As a result, we define Commander Reach as the percentage of network operatives (nodes) that retain a valid communication path to at least one of the three identified cell commanders.*
>
> ***Ryanair Flight Map**: This network represents the flight routes of Ryanair in Europe (Cardillo et al., 2013), with 128 nodes and 601 edges. Nodes represent airports, and edges represent direct flight connections. We measure the functional integrity of the transport system using Global Efficiency ($E_{glob}$), rather than the Average Shortest Path Length ($APL$), because: $APL$ is mathematically undefined (or diverges to infinity) when a network fragments into disconnected components, which inevitably occurs during dismantling. Global Efficiency avoids this divergence by averaging the inverse geodesic distances.*
> $$E_{glob} = \frac{1}{N(N-1)} \sum_{i \neq j} \frac{1}{d_{ij}}$$
> Where $N$ is the number of airports and $d_{ij}$ is the shortest path length between airport $i$ and $j$. If no path exists, $\frac{1}{d_{ij}} = 0$. This metric correctly quantifies the remaining communication capacity of a fragmented network.*

---

> ### Author Response · Authors · 2025-11-25
> **Answer to Reviewer a3SW 3/6**
>
> ***School Contact Network**: We utilize a contact network collected from a French high school (Mastrandrea, 2015), with 327 nodes and 5,818 edges, where nodes represent students and edges represent close-proximity physical contacts capable of disease transmission. We simulate the spread of an infectious disease using a classic Susceptible-Exposed-Infectious-Recovered (SEIR) compartmental model (Anderson and May, 1991). Unlike basic SIR models, SEIR includes an "Exposed" state to account for the latency period typical of real-world pathogens. The simulation begins with 5% of nodes infected; over 200 discrete time steps, individuals progress from Susceptible (S) to Exposed (E) (based on contact with infected neighbors and rate $\beta=0.01$), then to Infected (I) (based on latency rate $\alpha=0.01$), and finally to Recovered $R$ (based on recovery rate $\gamma=0.01$). We average the results of 50 independent simulations. The functional integrity of the network is measured by the Final Outbreak Size, defined as the total percentage of the population that was infected and subsequently recovered. The size is normalized with the size of the largest connected component, from which the epidemic starts from.*
>
> ***Engineering Network Robustness Protocol**: To validate our method for "engineering robustness", we also reinforce these networks as defined in Subsection 4.7, choosing the top 5% of nodes and adding 10% of links, and rerun the dismantling process under the exact same conditions. Table 13 shows the quantitative results for our original dismantling metric, LCC AUC, alongside the functional metrics defined above. Notably, both metrics yield highly similar rankings of the top methods, further validating our choice of LCC AUC as a robust evaluation standard.”*
>
> We add the following sentence to the Abstract…
>
> *“We validate LGD-NA's practical utility on domain-specific functional metrics, spanning neuronal firing rates in the Drosophila Connectome, transport efficiency in flight maps, outbreak sizes in contact networks, and communication pathways in terrorist cells.”*
>
> to the Introduction…
>
> *“We further validate the practical utility of our dismantling framework and robustness engineering method by demonstrating their impact on domain-specific functional metrics, including neuronal firing rates in the Drosophila Connectome, flight map efficiency, epidemic sizes, and communication reachability in terrorist cells.”*
>
> and finally to the conclusion:
>
> *“Crucially, we demonstrate the practical utility of our framework across diverse domains, from informing the design of neuromorphic circuits and reinforcing transport hubs, to disrupting terrorist cells and estimating the 'hidden resilience' of adversarial networks with unobserved links.”*
>
> **- Table 13: Network-specific evaluation metric and the original LCC AUC performance metric for four real-world networks, for NBC, CND, and RA2.**
>
> Drosophila Connectome||Firing Rate (Freq. = 100 Hz) AUC||
> | :--- |:---:|:---:|:---:|
> ||Baseline|Reinforced|
> NBC|**0.030**|**0.139**|+363%
> CND|0.093|0.180|+93%
> RA2|0.042|0.153|+261%
>
>
> Paris/Brussels Terrorist Cell||Commander's Reach AUC||
> | :--- |:---:|:---:|:---:|
> ||Baseline|Reinforced|
> NBC|0.182|0.305|+67%
> CND|0.149|**0.182**|+22%
> RA2|**0.095**|0.274|+188%
>
>
> Ryanair Flight Map||Global Efficiency AUC||
> | :--- |:---:|:---:|:---:|
> ||Baseline|Reinforced|
> NBC|0.040|**0.111**|+179%
> CND|**0.039**|0.121|+207%
> RA2|0.040|0.116|+189%
>
>
> School Contact||Final Outbreak Size AUC||
> | :--- |:---:|:---:|:---:|
> ||Baseline|Reinforced|
> NBC|0.351|0.366|+4%
> CND|0.288|0.339|+17%
> RA2|**0.285**|**0.312**|+9%
>
>
>
> **2. The metrics used are totally abstract**
>
> We agree with the Reviewer that the LCC AUC can appear abstract, so we’ve added a paragraph in Appendix M to clarify why we chose the LCC AUC as our main evaluation metric.
>
> *"We employ LCC AUC as our primary evaluation metric because it provides a unified standard for comparing methods across 32 distinct complex system domains. As the established standard in the vast majority of dismantling studies (Artime et al., 2024), it allows for direct comparisons between diverse networks from disparate fields and different dismantling algorithms. While dynamical metrics offer specific insights, simulating 'live' system dynamics for every network is computationally unfeasible and conceptually inconsistent given the broad scope of domains. Crucially, our functional analysis in Subsection 4.7 demonstrates that rankings based on LCC AUC align closely with domain-specific functional metrics, validating LCC fragmentation as a reliable proxy for functional disruption."*

---

> ### Author Response · Authors · 2025-11-25
> **Answer to Reviewer a3SW 4/6**
>
> **3. There is also the strong claim: "a novel strategy to engineer network robustness". The latter claim is unsubstantiated. Robustness is undefined; you are engineering graph-theoretical properties, not a precise notion of robustness. Your notion of NETWORK ROBUSTNESS is theoretical only. Please define it precisely. How does NETWORK ROBUSTNESS apply to the 3 applications pointed out above, in a precise sense: fault tolerance, communications, security? I am looking for validation of the strong claim: "a novel strategy to engineer network robustness".**
>
> We agree that our initial use of "robustness" lacked a formal definition. We have added the following clarification to Subsection 4.6:
>
> *"Robustness is defined as the ability of a system to continue functioning when subjected to perturbations (Artime et al., 2024). In this initial context, we define attack tolerance, quantified by the LCC AUC, as a robustness measure itself, representing the system's structural integrity under dismantling attacks.”*
>
> However, we acknowledge that specific networks operate under distinct functional constraints. Consequently, we conducted the functional experiments described in Subsection 4.7 to demonstrate how our robustness engineering method improves these domain-specific functional metrics too.
>
> We have added the following paragraph to Appendix H to explicitly link these results to the three key concepts raised by the Reviewer:
>
> *“Our results confirm that engineering robustness translates into functional gains across the four real networks studied:*
>
> - ***Fault Tolerance**: In the Drosophila Connectome and Ryanair Flight Map, the analysis informs the design of fault-tolerant neuromorphic circuits and identifies critical hubs where reinforcement prevents systemic transport failure.*
>
> - ***Security & Communications**: For adversarial systems (Terrorist Cell and School Contact Network), we can calculate the theoretical robustness ceiling, by accounting for unobserved links (e.g., dormant ties in covert networks) and quantify the margin of error required to successfully disrupt communications or secure the network against epidemics despite incomplete data."*
>
> **4. The paper makes overly general claims about manifolds and the applications of this approach that must be scaled back, e.g, "network geometry captures essential structural and dynamical properties of complex systems".**
>
> We agree with the Reviewer that our wording was not specific enough, so we have revised the wording of the following paragraph in the Introduction:
>
> *“Latent geometry has been recognized as a key principle for understanding the structure and complexity of real-world networks. Recent works in network science suggest that the latent geometry of complex networks could explain critical network characteristics such as small-worldness, degree heterogeneity, clustering, and navigability, and drives critical processes like efficient information flow (Boguna et al., 2021; 2009; Kleinberg, 2000; Wu et al., 2015; Serrano et al., 2008; Krioukov et al., 2010; Muscoloni & Cannistraci, 2019; 2018a).“*

---

> ### Author Response · Authors · 2025-11-25
> **Answer to Reviewer a3SW 5/6**
>
> **5. You need to be more precise about your use of manifolds and manifold theory. In complex systems, a latent manifold is a hidden, often lower-dimensional structure that captures the essential dynamics or configurations of the system. These manifolds are not directly observable but can be inferred from data using techniques like:**
> - **Nonlinear dimensionality reduction (e.g., t-SNE, UMAP, Isomap)**
> - **Autoencoders and variational autoencoders (VAEs)**
> - **Diffusion maps or spectral embeddings**
> - **Manifold learning in dynamical systems (e.g., Koopman operator theory)**
>
> **Graph metrics like small-worldness, degree heterogeneity, clustering coefficient, and community structure are NOT manifolds themselves, but they characterize the topology of networks that may be embedded in or arise from latent manifolds.**
> - **Community structure can hint at stratification or clustering on a manifold.**
> - **Small-worldness suggests short geodesic distances on a latent space.**
> - **Degree heterogeneity may reflect curvature or singularities in the manifold.**
>
> **Manifolds in complex systems can be:**
> - **Non-Euclidean: Curved, with nontrivial Riemannian metrics.**
> - **Stratified: Composed of multiple manifolds of different dimensions glued together (e.g., hybrid systems with discrete modes).**
> - **Singular: Containing points where the manifold structure breaks down (e.g., bifurcations, phase transitions).**
> - **Multi-scale: Different dynamics dominate at different scales, requiring nested or hierarchical manifolds.**
> **So while these metrics are topological descriptors, they can be indirect indicators of underlying manifold geometry.**
> **For example, a system that can operate in multiple modes may have different structural properties important per mode. This corresponds to a system (in your sense) having different small-world metrics representing different modes. Typically, people use persistent homology to show structural consistency across dynamics---what you compute is different from this.**
>
> We thank the Reviewer for this precise and valuable distinction between topological descriptors (such as small-worldness or community structure) and the underlying latent manifolds from which they arise. We fully agree that graph metrics are not manifolds themselves, but rather "indirect indicators of underlying manifold geometry."
>
> We have added a new section, Appendix B: Theoretical Distinctions between Graph Metrics and Latent Manifolds, where we explicitly adopt the distinctions provided by the Reviewer.
>
> *“To avoid ambiguity regarding the use of manifold theory in complex systems, we clarify the distinction between topological descriptors and latent geometric spaces. In this work, we define the latent manifold as the hidden, lower-dimensional structure that captures the essential configuration of the system.*
>
> *To infer the latent manifold from high-dimensional data, a range of general dimensionality reduction and manifold learning techniques can be applied. These approaches seek to map the data points into a continuous, lower-dimensional space where geometric proximity reflects similarity in the original space. They can be broadly categorized as methods preserving local structure (e.g., t-SNE, UMAP, and Minimum Curvilinear Embedding (MCE)), methods based on calculating intrinsic distances (e.g., Isomap (ISO) and its variants), spectral methods (e.g., Laplacian eigenmaps and Diffusion Maps), and deep learning techniques (e.g., Autoencoders and VAEs) that learn the latent code necessary for data reconstruction. Finally, specialized manifold learning approaches in dynamical systems (e.g., Koopman operator theory) can transform complex, nonlinear dynamics into simpler, linear representations within a manifold.*
>
> *When data is organized as a complex network, the latent manifold is typically inferred using network embedding techniques specifically designed to preserve the network's topology. These methods fall into three broad categories: spectral methods (e.g., spectral clustering) which use the algebraic properties of the graph matrices; deep learning approaches (e.g., DeepWalk, Node2Vec, and Graph Autoencoders (GAE)) which learn representations using neural networks trained on structural information like random walks, while Graph Neural Networks (GNNs) have emerged as the state-of-the-art for learning task-specific embeddings using topology and node/edge features; and geometric approaches such as Hyperbolic network embeddings. These geometric methods (e.g., Poincaré embeddings, Hypermap) utilize non-Euclidean geometries, such as negative curvature, to efficiently capture the hierarchical and scale-free properties of complex networks. Specific algorithms like LPCS generate node coordinates by analyzing and ordering the network's community structure.*

---

> ### Author Response · Authors · 2025-11-25
> **Answer to Reviewer a3SW 6/6**
>
> *We distinguish this latent manifold from graph metrics. For example, standard topological graph descriptors such as small-worldness, community structure, and degree heterogeneity are not direct descriptors of the manifold themselves. However, since the network is sampled from a specific latent space, these observed properties are influenced by the manifold's geometry. Consequently, these topological metrics do characterize the topology of a network that is embedded in a specific latent space: for instance, small-worldness suggests short geodesic distances, community structure can imply stratification or clustering, and degree heterogeneity may reflect features such as local curvature or singularities.*
>
> *Our approach uses the topology of the observable network to infer the geometric distance between nodes within the network’s latent manifold, thereby allowing us to exploit the manifold's geometric properties for dismantling.*
>
> *Note that we include a GNN-based dismantling algorithm, Graph Dismantling with Machine learning (GDM) (Grassia et al., 2021), in our experiments. GDM is a GNN that is trained on optimally dismantled networks, and is considered a state-of-the-art dismantling algorithm (Artime et al., 2024, Grassia et al., 2021) that can implicitly capture features of the underlying latent geometry of the target network. The fact that our LGD-NA methods consistently outperform GDM in all situations suggests that our estimators might be yielding a more accurate estimation of the target network’s latent geometry.”*
>
> To empirically validate our claims regarding the underlying manifold geometry, we refer the Reviewer to the extensive analysis conducted in response to Reviewer jELD (now included in Appendix C: Geometric validation of LGD-NA estimators, with the additional Tables 21 (https://anonymous.4open.science/r/15012-3E88/table_21.png) and 22 and Figure 7 (https://anonymous.4open.science/r/15012-3E88/figure_07.png). In that analysis, we generated synthetic networks using the Non-Uniform Popularity-Similarity Optimization (nPSO) model, which provides ground-truth node coordinates in a hyperbolic space. We then measured the correlation between our heuristic estimators (CND and RA2) and the actual hyperbolic distances between nodes.
>
> For the Reviewer's convenience, we reproduce the key findings below. These results demonstrate a strong, statistically significant correlation between our estimators and the true geometric distances, confirming that our approach effectively captures the structural properties of the latent manifold.
>
> *“We further quantify this relationship by computing the correlation between our estimators and the ground-truth geometric distances in nPSO networks. We compare the link distance estimates for CND and RA2 against the true hyperbolic distances. Table 21 reports the mean Pearson correlation (averaged over 10 seeds), visualized using a color gradient where green corresponds to values approaching 1 and red to values approaching -1. The results clearly show that both CND and RA2 demonstrate a strong correlation with the true geometric distance. Table 22 additionally shows the Fisher p-value in parentheses. The Fisher p-value, which tests the null hypothesis that no correlation exists, is effectively zero in all cases, confirming all reported correlations are highly statistically significant. This confirms our LGD-NA measures’ ability to accurately estimate the geometric distance between nodes. Through the node aggregation step in our LGD-NA framework, it is then able to identify those nodes that connect far-away regions in the latent space.”*
>
> - **Table 21: Pearson correlation between estimated link weights from CND and RA2 versus true geometric distances in nPSO networks. Mean values over 10 seeds are reported. The power-law exponent 𝛾 represents the scale-freeness found in real-world networks. 𝞺 is the density of the networks. Fixed parameters are the number of nodes, N=500, and the number of communities, C=5. The temperature T controls the level of clustering (lower temperatures yield stronger clustering).**
>
> ||||𝞺=0.04|𝞺=0.08|𝞺=0.2
> | :--- |:---:|:---:|:---:|:---:|:---:|
> 𝛾=2|CND|T=0.3|0.534|0.641|0.664
> |||T=0.6|0.602|0.675|0.719
> |||T=0.9|0.649|0.690|0.746
> ||RA2|T=0.3|-0.044|0.062|0.319
> |||T=0.6|0.066|0.239|0.562
> |||T=0.9|0.140|0.361|0.682
> 𝛾=3|CND|T=0.3|0.329|0.370|0.394
> |||T=0.6|0.543|0.512|0.473
> |||T=0.9|0.607|0.553|0.510
> ||RA2|T=0.3|0.301|0.388|0.530
> |||T=0.6|0.441|0.542|0.625
> |||T=0.9|0.473|0.588|0.669

---

> ### Author Response · Authors · 2025-12-03
> **Additional Answer to Reviewer a3SW**
>
> We would like to draw the attention of the Reviewer to some additional experiments we made about the geometric validation of our latent-geometry estimators used in our LGD-NA framework.
>
> 1. We made Figure 7 clearer (https://anonymous.4open.science/r/15012-3E88/figure_07.png), we agree that the previous version was slightly unclear and have now made the necessary changes. We now clearly see how our estimators identify the nodes at the centre of the hyperbolic disk.
>
> 2. We made additional analysis on how our estimators were able to estimate the true geometric distance in the nPSO networks (a network with a known hyperbolic geometry). We have added Figure 8 (https://anonymous.4open.science/r/15012-3E88/figure_08.png) and Tables 4 (https://anonymous.4open.science/r/15012-3E88/table_04.png) and 20 in Appendix C explaining these new experiments. We believe this validates in an even stronger manner our claims about our latent-geometry estimators. Below are the new paragraphs in Appendix C that explains these experiments:
>
> *“To quantitatively support our claim, we evaluate how well the latent geometry estimators approximate the true hyperbolic distances. We use the hyperbolic distance correlation (HD-correlation) metric, the Pearson correlation between all pairwise geometrical shortest path distances in the networks’ original hyperbolic space and the weighted shortest path distances using the latent-geometry estimators as edge weights (Muscoloni et al., 2017). The higher this correlation, the better the latent-geometry estimator is able to recover the geometrical distances between pairs of nodes in a network’s underlying geometry.*
>
> *Table 4 shows a high HD-correlation for both CND and RA2 across all tested nPSO configurations, confirming that these measures used in our dismantling framework are effective latent geometry estimators. This is further supported by the statistical significance reported in Table 20.*
>
> *The Pearson correlation is visualized in Figure 8 for different parameters, visualizing how well the distance approximation changes as the network becomes less hyperbolic. As expected, for $\gamma=2$, the correlation decreases for both estimators with increasing temperature (i.e., reduced clustering and hyperbolicity). For the less hyperbolic $\gamma=3$ networks, this decreasing trend persists for CND but not for RA2. This suggests that CND remains a robust estimator of the latent geometry even when hyperbolic structure is less pronounced, whereas RA2's performance is more dependent on strongly hyperbolic conditions, consistent with our dismantling experiments.*
>
> *This visual and quantitative evidence demonstrates our LGD-NA measures' ability to accurately estimate the geometric distance between nodes. Consequently, the node aggregation step in our LGD-NA framework can successfully identify nodes that connect distant regions in the latent space.”*
>
> 3. Finally, we have uploaded all the referenced figures and tables at the following link, for quick and easy access for the benefit of the Reviewer: https://anonymous.4open.science/r/15012-3E88.

---

### Official Review · Reviewer_jELD · 2025-10-31

**Soundness:** 3
**Presentation:** 2
**Contribution:** 2
**Rating:** 6
**Confidence:** 2

**Summary:**

The paper proposes Latent Geometry-Driven Network Automata (LGD-NA) for network dismantling. The general idea behind is to estimate effective link distances using local “network automata” rules and rank nodes by the sum of their incident dissimilarities, where highest-scoring nodes are removed with dynamic recomputation. This paper evaluates the proposed methods across a very large “ATLAS” of 1,475 real-world networks covering 32 domains, using AUC of the LCC curve (10% threshold) as the metric. Additionally, it also studies optional reinsertion strategies, where a GPU implementation is reported to yield substantial speedups and latent-geometry estimators (including betweenness as a global estimator) can explain why these strategies work. Finally, this paper leverages CND’s explainability to “engineer robustness” by adding edges among neighbors of critical nodes.

**Strengths:**

1. The method offers an intuitive latent geometry framing with dynamic recomputation, which remains simple, general, and effective in practice.

2. The evaluation spans 1,475 networks across 32 domains and follows a clear LCC AUC metric and protocol.

3. Matrix formulations together with a GPU implementation yield speedups that make the approach scalable.

4. The paper translates insights into an easy robustness intervention by closing triangles among neighbors of critical nodes.

**Weaknesses:**

1. There are no theoretical guarantees that connect the proposed geometry estimators to near optimal dismantling orders, leaving the case largely empirical.

2. Runtime comparisons underrepresent strong GPU or approximate betweenness baselines, weakening claims about practical efficiency.

3. Mean field ranking and a single ten percent threshold may hide domain specific behavior, and guidance on when CND versus RA2 is preferable is limited.

**Questions:**

1. Could you clarify whether CND or RA2 actually outperform NBC in accuracy across the dataset or whether their advantage is primarily speed, and include a consolidated table with AUC gaps with and without reinsertion.

2. What explains the lack of GPU gains for NBC and would approximate betweenness or hybrid CPU GPU pipelines change the outcome.

3. How sensitive are the conclusions to thresholds other than ten percent and to alternative fragmentation metrics, especially when stratified by domain or graph statistics.

---

> ### Author Response · Authors · 2025-11-21
> **Answer to Reviewer jELD 1/6**
>
> We thank the Reviewer for their constructive feedback and for recognizing the intuitive nature and effectiveness of our LGD-NA framework, the scalability achieved by our GPU implementation, and the ability of our method to engineer network robustness. We address the specific questions and concerns raised below.
>
> **1. There are no theoretical guarantees that connect the proposed geometry estimators to near optimal dismantling orders, leaving the case largely empirical.**
>
> We thank the Reviewer for raising this fundamental point. We address this in two parts: the theoretical constraints of the problem, and the specific validity of our estimators.
>
> First, we wish to highlight that optimal network dismantling is an NP-hard problem. Consequently, no efficient algorithm currently possesses theoretical guarantees for optimal dismantling. To clarify this to the reader, we have revised Line 51 in the Introduction:
>
> *“Efficient network dismantling is challenging because identifying the minimal set of nodes for optimal disruption is an NP-hard problem: no known algorithm can solve it efficiently for large networks (Artime et al., 2024), forcing the field to rely on heuristic approximations.”*

---

> ### Author Response · Authors · 2025-11-21
> **Answer to Reviewer jELD 2/6**
>
> While we cannot guarantee an optimal dismantling order, we can demonstrate that our latent-geometry estimators can accurately identify important nodes as well as nodes that connect distant regions in the latent space, making these nodes prime candidates for efficient dismantling.
>
> We have revised Appendix G: RA measures in hyperbolic networks and renamed it to Appendix C: Geometric validation of LGD-NA estimators and included the following analysis. Note that we have added Tables 21 (https://anonymous.4open.science/r/15012-3E88/table_21.png) and 22, and added Figure 7 (https://anonymous.4open.science/r/15012-3E88/figure_07.png) to use the CND measure instead of RA2, for the sake of consistency to use our best-performing LGD-NA measure.
>
> *“To provide visual and empirical validation for our latent-geometry estimators, we analyze the ability of our latent-geometry estimators to identify node importance and estimate link distances using synthetically generated networks with a known geometry. As previously mentioned, the RA measures were introduced to serve as pre-weighting strategies for approximating angular distances associated with node similarities in hyperbolic network embeddings (Muscoloni et al., 2017).*
>
> *To investigate this, we synthetically generate networks using the non-uniform Popularity-Similarity Optimization (nPSO) model (Muscoloni & Cannistraci, 2018b). The nPSO model is built on the principle that radial coordinates represent hierarchy (popularity) while angular coordinates represent similarity. It produces networks that are both scale-free (characterized by a power-law degree distribution, meaning a network has a few highly connected hubs while the majority of nodes have few links) and clustered with distinct communities, closely mimicking the structure of many real-world complex systems. We utilize the nPSO network model specifically for this task because these networks are generated with known node coordinates and a known underlying hyperbolic geometry, making them highly suitable for validating geometry-related measures in network science.*
>
> *We generate various nPSO networks keeping the number of nodes (N = 500) and communities (C = 5) fixed. We test different network topologies by varying:*
>
> - *The power-law exponent γ ∈ {2, 3}, which represents common lower and upper bounds for real-world scale-free networks.*
> - *The number of nodes a new node will connect to when being added to the network, m ∈ {10, 20, 50}. This value represents approximately half of the average node degree, making the network more or less connected. This results in networks with three different density levels ⍴ ∈ {0.04, 0.08, 0.2}.*
> - *The temperature T ∈ {0.3, 0.6, 0.9}, which controls the level of clustering (lower temperatures yield stronger clustering).*
>
> *Figure 7 visualizes synthetically generated nPSO networks, with nodes colored according to their CND score, where red represents higher CND scores and blue lower ones. The figure clearly shows that nodes with high CND scores are consistently those with high radial centrality (i.e., the hubs located near the center of the hyperbolic disk). This relationship is particularly evident for gamma = 2, where the skewed degree distribution creates a clearer distinction between central hubs and peripheral nodes. The trend persists for gamma = 3, though it is less pronounced due to the presence of fewer “super-hubs”. This provides strong visual evidence that the CND estimator effectively identifies the most structurally important nodes.*
>
> *We further quantify this relationship by computing the correlation between our estimators and the ground-truth geometric distances in nPSO networks. We compare the link distance estimates for CND and RA2 against the true hyperbolic distances. Table 21 reports the mean Pearson correlation (averaged over 10 seeds), visualized using a color gradient where green corresponds to values approaching 1 and red to values approaching -1. The results clearly show that both CND and RA2 demonstrate a strong correlation with the true geometric distance. Table 22 additionally shows the Fisher p-value in parentheses. The Fisher p-value, which tests the null hypothesis that no correlation exists, is effectively zero in all cases, confirming all reported correlations are highly statistically significant. This confirms our LGD-NA measures’ ability to accurately estimate the geometric distance between nodes. Through the node aggregation step in our LGD-NA framework, it is then able to identify those nodes that connect far-away regions in the latent space.”*

---

> ### Author Response · Authors · 2025-11-21
> **Answer to Reviewer jELD 3/6**
>
> **- Table 21: Pearson correlation between estimated link weights from CND and RA2 versus true geometric distances in nPSO networks. Mean values over 10 seeds are reported. The power-law exponent 𝛾 represents the scale-freeness found in real-world networks. 𝞺 is the density of the networks. Fixed parameters are the number of nodes, N=500, and the number of communities, C=5. The temperature T controls the level of clustering (lower temperatures yield stronger clustering).**
>
> ||||𝞺=0.04|𝞺=0.08|𝞺=0.2
> | :--- |:---:|:---:|:---:|:---:|:---:|
> 𝛾=2|CND|T=0.3|0.534|0.641|0.664
> |||T=0.6|0.602|0.675|0.719
> |||T=0.9|0.649|0.690|0.746
> ||RA2|T=0.3|-0.044|0.062|0.319
> |||T=0.6|0.066|0.239|0.562
> |||T=0.9|0.140|0.361|0.682
> 𝛾=3|CND|T=0.3|0.329|0.370|0.394
> |||T=0.6|0.543|0.512|0.473
> |||T=0.9|0.607|0.553|0.510
> ||RA2|T=0.3|0.301|0.388|0.530
> |||T=0.6|0.441|0.542|0.625
> |||T=0.9|0.473|0.588|0.669
>
> **2. Runtime comparisons underrepresent strong GPU [...] baselines [...] weakening claims about practical efficiency. [...] What explains the lack of GPU gains for NBC?**
>
> We wish to further clarify the technical reasons explaining the lack of GPU gains for NBC and our subsequent decision to exclude it as a baseline.
>
> - We tested the GPU implementation of NBC provided by cuGraph and found it to be significantly slower than the highly optimized C++ implementation in graph-tool (CPU).
> - This makes sense because, unlike our LGD-NA method, which leverages matrix multiplication, an operation ideally suited for GPU, NBC relies on counting all-pairs shortest paths. This operation is inherently sequential and cannot be parallelized effectively on GPUs.
> - While literature on GPU-accelerated NBC exists, reliable, general-purpose implementations are scarce. Most are hardware-specific, not publicly available, or rely on heuristics tailored to specific network topologies.
>
> While these constraints are noted in Subsection 4.5 and Appendix J, we have also added an explicit clarification to the main text as an additional paragraph to Subsection 4.5:
>
> *“We report only the CPU running time for NBC, as its GPU implementation did not yield any speedup. While some studies report GPU implementations of NBC with improved performance (Fan et al., 2017; Shi & Zhang, 2011; Pande & Bader, 2011; McLaughlin & Bader, 2018; Sariyuce et al., 2013; Bernaschi et al., 2016), these are often limited by hardware-specific optimizations, data-specific assumptions (e.g., small-world, social, or biological networks), and the use of heuristics that are tailored to specific settings rather than offering general solutions. Moreover, publicly available code is rare, making these approaches difficult to reproduce or integrate. Overall, NBC is not naturally suited for GPU implementation, as it does not rely on matrix multiplication, but is based on computing shortest path counts between all node pairs.”*
>
> We have also added the exact runtimes for the CPU versions of NBC, RA2, and CND and the GPU versions of RA2 and CND, in Table 14.
>
> Finally, for the Reviewer's interest, we also added the dismantling time for selected networks for NBC on CPU and GPU (hardware specifications can be found in Appendix N), in Table 15.
>
> **- Table 14: Average runtime (in seconds) by field and method for dynamic dismantling. Evaluated on networks of up to 23,000 nodes and 507,000 edges (n = 1,475). Average number of nodes N and number of edges E by field. In bold the fasted method.**
>
> Field | N$_{mean}$|E$_{mean}$|CND-CPU | CND-GPU |NBC-CPU |RA2-CPU |RA2-GPU |
> | :--- |:---:|:---:|:---:|:---:| :---:|:---:|:---:|
> Biomolecular |2,997 |11,855 |1,688.9|37.7 |174.5 |3,699.8 |**29.4** |
> Brain |97 |1,535 |7  |4.2 |**0.2** | 7.7 |2.4 |
> Covert |107 |266 |7.3 |0.9 |**0.2** |19.8 |0.6 |
> Foodweb |117 |1,087 |24.2 |2.4 |**0.2** |25.4 |1.9 |
> Infrastructure |664 |1,332 |2,610.8 |34.9 |**11.6** |2,441.8 |27 |
> Internet |5,708 |19,601 |6,149.2 |34.2 |138.6 |9,801.8 |**31.9**|
> Misc |2,880 |19,921 |3,641.8|54.9 |439.3 |5,065.1 |**53.2** |
> Social |3,267 |53,977 |8,322.5 |**149.4**|4,840.6 |12,474.4 |161 |
>
>
> **- Table 15: Runtime (in seconds) for NBC run on CPU (graph-tool) and GPU (cuGraph) on a subset of networks. Number of nodes N and number of edges E. In bold the fasted method.**
>
> ||N|E|NBC-GPU|NBC-CPU|
> | :--- |:---:|:---:|:---:|:---:|
> Foodweb Blackrock|86|375|4.7|**0.04**|
> Phonecall 2012|193|1,030|20.7|**0.08**|
> Rat Transcription 2010|524|1,081|37.9|**0.10**|
> Roadmap Winnipeg|1,040|1,595|848.3|**0.41**|
>
> **3. Runtime comparisons underrepresent [...] approximate betweenness baselines, weakening claims about practical efficiency.**
>
> We appreciate the Reviewer highlighting this crucial comparison. To address this, we have conducted new experiments comparing our methods against a simple NBC approximation strategy. We have documented these findings in a new section, Appendix K: NBC Approximators, and added Table 16 to report the specific results.

---

> ### Author Response · Authors · 2025-11-21
> **Answer to Reviewer jELD 4/6**
>
> *“While the high computational cost of Node Betweenness Centrality (NBC) has motivated the development of numerous approximators (Bader et al., 2007; Bergamini & Meyerhenke, 2015; Haghir Chehreghani, 2013; Riondato & Kornaropoulos, 2014), comparing against them is challenging due to the scarcity of standardized, publicly available code and the complexity of their sampling algorithms, which are often performant only for specific domains or incompatible with disconnected graph structures.
> To address this, we implemented two standard randomized pivoting strategies for approximation. NBC-20 estimates betweenness centrality using a random sample of 20% of the nodes. NBC-log uses a random sample of 10* log2(N) nodes. NBC-20 prioritizes accuracy by always scanning a fixed slice of the network (20%), whereas NBC-log prioritizes speed by scanning a much smaller, logarithmically scaled subset that grows very slowly as the network gets larger. We evaluated these baselines against the exact NBC and our CND method on a subset of 157 networks selected for their size (Nmin=2235, Nmax=9885, Emin=10075, Emax=506437), where exact NBC calculation begins to become computationally expensive.*
>
> *Table 15 reports the dismantling performance (AUC) and total runtime, averaged by field. First, we see that the NBC approximators perform comparably to the exact NBC, even occasionally outperforming it. However, this performance increase of NBC approximators should not be overstated due to the smaller sample size of this experiment. Second, while NBC approximators are significantly faster than exact NBC, CND remains faster than both approximation methods in almost all domains. A notable exception occurs in the Infrastructure field. Here, the usually slowest method NBC is actually the fastest in terms of total runtime because it takes a significantly lower number of removals to dismantle the network. Consequently, even though CND is faster per step, the NBC-based methods result in a lower total runtime simply because the dismantling threshold is reached much earlier.*
>
> *Finally, it is critical to distinguish the theoretical foundations of these approaches. Existing NBC approximators focus on accelerating the estimation of a global metric. In contrast, LGD-NA leverages purely local topological information to directly estimate pairwise distances in the latent metric space. This distinction allows LGD-NA to bypass the need for global knowledge or more complex sampling strategies. Although approximation techniques improve NBC's speed, their reliance on sampling global paths is inherently less efficient than our strictly local approach, as we validate in Table 16. Furthermore, sampling global information remains vulnerable to missing data and adversarial noise. The strength of LGD-NA, therefore, lies in its ability to achieve high dismantling performance by directly utilizing local geometric insights, rather than attempting to approximate a computationally intensive global metric.”*
>
> **- Table 16: Average runtime (in seconds) and LCC AUC by field and method for dynamic dismantling. Evaluated where Nmin = 2, 235, Nmax = 9, 885, Emin = 10, 075, Emax = 506,437, (n = 157). In bold the fastest method per field. Average number of nodes N and number of edges E by field.**
>
> Field|N$_{mean}$|E$_{mean}$|Method|Running time (sec)|AUC
> | :--- |:---:|:---:|:---:|:---:|:---:|
> Biomolecular|5,528|26,804|NBC-20|437 |0.081
> ||||NBC|453|0.082
> ||||**CND**|**86**|0.087
> Infrastructure|5,803|15,068|**NBC**|**76**|0.021
> ||||NBC-20|105|0.022
> ||||CND|203|0.099
> Internet|6,623|31,292|NBC-20|49|0.016
> ||||NBC|196|0.016
> ||||**CND**|**29** |0.02
> Misc|5,444|53,289|NBC|1,220|0.106
> ||||NBC-20|655|0.106
> ||||**CND**|**121**|0.134
> Social|5,778|156,918|NBC-20|1,368 |0.274
> ||||NBC|14,961|0.277
> ||||**CND**|**405**|0.325
>
> **4. Would hybrid CPU GPU pipelines change the outcome?**
>
> We clarify that a hybrid pipeline would not alter the outcome for NBC, primarily because the bottleneck of NBC (shortest-path counting) does not lend itself to efficient CPU-GPU splitting.
>
> We also highlight that our LGD-NA framework operates as a hybrid pipeline: the sequential logic of the dismantling process (node selection and removal) is handled by the CPU, while the computationally intensive estimation of latent geometry (matrix multiplication) is offloaded to the GPU. NBC cannot benefit from this architecture because it lacks the parallelizable matrix operations that make GPU offloading effective.
>
> We have added the following summary to Appendix J to clarify this:

---

> ### Author Response · Authors · 2025-11-21
> **Answer to Reviewer jELD 5/6**
>
> *"It is important to note that our LGD-NA implementation inherently utilizes a hybrid workflow: the sequential dismantling logic is managed by the CPU, while the expensive latent geometry estimations (relying on matrix multiplication) are offloaded to the GPU. This architecture is highly effective for LGD-NA but is not applicable to NBC. For NBC, the core computational burden is the calculation of all-pairs shortest paths, a task that does not lend itself well to GPU computations, meaning a hybrid pipeline yields no significant performance gain, and thus solely runs on CPU.”*
>
> **5. Mean field ranking and a single ten percent threshold may hide domain specific behavior, and guidance on when CND versus RA2 is preferable is limited.**
>
> We appreciate the Reviewer's insight regarding the potential masking of domain-specific behaviors. To address this, we have conducted an analysis of the "pure win rate" (excluding draws) for each of our LGD-NA methods across every individual field. We have included these results as Table 18. Specifically, we have added the following commentary to Appendix M:
>
> *"Our analysis of pure win rates (draws are excluded) in Table 18 reveals distinct domain-specific strengths. CND achieves the highest win rate in Biomolecular, Foodweb, Infrastructure, Internet, and Social networks. In contrast, RA2 is the preferred method for Brain and Covert networks. Notably, RA2num does not emerge as the top-performing method in any of the tested domains."*
>
> Finally, we address the robustness of the ranking regarding the 10% threshold in point 7 below.
>
> **- Table 18:  Pure win rate (draws excluded), for LGD-NA measures, without reinsertion ($n=1{,}296$). In bold the method with the highest win rate per field.**
>
> ||CND|RA2|RA2$_{num}$
> | :--- |:---:|:---:|:---:|
> Biomolecular|**52%**|30%|17%
> Brain|28%|**68%**|4%
> Covert|16%|**42%**|41%
> Foodweb|**67%**|16%|16%
> Infrastructure|**65%**|16%|19%
> Internet|**52%**|46%|2%
> Misc|41%|**45%**|14%
> Social|**51%**|29%|20%
>
> **6. Could you clarify whether CND or RA2 actually outperform NBC in accuracy across the dataset or whether their advantage is primarily speed, and include a consolidated table with AUC gaps with and without reinsertion.**
>
> We thank the reviewer for identifying the ambiguity in our wording regarding the top-performing algorithm. We intended to convey that LGD-NA outperforms all other dedicated dismantling algorithms (GDM, CoreGDM, GND, CI, EI, Min-Sum). As for centrality metrics used for dismantling, Node Betweenness Centrality (NBC) performs the best overall, with LGD-NA following closely in second place. The main limitation of using NBC in real-world scenarios is its inherent inefficiency (O(VE) time complexity and cannot be accelerated by GPUs), making our GPU-optimized LGD-NA framework the most practical solution for large-scale networks.
>
> We now resolve this ambiguity by revising Lines 408 and 508 to explicitly specify *"dismantling algorithms."* Furthermore, we refined the fourth and final paragraph of Subsection 4.4  to read:
>
> *“LGD-NA consistently outperforms all other non-latent geometry-driven dismantling algorithms, including those relying on spectral Laplacian-based methods and machine learning. The only measure that still outperforms LGD-NA is the Node Betweenness Centrality (NBC) metric (which is also latent-geometry-driven), applied to dynamic dismantling. These results strongly demonstrate the practical reliability of our latent geometry-driven dismantling framework, LGD-NA.”*
>
> We clarify that NBC consistently outperforms CND and RA2 in terms of AUC. As demonstrated in Table 36, NBC maintains this advantage across all scenarios: without reinsertion, with specific reinsertion methods, and when the optimal reinsertion method is selected for each algorithm. However, NBC is very slow, making our GPU-optimized LGD-NA framework the best viable option for larger networks.
>
> **- Table 36: LCC AUC by field for CND and NBC without and with different reinsertion techniques (R1, R2, R3). In bold the best performing method.**
>
> ||CND|NBC|
> | :--- |:---:|:---:|
> Biomolecular|0.072|**0.058**
> Brain|0.388|**0.362**
> Covert|0.122|**0.110**
> Foodweb|0.215|**0.195**
> Infrastructure|0.048|**0.040**
> Internet|0.058|**0.049**
> Misc|0.204|**0.171**
> Social|0.232|**0.204**
>
> ||CND-R1|NBC-R1
> | :--- |:---:|:---:|
> Biomolecular|0.056|**0.049**
> Brain|0.381|**0.368**
> Covert|0.116|**0.105**
> Foodweb|0.210|**0.197**
> Infrastructure|0.045|**0.041**
> Internet|0.057|**0.053**
> Misc|0.186|**0.173**
> Social|0.198|**0.188**
>
>
>
> ||CND-R2|NBC-R2
> | :--- |:---:|:---:|
> Biomolecular|0.057|**0.048**
> Brain|0.380|**0.366**
> Covert|0.115|**0.104**
> Foodweb|0.211|**0.196**
> Infrastructure|0.045|**0.041**
> Internet|0.057|**0.053**
> Misc|0.187|**0.171**
> Social|0.198|**0.185**
>
> ||CND-R3|NBC-R3
> | :--- |:---:|:---:|
> Biomolecular|0.057|**0.048**
> Brain|0.380|**0.366**
> Covert|0.115|**0.104**
> Foodweb|0.211|**0.196**
> Infrastructure|0.045|**0.041**
> Internet|0.057|**0.053**
> Misc|0.187|**0.171**
> Social|0.198|**0.186**

---

> ### Author Response · Authors · 2025-11-21
> **Answer to Reviewer jELD 6/6**
>
> **7. How sensitive are the conclusions to thresholds other than ten percent and to alternative fragmentation metrics, especially when stratified by domain or graph statistics.**
>
> We fully agree with the reviewer that more dismantling thresholds should be evaluated to solidify our results. To address the concern regarding sensitivity, we have expanded our analysis to evaluate performance at varying thresholds. We have added Table 19, which displays the mean field ranking at 10%, 25%, and 50% thresholds. As illustrated, the rankings remain broadly consistent across these levels, indicating that the relative performance of the methods is robust and not sensitive to the specific choice of the 10% cutoff. We have added the following analysis to Appendix M:
>
> *“Robustness to Threshold Variations: As shown in Table 19, the mean field rankings of the methods are broadly consistent across removal thresholds of 10%, 25%, and 50%. While permutations occur, the dominance of NBC and CND is consistent across different thresholds, confirming that our conclusions are not artifacts of the 10% threshold."*
>
> - **Table 19: Mean field ranking for different threshold levels (n = 1,296). In Bold the best method by threshold.**
>
> ||10%|25%|50%
> | :--- |:---:|:---:|:---:|
> NBC|**1**|**1**|**1**
> CND|3.625|3.5|3.25
> RA2|4 |3.875 |5.125
>
> We once again thank the Reviewer for their valuable feedback. We have ensured that all points raised are fully addressed and clarified in the revised manuscript. We welcome any further questions regarding our response.

---

> ### Author Response · Authors · 2025-11-25
> **Additional answer: alternative fragmentation metrics**
>
> Dear Reviewer,
>
> We are writing to provide a supplementary update regarding your question: **"How sensitive are the conclusions to thresholds other than ten percent and to alternative fragmentation metrics, especially when stratified by domain or graph statistics."**
>
> We have added a new subsection (Section 4.7) to the main text with Figure 4 (https://anonymous.4open.science/r/15012-3E88/figure_04.png). This subsection validates the effectiveness of our LGD-NA framework for dismantling using domain-specific performance indicators (different from the standard LCC AUC used in the study) across four real-world networks.
>
> - **Drosophila Connectome**: Metric is the Sensory Neuron Firing Rate.
> - **Paris/Brussels Terrorist Cell**: Metric is Commander Reachability.
> - **Ryanair Flight Map**: Metric is Global Efficiency.
> - **School Contact Network (Epidemics)**: Metric is Final Outbreak Size (SEIR Model).
>
> The results in Table 13 demonstrate a strong consistency that strategies identified as effective by our original evaluation metric (LCC AUC) are equally effective when measured by these diverse functional metrics. This confirms the broad utility and predictive power of our LGD-NA methods across varied real-world scenarios.
>
> Below is the new subsection added to the manuscript.
>
> *“ 4.7. Real-World Applications: Fault Tolerance, Security, and Communications*
>
> *To demonstrate the practical utility of LGD-NA, we evaluate its performance on four distinct real-world systems using domain-specific functional metrics (full experimental details in Appendix I).*
>
> - *Drosophila Connectome (Shiu et al., 2024) (Fault Tolerance): We utilize a Spiking Neural Network (SNN) model of the sugar-sensing circuit. The metric is the sensory neuron firing rate required to trigger the proboscis extension response.*
>
> - *Paris/Brussels Terrorist Cell (Gutfraind and Genkin, 2017) (Security & Communications): We analyze the network responsible for the 2015 Paris and 2016 Brussels attacks. The metric is Commander Reach, defined as the percentage of operatives able to communicate with at least one of the three key commanders.*
>
> - *Ryanair Flight Map (Cardillo et al., 2013) (Fault Tolerance): A transportation network where we measure Global Efficiency ($E_{glob}$).*
>
> - *School Contact Network (Mastrandrea et al., 2015) (Security/Epidemics): We simulate an epidemic using an SEIR model (Anderson and May, 1991). The metric is the Final Outbreak Size.*
>
> *Our results in Figure 4 show that dismantling strategies effectively degrade the functional performance across all four systems. In the Drosophila Connectome and Paris/Brussels Terrorist Cell, we observe particularly sharp drops in performance metrics after removing only a small fraction of nodes (~5%). We observe a more gradual deterioration in the global efficiency of the Ryanair Flight Map and the viral spread within the School Contact network. This functional collapse is particularly significant for the two adversarial scenarios (Paris/Brussels Terrorist Cell and School Contact Network): it confirms that LGD-NA is effective for security and communication disruption, efficiently suppressing epidemic outbreaks and isolating hostile leadership with minimal intervention. [...]*
>
> *For the Drosophila Connectome, the analysis informs the resilient and redundant design of fault-tolerant neuromorphic circuits by mimicking its biological wiring (Suarez et al., 2021, Hame et al., 2021). In the Ryanair Flight Map, it identifies specific hubs where reinforcement prevents systemic failure. Finally, for adversarial networks (Paris/Brussels Terrorist Cell and Epidemic), our robustness analysis serves a diagnostic purpose when faced with incomplete data. Since social networks, and especially covert ones, often contain unobserved links (e.g., dormant ties or unreported contacts), calculating an empirical robustness ceiling allows us to estimate the margin of error required for successful security operations with partial observability.”*
>
> **- Table 13: Network-specific evaluation metric and the original LCC AUC performance metric for four real-world networks, for NBC, CND, and RA2.**
>
> |Drosophila Connectome|LCC AUC|Firing Rate (Freq. = 100 Hz) AUC|
> |:--- |:---:|:---:|
> ||||
> NBC|**0.410**|**0.030**|
> CND|0.454|0.093|
> RA2|0.459|0.042|
>
> |Paris/Brussels Terrorist Cell|LCC AUC|Commander's Reach AUC|
> | :--- |:---:|:---:|
> ||||
> NBC|0.123|0.182|
> CND|0.125|0.149|
> RA2|**0.114**|**0.095**|
>
> |Ryanair Flight Map|LCC AUC|Global Efficiency AUC|
> | :--- |:---:|:---:|
> ||||
> NBC|0.149|0.040|
> CND|0.143|**0.039**|
> RA2|**0.142**|0.040|
>
> |School Contact|LCC AUC|Final Outbreak Size AUC|
> | :--- |:---:|:---:|
> ||||
> NBC|**0.359**|0.351|
> CND|0.420|0.288|
> RA2|0.418|**0.285**|

---

> ### Comment · Reviewer_jELD · 2025-11-26
> **Response to the Authors**
>
> Thanks for the responses. My initial evaluation remains unchanged at this stage.

---

> > ### Author Response · Authors · 2025-12-03
> >
> > We thank the Reviewer for reading our replies to their questions. We would like to clarify, correct, and add to our previous answers; which we acknowledge might have caused some confusion and prevented us from fully addressing their questions.
> >
> > 1. We corrected Table 14, as there were some rendering issues in the initial response.
> >
> > 2. We separated the different tables in Point 6 to make the comparison between NBC and CND clearer for different reinsertion methods.
> >
> > 3. We made Figure 7 clearer. We now clearly see how our estimators identify the nodes at the centre of the hyperbolic disk (https://anonymous.4open.science/r/15012-3E88/figure_07.png).
> >
> > 4. We made additional analysis on how our estimators were able to estimate the true geometric distance in the nPSO networks (a network with a known hyperbolic geometry). We have added Figure 8 (https://anonymous.4open.science/r/15012-3E88/figure_08.png) and Tables 4 (https://anonymous.4open.science/r/15012-3E88/table_04.png) and 20 explaining these new experiments . We believe this validates in an even stronger manner our claims about our latent-geometry estimators. Below are the new paragraphs in Appendix C that explain these experiments:
> >
> > *“To quantitatively support our claim, we evaluate how well the latent geometry estimators approximate the true hyperbolic distances. We use the hyperbolic distance correlation (HD-correlation) metric, the Pearson correlation between all pairwise geometrical shortest path distances in the networks’ original hyperbolic space and the weighted shortest path distances using the latent-geometry estimators as edge weights (Muscoloni et al., 2017). The higher this correlation, the better the latent-geometry estimator is able to recover the geometrical distances between pairs of nodes in a network’s underlying geometry.*
> >
> > *Table 4 shows a high HD-correlation for both CND and RA2 across all tested nPSO configurations, confirming that these measures used in our dismantling framework are effective latent geometry estimators. This is further supported by the statistical significance reported in Table 20.*
> >
> > *The Pearson correlation is visualized in Figure 8 for different parameters, visualizing how well the distance approximation changes as the network becomes less hyperbolic. As expected, for $\gamma=2$, the correlation decreases for both estimators with increasing temperature (i.e., reduced clustering and hyperbolicity). For the less hyperbolic $\gamma=3$ networks, this decreasing trend persists for CND but not for RA2. This suggests that CND remains a robust estimator of the latent geometry even when hyperbolic structure is less pronounced, whereas RA2's performance is more dependent on strongly hyperbolic conditions, consistent with our dismantling experiments.*
> >
> > *This visual and quantitative evidence demonstrates our LGD-NA measures' ability to accurately estimate the geometric distance between nodes. Consequently, the node aggregation step in our LGD-NA framework can successfully identify nodes that connect distant regions in the latent space.”*
> >
> > 5. We made additional runtime experiments that can better answer the Reviewer’s questions around GPU acceleration. We used the nPSO model in a controlled environment to determine when GPU acceleration provided a meaningful advantage. Our results show that CND on GPU begins to outperform NBC on CPU over approximately 1,000 nodes or 100,000 edges. We have added Figures 16 (https://anonymous.4open.science/r/15012-3E88/figure_16.png) and 17 (https://anonymous.4open.science/r/15012-3E88/figure_17.png) and Table 37 to Appendix J to visualize these findings. Appendix J now contains the following paragraph:
> >
> > *“To empirically validate these runtime advantages in a controlled setting, we conducted additional experiments using the nPSO model (Muscoloni & Cannistraci, 2018) with network sizes ranging from 10 to 5,000 nodes and densities of 4\%, 8\%, and 20\%. We keep the temperature (lower temperature yields higher clustering) fixed (T=0.3) and adjust the number of communities to suit the size of the network ($C=2$ for $N \in \{10, 50, 100\}$, $C=5$ for $N \in \{500, 1,000\}$, $C=10$ for $N \in \{5,000\}$). Our results demonstrate that GPU-accelerated LGD-NA methods begin to show running time advantages over NBC-CPU when networks exceed approximately 1,000 nodes (see Figure 17) or 100,000 edges (see Figure 16). This  threshold aligns with our observations in real-world networks (see Figures 3 and 18, and Table 14), where GPU methods achieve superior running times for larger-scale networks such as biomolecular, internet, and social networks, while offering no runtime benefit for smaller networks where CPU implementations remain efficient.”*
> >
> > 6. Finally, we have uploaded all the referenced figures and tables at the following link, for quick and easy access for the benefit of the Reviewer: https://anonymous.4open.science/r/15012-3E88.

---

### Official Review · Reviewer_vDje · 2025-10-31

**Soundness:** 3
**Presentation:** 3
**Contribution:** 3
**Rating:** 6
**Confidence:** 2

**Summary:**

The submission makes a contribution in the area of network dismantling, which largely has to do with understanding vulnerabilities of large and complex networks.  The paper proposes a new framework that provides insight into the geometry of the latent manifold and applies it to a wide range of networks with excellent performance.

**Strengths:**

The paper is quite well written.  The experimental study also appears to be extensive with accompanying code that will be freely distributed.

The experimental study is extensive and compares to a lot of other methods.

**Weaknesses:**

Limitations were not discussed in the main body and relegated to the appendix.  In the revision, it is important to include at least a brief discussion of the challenges and limitations of the contribution in the main paper.

**Questions:**

Although I do not work in the area of network dismantling, a classical approach that is inherently geometric and is also known to be powerful for robust network design is spectral graph theory, leveraging the spectra of graph Laplacians.  Is there any existing work on the use of Laplacians for this task?

**Details Of Ethics Concerns:**

The area of the work deals with network vulnerabilities and attacks, which may have ethical implications.  These are discussed in the appendix and appear to me to be sufficient.

---

> ### Author Response · Authors · 2025-11-14
>
> We thank the reviewer for their insightful and positive feedback, and for highlighting the strengths of our work, including its contribution to the network dismantling field and the comprehensive experimental validation.
>
> **1. Limitations were not discussed in the main body and relegated to the appendix.**
>
> We thank the reviewer for this valuable comment. We completely agree that it would be beneficial to the paper to include a discussion of the limitations in the main text. As suggested, we will add the following paragraph to the main text, which summarizes the limitations of our work:
>
> *“Our study has two main limitations. First, hardware constraints precluded testing on extremely large networks, though our results are validated across 1,475 real-world networks across a vast range of disciplines. Second, while practical runtimes could deviate from theoretical expectations, all methods were executed under identical hardware and optimization settings to ensure fair comparison.*
>
> *In addition, we acknowledge the dual-use potential of this research, as understanding network vulnerabilities is critical for both designing targeted attacks and engineering robust defensive strategies. To mitigate this, we proactively demonstrate a constructive application for enhancing network robustness and believe the societal benefit of openly publishing these defensive tools outweighs the risk of misuse.”*
>
> **2. Is there any existing work on the use of Laplacians for this task?**
>
> We thank the reviewer for this insightful question. The reviewer is completely correct that spectral graph theory, particularly methods leveraging the graph Laplacian, represents a powerful approach to network analysis.
>
> To address the reviewer's query, such methods do indeed exist, and we included a prominent one, Generalised Network Dismantling (GND), as a key baseline in our study. GND is a spectral-cut strategy that operates by leveraging the spectrum of the graph Laplacian. It specifically uses the Fiedler vector (the eigenvector associated with the second-smallest eigenvalue) to identify weak links and efficiently fragment the network.
>
> Our experimental results show that our proposed framework outperforms the GND method in all tested scenarios (both with and without reinsertion, and across all fields).
>
> We agree that this comparison is important and could be made more explicit in the main text. We will revise lines 22, 93, and 479 to clarify that our method's state-of-the-art performance is demonstrated not only against machine learning-based methods but also against powerful spectral Laplacian-based approaches like GND.
>
> We again thank the reviewer for their valuable feedback and will ensure these two points are included and clarified in the main article.

---

> > ### Author Response · Authors · 2025-11-17
> >
> > We have just uploaded a new version of the manuscript with the two changes mentioned in the previous comment. We apologize not to have done this directly at the same time.
> >
> > We hope the changes are clear enough to the reviewer and we welcome any further questions or feedback regarding our response.
> >
> > Sincerely,
> >
> > The Authors of Paper 15012

---

### Author Response · Authors · 2025-12-03
**General Response for All Reviewers and Area Chairs 1/2**

We thank all the reviewers for their deep, insightful feedback and very useful comments, which have consequently made the paper much stronger.

We are pleased that the reviewers recognise our LGD-NA framework as an *"intuitive”* (jELD), *"novel approach"* (a3SW), offering *"a clean and useful angle"* (MAH5). The core method of estimating geometric distances on a network’s latent manifold is confirmed to *"expose critical structural information"* (a3SW) and is *"highly effective"* (a3SW). Our experimental setup was confirmed to be *"extensive"* (vDje, a3SW) across a *"large-scale, including 1,475 networks across 32 domains"* (MAH5). Our simplest latent-geometry measure, CND, is found to be *“highly effective”* (a3SW), our GPU implementation provides *"speedups that make the approach scalable"* (jELD) and our robustness-engineering method is *“easy”* to use (jELD). Reviewers also commend the paper's clarity, noting it is *"quite well"* (vDje) and *“clearly written”* (MAH5), with an *"explicit"* pipeline and metrics (MAH5), and a *"clear LCC AUC metric and protocol"* (jELD).

To provide greater clarity on the revisions made to our paper and the experiments we conducted to address the reviewers' questions, we have summarized the modifications and experiments made during the rebuttal period below. Note that all figures and tables referenced in the rebuttal responses can be easily accessed through this anonymous link: https://anonymous.4open.science/r/15012-3E88.

**Additional experiments:**

- **Functional Analysis:** Conducted additional experiments on four representative real-world networks (Drosophila Connectome, Flight Map, Terrorist Cell, Contact network (epidemics)), using specific functional metrics, motivating the applicability of our dismantling and robustness-engineering methods to various real-world scenarios that represent key applications in Fault Tolerance, Communications and Security (added Subsection 4.7, Figure 4, Table 13, and Appendix I) (a3SW, jELD26)
- **Geometric validation of LGD-NA estimators:** Provided visual and empirical evidence for the ability of our latent-geometry estimators to estimate true geometric distances, and thus identifying node importance. Used an artificial ground truth geometric generative network model, the nPSO network model, which has known underlying geometry to validate our estimators (added Appendix C, Figures 7, 8, and Tables 4, 20, 21, 22). (a3SW, jELD26)
- **Directed Networks:** Modified our network dismantling algorithm to be applicable on directed networks, and showed that our method, treating networks as undirected, still outperforms this directed variant as well as the SOTA dismantling algorithm designed specifically for directed networks (added Appendix L, Table 17). (MAH5)
- **NBC Approximators:** Implemented NBC approximators (for runtime speed up), showing that our LGD-NA methods still have a runtime advantage over them (added Appendix K, Table 16). (jELD26, MAH5)
- **GPU advantage:** Ran experiments on synthetically generated networks by the nPSO model to show at what network size does our LGD-NA GPU-based methods offer a runtime advantage (> 1,000 nodes and >100,000 edges). (jELD26, MAH5)
- **NBC GPU:** Ran experiments on NBC-GPU to quantitatively show its underperformance versus NBC-CPU (Appendix J, Table 15). (jELD26, MAH5)
- **Sensitivity to Dismantling Thresholds:** Ran experiments for various dismantling thresholds (10%, 25%, 50%) showing that the ranking of evaluated methods is stable across different terminal conditions(Appendix M, Table 19). (jELD)

---

### Author Response · Authors · 2025-12-03
**General Response for All Reviewers and Area Chairs 2/2**

**Clarifications:**

- Explained in detail the distinction between topological graph metrics and the underlying latent manifold in a new Appendix section (B). (a3SW)
- Defined more clearly the term “robustness”, and demonstrated the effectiveness and practical use of our robustness-engineering method thanks to our additional functional experiments, in Appendix H. (a3SW)
- Clarified the purpose and applicability of dismantling, especially when applied to Fault Tolerance, Communications and Security, in the Introduction and in Subsection 4.7, with the supporting evidence from our new functional analysis. (a3SW)
- Motivated the use of our evaluation metric, LCC AUC, and showed how it closely follows other functional metrics we tested in the functional analysis (Appendix M). (a3SW)
- These functional metrics also confirmed the effectiveness of our methods with different fragmentation/evaluation metrics (Table 13). (jELD)
- Provided field-specific analysis on when RA2, CND, or RA2-num is preferable through per-field win rates in Appendix M. (jELD)
- Clarified why NBC GPU is conceptually slower than NBC CPU, and provided quantitative evidence for it (Table 15). (jELD26, MAH5)
- Explained in Appendix O why weighted graphs were not considered and their potential as a future research direction. (MAH5)
- Explicitly explained the limitations of our work in the main text, whereas previously we discussed limitations in the Appendix. (vDje)
- Clarified in the main text that our methods do outperform spectral Laplacian-based methods. (vDje)
- Clarified that our methods are parameter-free, since they only operate on the topology of the network, and this involves no (parameter) tuning of any sort. (MAH5)
- Discussed the possibility of CPU-GPU hybrid workflows for both NBC (not possible) and CND (already implemented) (Appendix J). (jELD26)
- Clarified quantitative dismantling results for all dismantling methods, reinsertion methods and confirmed the stability of rankings across different dismantling thresholds, reinsertion methods, and evaluation metrics (Figures 10, 11, 12, Tables 8, 36). (jELD26, MAH5)

---

### Meta-Review · Area_Chair_xjkR · 2026-01-07

**Summary:**

This paper studies the problem of network dismantling. Which is the process of removing nodes and edges to structurally disrupt the graph. In this case to take a connected graph and make it highly disconnected.

The initial scores for the paper are most borderline and leaning reject with scores of 6 (vDje), 6 (jELD), 4 (MAH5), 2 (a3SW). The main concerns of the reviewers are

1. Since the problem can be formulated as a theoretical computer science problems. Reviewers were expecting theoretical analysis of the proposed method (jELD)

2. Reviewers also have concerns on the reported metric AUC of mean connected component size and of the time taken by the different methods, specifically against methods that approximate betweenness measures. (jELD, MAH5). Reviewer a3SW also has concerns about the abstractness of the metrics.

3. Reviewers also have concerns about the use of geometry in the paper (a3SW). Specifically distinguishing latent manifold notions from graph-topological descriptors.

4. The scope is limited. Only focused on small undirected graphs (MAH5). Weighted graphs and large graphs are not considered

**Reviewer Concerns:**

### Addressed

The authors substantially addressed the main experimental and positioning concerns. They added

1. Functional, domain-specific dismantling validations on four real systems (new Subsection 4.7 / Appendix I), showing that dismantling degrades system performance under task-relevant metrics and that their robustness intervention improves both topological and functional metrics.

2. Threshold-sensitivity analyses at 10/25/50% showing the mean-field rankings are broadly consistent, mitigating the concern that conclusions were an artifact of the 10% cutoff.

3. Stronger runtime/baseline coverage, including explicit CPU/GPU runtime tables and NBC approximation baselines.

4. Clearer conceptual framing distinguishing graph metrics vs latent manifolds

5. Added directed graph experiments

### Unaddressed

However, the paper is still completely empirical with no theoretical guarantees and the method is still only tested on small graphs.

**Reviewer Scores:**

Reviewers vDje and jELD were borderline accept initially and I expect them to stay there Note that both reviewers have low confidence.

Reviewer MAH5 was borderline reject and during the discussion they increased their score to borderline accept.

Finally, Reviewer a3SW was voting reject with a high confidence (4). Given the responses. I believe it is likely that the reviewer would increase their score to a 4.

As such the paper is borderline. I recommend accept. The paper presents a simple, general dismantling framework with unusually broad empirical coverage, and the revision adds the missing pieces reviewers asked for, task-relevant functional validation, robustness to threshold choice, and stronger compute/baseline comparisons (including NBC approximators).  The remaining issues (lack of guarantees; weighted/directed extension) are important but, in my view, do not preclude acceptance given the strengthened evidence and clearer scope statements

---

### Decision · Program_Chairs · 2026-01-26

Accept (Poster)